# Doubly Robust Distributionally Robust Offline Contextual Pricing

**Min Xu**[1]  **Xinyi Yin**[2]  **Yunfan Zhang**[3]  **Yuxuan Han**[3]  **Houcai Shen**[1]  **Caihua Chen**[1]

## Abstract

Offline contextual pricing often relies on logged observational data, but faces challenges from distributional shifts between training and deployment environments. Distributionally robust optimization (DRO) provides a principled approach to off-policy evaluation and learning (OPE/L). However, existing methods are mostly limited to discrete actions. Recent work has explored DRO for continuous treatments using inverse propensity weighting (IPW), while such IPW-based estimators can be sensitive to the convergence rate of propensity score estimates, particularly when estimated non-parametrically, which may lead to larger estimation errors and regret. In this work, we develop a doubly robust (DR) framework for distributionally robust OPE/L in continuous pricing settings. For evaluation, we propose a localized DR estimator that addresses the computational challenges of worst-case expectations by fitting only a small number of regressions, comparable to standard non-robust DR, while achieving semiparametric efficiency under mild product rate conditions. For learning, we leverage the inherent smoothness of demand noise to handle pricing-specific discontinuities in revenue outcomes (e.g., threshold-based purchase decisions), establishing a finite-sample regret bound of $\tilde{\mathcal{O}}_p(T^{-s/(2s+1)})$ for smoothness orders $s = 1, 2$. This bound improves upon existing regret rates in existing DRO-based off-policy learning (OPL) for continuous treatments. Extensive experiments under various levels of distribution shift validate our proposed framework.

## 1. Introduction

Personalized pricing has gained popularity on online platforms, as firms can account for consumer heterogeneity by setting prices based on individual contextual information (Cohen et al., 2020; Chen & Gallego, 2021; Tang et al., 2025; Zhang et al., 2025). While randomized controlled trials (RCTs) provide a reliable way to estimate treatment effects, see Colnet et al. (2024) for a review, the large number of possible price options (Ye et al., 2025) and the risks associated with real-time exploration often make off-policy evaluation and learning (OPE/L) from historical observational data a more practical and safer approach (Kallus & Zhou, 2018; Kitagawa & Tetenov, 2018; Athey & Wager, 2021; Zhou et al., 2023; Carranza & Athey, 2025).

Standard inverse probability weighting (IPW) (Horvitz & Thompson, 1952; Swaminathan & Joachims, 2015) and doubly robust (DR) estimators (Dudík et al., 2011; Zhao et al., 2015), have been well-developed for discrete treatment settings (Jiang & Li, 2016; Athey & Wager, 2021; Mbakop & Tabord-Meehan, 2021; Zhan et al., 2021; Fang et al., 2025b). Generally, direct application of these methods to continuous treatment settings is challenging, as it would reject most observed data due to the zero-probability of exact treatment matches, leading to high variance. To address these limitations, recent work has introduced kernel-based smoothing techniques that enable more stable estimation in continuous domains (Kallus & Zhou, 2018; Lee et al., 2022; Colangelo & Lee, 2026; Ai et al., 2026). However, even with kernel smoothing, OPE/L remains vulnerable to distributional shifts between training and deployment environments, which are common in dynamic markets due to concept drift or covariate shifts (Si et al., 2023; Wang et al., 2024; 2025). Distributionally robust optimization (DRO) provides a principled framework to mitigate such risks by optimizing for worst-case performance within an uncertainty set (Delage & Ye, 2010; Ben-Tal et al., 2013; Hu & Hong, 2013). DRO has been successfully applied to OPE/L in discrete treatment settings (Kallus et al., 2022; Si et al., 2023; Wang et al., 2025). For continuous treatments, recent works have extended DRO using kernel-smoothed IPW estimators (Leung et al., 2025), which provide finite-sample convergence guarantees. We summarize the comparison between our result and prior works in Table 1. Nevertheless, these IPW-based approaches remain dependent on the convergence rate of

---
[1]School of Management and Engineering, Nanjing University, Nanjing, China [2]School of Mathematics, Nanjing University, Nanjing, China [3]Stern School of Business, New York University, New York, USA. Correspondence to: Caihua Chen <chchen@nju.edu.cn>.

*Proceedings of the 43$^{rd}$ International Conference on Machine Learning*, Seoul, South Korea. PMLR 306, 2026. Copyright 2026 by the author(s).

the propensity density estimates, where nonparametric estimation can lead to deteriorated estimation error and regret.

In this paper, we propose doubly robust algorithms for distributionally robust OPE/L, incorporating a scaled kernel function to handle continuous pricing settings. Unlike general continuous-treatment literature, which typically assumes a smooth outcome function (Ai et al., 2026; Colangelo & Lee, 2026), pricing inherently involves a discontinuous revenue function due to threshold purchase behaviors (Javanmard & Nazerzadeh, 2019; Cohen et al., 2020). Recently, Kallus & Zhou (2018) requires the smoothness of the outcome functions with respect to the treatment, which is also imposed on the conditional outcome density functions in Leung et al. (2025); Colangelo & Lee (2026). Nevertheless, we handle this rigorously through the $s$-smoothness of the cumulative distribution function (CDF) of noise in the demand function, for $s = 1, 2$. Our main contributions are as follows:

**Distributionally Robust Off-policy Evaluation (OPE).** While Neyman orthogonality and $\sqrt{Th}$-consistency ($h$ is the kernel bandwidth) have been established for continuous treatment in non-robust settings (Kennedy et al., 2017; Colangelo & Lee, 2026), extending it to a DRO framework poses a non-trivial challenge, as the DRO and DR components become highly coupled by a supremum over the log of moment generating functions. We propose the Localized Doubly Robust DRO OPE for Continuous Pricing (LDR$^2$O$^2$PE-CP) estimator and show that it achieves $\sqrt{Th}$-consistency and enjoys semiparametric efficiency even under the DRO objective (see Section 3). Our approach ensures robustness to both environment shifts and estimation errors, remaining computationally tractable by fitting nuisances only three times.

**Distributionally Robust Off-policy Learning (OPL).** For policy learning, we establish a finite-sample regret bound of $\tilde{\mathcal{O}}_p(T^{-s/(2s+1)})$, $s = 1, 2$, for a policy class with finite pseudo-dimension, even when nuisance functions are nonparametrically estimated at slow rates (see Section 4). This bound represents a theoretical improvement over the $\mathcal{O}_p(T^{-1/4})$ rate established in existing distributionally robust (Leung et al., 2025) and non-robust (Kallus, 2018) approaches for continuous treatments, and is comparable to rates reported in the non-robust continuous treatment literature (Ai et al., 2026).

See `https://github.com/OfflinePricing/DR-DRO-PRICING` for simulation and Expedia study.

### 1.1. Related Work

**Off-Policy Evaluation/Learning.** A large body of work has developed OPE/L methods for discrete treatment settings (Dudík et al., 2011; Kitagawa & Tetenov, 2018; Athey & Wager, 2021; Mbakop & Tabord-Meehan, 2021; Zhou et al.,

2023; Fang et al., 2025b), continuous treatment (Kallus & Zhou, 2018; Chernozhukov et al., 2019; Colangelo & Lee, 2026; Ai et al., 2026), and general treatment (Fang et al., 2025a). Although some allow the data to be adaptively collected (Zhan et al., 2021; 2024; Jin et al., 2021; 2025b), most remain non-robust to distributional shifts.

Recent efforts have incorporated distributional robustness to handle distribution shifts, primarily in discrete treatment settings (Kallus et al., 2022; Mu et al., 2022; Blanchet et al., 2023; Si et al., 2023; Guo et al., 2024; Jin et al., 2025a; Wang et al., 2025), and some also address multi-source shifts (Carranza & Athey, 2025; Imbens et al., 2025). The closest prior work is Leung et al. (2025), which develops a distributionally robust OPL framework for continuous treatments using kernel-smoothed IPW, with performance tied to propensity density convergence rates under nonparametric estimation. Our doubly robust approach achieves semiparametric efficiency under milder product rate conditions, yields a sharper finite-sample regret bound than Leung et al. (2025).

**Contextual Pricing.** Recently, many papers have been focusing on contextual pricing, see Den Boer (2015); Chen & Gallego (2021) for a review. Offline contextual pricing is also practically relevant in real marketplaces and platform operations (e.g., Zhou et al. (2024); Zhang et al. (2026)). Existing works can be categorized by the assumptions on the two unknown functions, namely the mean utility and the CDF of the noise. Javanmard & Nazerzadeh (2019); Cohen et al. (2020); Ban & Keskin (2021); Miao et al. (2022) assume the noise CDF is known in advance, and impose a parametric model on the product's market value. Alternatively, Shah et al. (2019); Fan et al. (2024); Luo et al. (2024) allow the noise distribution also to be unknown, and use a semiparametric model where mean utility is parametrically organized. Furthermore, Nambiar et al. (2019); Chen & Gallego (2021); Hu et al. (2022); Bu et al. (2025) adopt a fully nonparametric approach to model both the mean utility function and the noise distribution. Chen et al. (2024) propose a doubly nonparametric random utility model for demand characterization, which is most relevant to our work.

## 2. Problem Formulation

### 2.1. Offline Contextual Pricing

We consider a contextual pricing setting where a context vector $X \in \mathcal{X} \subseteq \mathbb{R}^d$ and a random noise term $Z \in \mathbb{R}$ are drawn from a joint distribution $\mathbf{P}_0$. $X$ and $Z$ are independent and the marginal distributions are denoted by $F_X$ and $F_Z$, respectively. For a given context $X$, the buyer's valuation is modeled as $v(X, Z) = f(X) + Z$, where $f : \mathcal{X} \to \mathbb{R}$ is the unknown deterministic component of the valuation.

In the offline learning setting, we are given a training dataset $\mathcal{D} = \{(x_t, p_t, y_t)\}_{t=1}^{T}$ consisting of $T$ independent and i.i.d.

*Table 1.* Comparison of convergence rates in the OPL literature. We report regret upper bounds with respect to sample size $T$, treating policy class complexity measures (e.g., entropy integrals) as constant factors. For our work and Ai et al. (2026), $s = 1, 2$ corresponds to the smoothness order of the outcome functions, while Colangelo & Lee (2026) and Leung et al. (2025) require three-times differentiability ($s = 3$) of conditional outcome densities w.r.t. treatment. Here $\tilde{\mathcal{O}}$ hides logarithmic factors in $T$.

| Paper | Action Space | Distributional Shifts? | Regret Bound |
|---|---|---|---|
| Kitagawa & Tetenov (2018); Athey & Wager (2021); Mbakop & Tabord-Meehan (2021) | Binary | $\times$ | $\mathcal{O}_p(T^{-1/2})$ |
| Dudík et al. (2011); Swaminathan & Joachims (2015); Kallus (2018); Zhou et al. (2023) | Discrete | $\times$ | $\mathcal{O}_p(T^{-1/2})$ |
| Kallus & Zhou (2018); Colangelo & Lee (2026); Ai et al. (2026) | Continuous | $\times$ | $\mathcal{O}_p(T^{-s/(2s+1)})$ |
| Kallus et al. (2022); Si et al. (2023); Wang et al. (2025) | Discrete | $\checkmark$ | $\mathcal{O}_p(T^{-1/2})$ |
| Leung et al. (2025) | Continuous | $\checkmark$ | $\mathcal{O}_p(T^{-1/4})$ |
| Ours | Continuous | $\checkmark$ | $\tilde{\mathcal{O}}_p(T^{-s/(2s+1)})$ |

triples. Specifically, for each $t$, a context vector $x_t$ and a noise term $z_t$ are drawn from the underlying distribution $\mathbf{P}_0$. Subsequently, a price $p_t \in \mathcal{P}$ is selected by a known logging policy $\pi_0$ with probability $\pi_0(p_t|x_t)$, where $\mathcal{P} \triangleq [p_{\min}, p_{\max}]$ denotes the set of feasible prices. Without loss of generality, we assume the prices are normalized such that $p_{\min} > 0$ and $p_{\max} = 1$. Then, the binary purchase decision $y_t = \mathbf{1}\{z_t \geq p_t - f(x_t)\}$ is observed. Consequently, the observed revenue at time $t$ is given by $r_t = p_t y_t$. We make the following assumptions on the data-generating process.

**Assumption 2.1** (Unconfoundedness). The purchase decision $Y(p) := \mathbf{1}\{v(X, Z) \geq p\}$ is independent of the price $P$ conditional on the context $X$, i.e., $Y(p) \perp\!\!\!\perp P \mid X$.

**Assumption 2.2** (Overlap). There exists a constant $\eta > 0$ such that $\pi_0(p|x) \geq \eta$ for all $(x, p) \in \mathcal{X} \times \mathcal{P}$.

**Assumption 2.3.** There exists constants $\underline{\rho}, \overline{\rho} \in (0, 1)$ such that the joint distribution $\mathbf{P}_0$ satisfies

$$\underline{\rho} \leq \mathbf{P}_0(v(X, Z) < P \mid X) \leq \overline{\rho}, \ \forall P \in \mathcal{P}, X \in \mathcal{X}.$$

**Assumption 2.4** (Smoothness of Noise Distribution). The noise distribution $F_Z$ is $s$-differentiable with bounded $s$-order derivative, i.e., $\sup_z |F_Z^{(s)}(z)| \leq L_Z < \infty$ for $s = 1$.

Assumption 2.2 guarantees that every feasible price has a non-zero probability of being sampled for any given context (Kallus & Zhou, 2018; Si et al., 2023; Leung et al., 2025). This ensures sufficient exploration within the training data, making it possible to assess the performance of any new pricing strategy. Assumption 2.3 implies that there always exists the non-negligible probabilities that the customer will reject/accept the offer, regardless of the context. This boundedness is made to prevent our estimator from diverging to infinity. Assumption 2.4 is standard in the literature (Javanmard & Nazerzadeh, 2019; Chen & Gallego, 2021; Chen et al., 2024; Fan et al., 2024; Gong et al., 2025). And we discuss a stronger assumption that $F_Z$ has a bounded second-order derivative, $s = 2$ (Kallus & Zhou, 2018; Leung et al., 2025), which leads to an improved result in Section 4. These different smoothness assumptions align with the differentiability of the outcome regression in Ai et al. (2026).

### 2.2. Distributionally Robust Performance

A seller's policy $\pi \in \Pi$ is defined as a deterministic function $\pi : \mathcal{X} \to \mathcal{P}$ mapping contexts to prices, and the revenue is

$$r(X, \pi(X), Z) \triangleq \pi(X)\mathbf{1}\{Z \geq \pi(X) - f(X)\}.$$

We evaluate the performance of a policy $\pi$ based on its expected revenue, formalized by the policy value function:

**Definition 2.5.** The policy value function $R : \Pi \to \mathbb{R}$ is defined as: $R(\pi) \triangleq \mathbb{E}_{\mathbf{P}_0}[r(X, \pi(X), Z)]$.

Given the distributional shift between training and testing environments, we adopt a distributionally robust framework that explicitly accounts for potential deviations from the nominal training distribution $\mathbf{P}_0$. We define distributional uncertainty set $\mathcal{U}_{\mathbf{P}_0}(\delta)$ centered at $\mathbf{P}_0$ with radius $\delta$ as:

$$\mathcal{U}_{\mathbf{P}_0}(\delta) \triangleq \{\mathbf{P} : \mathbf{P} \ll \mathbf{P}_0 \text{ and } D(\mathbf{P}\|\mathbf{P}_0) \leq \delta\},$$

where $\mathbf{P} \ll \mathbf{P}_0$ denotes absolute continuity with respect to $\mathbf{P}_0$ and $D(\mathbf{P}\|\mathbf{P}_0) \triangleq \int \log(\frac{d\mathbf{P}}{d\mathbf{P}_0})d\mathbf{P}$ is the Kullback-Leibler (KL) divergence between two probability measures $\mathbf{P}$ and $\mathbf{P}_0$.

**Definition 2.6.** For a given $\delta > 0$, the distributionally robust value function $R_\delta : \Pi \to \mathbb{R}$ is defined as:

$$R_\delta(\pi) := \inf_{\mathbf{P} \in \mathcal{U}_{\mathbf{P}_0}(\delta)} \mathbb{E}_{\mathbf{P}}[r(X, \pi(X), Z)]$$

The optimal policy $\pi^*$ is a policy that maximizes the distributionally robust value function: $\pi^* \in \arg\max_{\pi \in \Pi} R_\delta(\pi)$. While such an optimal policy does not always exist, one can construct a sequence of policies converging to the supremum $\sup_{\pi \in \Pi}\{R_\delta(\pi)\}$; for simplicity, we assume the maximum is attainable throughout our analysis. We formalize the learning metric via the discrepancy between the performance of the optimal policy and the performance of the learned policy $\pi$, defined as the distributionally robust regret:

**Definition 2.7.** The distributionally robust regret $\text{Reg}(\pi)$ of a policy $\pi \in \Pi$ is defined as $\text{Reg}(\pi) \triangleq R_\delta(\pi^*) - R_\delta(\pi)$.

Recalling the strong duality results from Kallus et al. (2022) and Si et al. (2023), we transform the original infinite-dimensional optimization over $\mathbf{P} \in \mathcal{U}_{\mathbf{P}_0}(\delta)$ to a dual problem involving a single scalar $\alpha$. For any policy $\pi \in \Pi$, we

assume that the distributionally robust value $R_\delta(\pi) > 0$, which prevents degeneration into the trivial no-purchase outcome. It is natural in our setting and simply requires that the perturbation budget $\delta$ is not overly large, see Appendix A for a detailed discussion.

**Lemma 2.8.** *Suppose Assumption 2.1 and 2.3. The distributionally robust value $R_\delta(\pi)$ is equivalent to*

$$R_\delta(\pi) = \sup_{\alpha \geq 0} \varphi(\pi, \alpha) \triangleq -\alpha \log W(\pi, \alpha) - \alpha \delta, \quad (1)$$

*where $W(\pi, \alpha) \triangleq \mathbb{E}_{\mathbf{P}_0} [\exp(-r(X, \pi(X), Z)/\alpha)]$. Furthermore, $\varphi(\pi, \cdot)$ is strictly concave with respect to $\alpha$, and it admits a unique maximizer $\alpha^*(\pi) \in (0, \bar{\alpha}]$, where $\bar{\alpha} \triangleq 1/\delta$.*

Unlike the discrete setting, using the indicator function (Dudík et al., 2011; Si et al., 2023; Zhan et al., 2024) to estimate the robust policy value $R_\delta(\pi)$ is not valid in the continuous setting. Kallus & Zhou (2018) and Leung et al. (2025) propose a kernel smoothing approach, where the indicator function is replaced by a kernel function $K(\cdot)$ with bandwidth $h$. They also assume that the logging policy $\pi_0$ is known and propose to estimate $W(\pi, \alpha)$ by IPW, based on normalizing the propensity ratios $\frac{K\left(\frac{p_t - \pi(x_t)}{h}\right)}{h\pi_0(p_t|x_t)}$:

$$\hat{W}_T^{IPW}(\pi, \alpha) = \sum_{t=1}^{T} \frac{\frac{K\left(\frac{p_t - \pi(x_t)}{h}\right)}{h\pi_0(p_t|x_t)}}{\sum_{t=1}^{T} \frac{K\left(\frac{p_t - \pi(x_t)}{h}\right)}{h\pi_0(p_t|x_t)}} \exp(-p_t y_t/\alpha),$$
$$(2)$$

then $\quad \hat{\pi}^{IPW} \in \arg\max_{\pi \in \Pi} \hat{R}_\delta^{IPW}(\pi), \quad (3)$

where $\quad \hat{R}_\delta^{IPW}(\pi) \triangleq \sup_{\alpha \in (0, \bar{\alpha}]} \left\{ -\alpha \hat{W}_T^{IPW}(\pi, \alpha) - \alpha \delta \right\}.$

However, in practice, the logging policy $\pi_0$ is unknown and needs to be learned from the data. Furthermore, neither the standard IPW estimator nor its self-normalized variant achieves semiparametric efficiency. To overcome these challenges, we adopt doubly robust methods.

## 3. Distributionally Robust Policy Evaluation

In this section, we develop a doubly robust framework for evaluating the distributionally robust value of a target policy. However, it is challenging to estimate the distributionally robust value directly, as it involves a non-linear dual optimization problem and, potentially, a continuum of nuisance regressions. Building upon the localized debiased machine learning framework proposed by Kallus et al. (2022; 2024), we extend the localization technique from discrete to continuous treatment settings. Unlike the discrete action case, our estimator incorporates kernel smoothing to handle the continuous policy space.

### 3.1. Localized Doubly Robust Estimation

As established in Lemma 2.8, the distributionally robust value $R_\delta(\pi)$ can be characterized via its dual form, and there exists a unique optimal dual parameter $\alpha^* := \alpha^*(\pi)$ that satisfies the following first-order condition

$$-\log W_0(\pi, \alpha^*) - \frac{W_1(\pi, \alpha^*)}{\alpha^* W_0(\pi, \alpha^*)} - \delta = 0. \quad (4)$$

where $W_j(\pi, \alpha) \triangleq \mathbb{E}[(\pi(X)Y(\pi(X)))^j \exp(-\pi(X)$ $Y(\pi(X))/\alpha)]$, $j = 0, 1$. Using the shorthand $W_j^* = W_j(\pi, \alpha^*)$, the robust policy value is then given by

$$R_\delta(\pi) = -\alpha^* \log W_0^* - \alpha^* \delta, \quad (5)$$

Therefore, estimating $\alpha^*(\pi)$ and $R_\delta(\pi)$ in (4) and (5) is equivalent to estimating the root of the following moment equation with parameter $\theta = [\alpha, W_0, W_1, R_\delta]^\top$:

$$\mathbb{E}[U(\pi(X)Y(\pi(X)); \alpha) + G(\theta)] = \mathbf{0}, \quad (6)$$

$$U(r; \alpha) = \begin{bmatrix} \exp(-r/\alpha) \\ r\exp(-r/\alpha) \\ 0 \\ 0 \end{bmatrix}, G(\theta) = \begin{bmatrix} -W_0 \\ -W_1 \\ -\delta - \log W_0 - \frac{W_1}{\alpha W_0} \\ -R_\delta - \alpha \log W_0 - \alpha\delta \end{bmatrix}.$$

In practice, direct estimation of (4) and (5) is infeasible because the counterfactual outcomes $Y(\pi(X))$ are not observed. Instead, we consider the following doubly robust moment equation in terms of the observed variables $V = (X, P, Y)$, with nuisances $\lambda_1, \lambda_2$ to be estimated:

$$\mathbb{E}[\psi(V; \theta, \lambda_1^*(V; \alpha), \lambda_2^*(V))] = \mathbf{0},$$
$$\psi(v; \theta, \lambda_1, \lambda_2) = \frac{K(\frac{p - \pi(x)}{h})}{h\lambda_2(x, p)} (U(py; \alpha) - \lambda_1(x, p; \alpha))$$
$$+ \lambda_1(x, \pi(x); \alpha) + G(\theta),$$

where we incorporate kernel function $K(\cdot)$ with bandwidth $h$ to handle the continuous policy space. And $\lambda_2^*(x, p) = \pi_0(p|x)$ is the logging policy,

$$\lambda_1^*(x, p; \alpha) = \mathbb{E}[U(PY; \alpha)|X = x, P = p]$$
$$= [m_0(x, p; \alpha), m_1(x, p; \alpha), 0, 0]^\top,$$
$$m_j(x, p; \alpha) = \mathbb{E}[(PY)^j \exp(-PY/\alpha)|X = x, P = p].$$

A significant technical hurdle in distributionally robust policy evaluation is that the outcome regressions, $m_j(\cdot; \alpha)$, are coupled with the dual parameter $\alpha$. Recalling that Neyman orthogonality for continuous treatment (Colangelo & Lee, 2026) implies that the Gâteaux derivatives of $\mathbb{E}[\psi(V; \theta, \lambda_1(V; \alpha), \lambda_2(V))]$ with respect to the nuisances $(\lambda_1, \lambda_2)$ are $\mathcal{O}(h^s)$ when evaluated at $\theta^* = (\alpha^*, W_0^*, W_1^*, R_\delta(\pi))$, $\lambda_1(\cdot; \alpha) = \lambda_1^*(\cdot; \alpha^*)$ and $\lambda_2 = \lambda_2^*$. This orthogonality ensures that our estimator remains first-order insensitive to the estimation errors of the nuisances as

the bandwidth $h \to 0$. Thus, we leverage the localization technique (Kallus et al., 2022; 2024) to utilize an initial guess $\hat{\alpha}_{init}$ and fit the nuisances only at this localized point, instead of a continuum of regressions for whole $\alpha \in (0, \bar{\alpha}]$, as provided the initial guess is sufficiently close to $\alpha^*$.

---

**Algorithm 1** Localized Doubly Robust DRO OPE for Continuous Pricing (LDR$^2$O$^2$PE-CP)

---

1: **Input:** Dataset $\mathcal{D}$, uncertainty radius $\delta$, policy $\pi$, kernel function $K$ with bandwidth $h$.
2: Randomly evenly partition $\mathcal{D}$ into $\{\mathcal{I}_1, \ldots, \mathcal{I}_L\}$.
3: **for** $\ell = 1$ to $L$ **do**
4:     Train $\hat{\pi}_0^{(\ell)}(p|x)$ to fit $\pi_0(p|x)$ using $\mathcal{I}_\ell^c = \mathcal{D} \setminus \mathcal{I}_\ell$.
5:     Randomly split $\mathcal{I}_\ell^c$ into two halves: $\mathcal{J}_1, \mathcal{J}_2$.
6:     $\alpha_{init}^{(\ell)} \leftarrow$ InitialEstimate$(\mathcal{J}_1, \delta, \pi)$.
7:     Train $\hat{m}_j^{(\ell)}(x, p; \hat{\alpha}_{init}^{(\ell)})$ to fit $m_j(x, p; \hat{\alpha}_{init}^{(\ell)})$ using data $\mathcal{J}_2$, $j = 0, 1$.
8: **end for**
9: Initialize $\alpha \leftarrow \frac{1}{L} \sum_{\ell=1}^L \alpha_{init}^{(\ell)}$.
10: Find $\hat{\alpha}$ that solves the estimated moment equation:

$$-\log \hat{W}_0(\alpha) - \frac{\hat{W}_1(\alpha)}{\alpha \hat{W}_0(\alpha)} - \delta = 0, \text{ where} \quad (7)$$

$$\hat{W}_j(\alpha) \triangleq \frac{1}{T} \sum_{\ell=1}^L \sum_{t \in \mathcal{I}_\ell} \hat{W}_j^{(t,\ell)}(\alpha), \quad j = 0, 1,$$

$$\hat{W}_j^{(t,\ell)}(\alpha) \triangleq \hat{m}_j^\ell(x_t, \pi(x_t); \hat{\alpha}_{init}^{(\ell)}) + \frac{K(\frac{p_t - \pi(x_t)}{h})}{h \hat{\pi}_0^{(\ell)}(p_t|x_t)}$$

$$\cdot \left( (p_t y_t)^j \exp(-p_t y_t / \alpha) - \hat{m}_j^{(\ell)}(x_t, p_t; \hat{\alpha}_{init}^{(\ell)}) \right).$$

11: Calculate $\hat{R}_\delta \leftarrow -\hat{\alpha} \log \hat{W}_0(\hat{\alpha}) - \hat{\alpha}\delta$.
12: **Return:** $\hat{\theta} = [\hat{\alpha}, \hat{W}_0(\hat{\alpha}), \hat{W}_1(\hat{\alpha}), \hat{R}_\delta]^\top$.

---

We propose Algorithm 1, Localized Doubly Robust DRO OPE for Continuous Pricing (LDR$^2$O$^2$PE-CP), to summarize the localized doubly robust evaluation procedure for continuous treatment. Following the two-level cross-fitting framework from Kallus et al. (2022), our implementation is tailored to handle the continuous action space. First, we randomly evenly partition $\mathcal{D}$ into $L$ disjoint folds $\{\mathcal{I}_\ell\}_{\ell \in [L]}$. We estimate the propensity estimator $\hat{\pi}_0^{(\ell)}$ using the out-of-fold data $\mathcal{I}_\ell^c = \mathcal{D} \setminus \mathcal{I}_\ell$. And we address the dependence of outcome regressions on $\alpha$ by fitting $\hat{m}_j^{(\ell)}$ only at a localized point $\hat{\alpha}_{init}^{(\ell)}$. Here, $\hat{\alpha}_{init}^{(\ell)}$ is a preliminary consistent estimate obtained by InitialEstimate, for example, via a continuous-treatment IPW estimator (Leung et al., 2025). By leveraging the localization technique, we circumvent these continuum regression problems, requiring the training of only three machine learning models. Finally, we solve the empirical moment equation for the dual parameter $\alpha$ by the Newton-Raphson method (Si et al., 2023; Leung et al., 2025) to find

the root of (7). The gradients are provided in Lemma C.1.

## 3.2. Asymptotic Analysis

Set $h = \mathcal{O}(T^{-\gamma_h})$. Define the estimation rates as follows,

$$\max_{j=0,1} \left\| \hat{m}_j^{(\ell)}(\cdot; \hat{\alpha}_{init}^{(\ell)}) - m_j(\cdot; \hat{\alpha}_{init}^{(\ell)}) \right\|_\infty \le o_p(T^{-\rho_m}),$$

$$\left\| \hat{\pi}_0^{(\ell)} - \pi_0 \right\|_\infty \le o_p(T^{-\rho_\pi}), \ |\hat{\alpha}_{init} - \alpha^*| \le o_p(T^{-\rho_\alpha}).$$

**Assumption 3.1.** We assume that $\rho_\pi + \rho_\alpha \wedge \rho_m \ge \frac{1 - \gamma_h}{2}$.

**Assumption 3.2.** Kernel function $K(u) : \mathbb{R} \to [0, \infty)$ has Lipschitz constant $L_K$ and satisfies:
(i) $\int K(u)\mathrm{d}u = 1$, $\int u K(u)\mathrm{d}u = 0$,
(ii) $\int |u|^k |K(u)|du < \infty$ and $\int |u|^k K^2(u)du < \infty$ for $k = 0, 1, 2$.

**Theorem 3.3.** *Let* $\theta^* = [\alpha^*, W_0^*, W_1^*, R_\delta(\pi)]^\top$ *be the solution of* (6). *Let* $\hat{\theta}$ *be the estimator defined in Algorithm 1. Then, under Assumptions 2.1-2.3, 3.1, 3.2, and* $\sqrt{Th^{1+2s}} \to 0$ *when Assumption 2.4 holds for s-order differentiability,* $s = 1, 2$, *we have*

$$\sqrt{Th}(\hat{\theta} - \theta^*) \xrightarrow{d} \mathcal{N}(\mathbf{0}, \Sigma^*), \ \Sigma^* = J^{*-1}\Omega^* J^{*-\top}, where$$

$$J^* = \begin{bmatrix} W_1^*/\alpha^{*2} & -1 & 0 & 0 \\ W_2^*/\alpha^{*2} & 0 & -1 & 0 \\ \frac{W_1^*}{\alpha^{*2} W_0^*} & -\frac{1}{W_0^*} + \frac{W_1^*}{\alpha^* W_0^{*2}} & -\frac{1}{\alpha^* W_0^*} & 0 \\ -\log W_0^* - \delta & -\frac{\alpha^*}{W_0^*} & 0 & -1 \end{bmatrix},$$

$$\Omega^* = \int K^2(u)du \cdot \mathrm{diag}(\omega_0, \omega_1, 0, 0),$$

$$\omega_j = \mathbb{E}\left[ \frac{\mathrm{Var}((PY)^j \exp(-PY/\alpha^*)|X, P = \pi(X))}{\pi_0(\pi(X)|X)} \right],$$

*and* $\Sigma^*$ *is the optimal covariance, then* $\hat{\theta}$ *achieves the semiparametric efficiency lower bound for* $\theta^*$.

Theorem 3.3 holds under some additional regularity conditions on the nuisance smoothness, which are formally stated in Appendix B. Algorithm 1 achieves $\sqrt{Th}$-consistency, aligned with the $\sqrt{Th}$-consistency under non-robust setting (Colangelo & Lee, 2026). Furthermore, our LDR$^2$O$^2$PE-CP estimator attains semiparametric efficiency, which enables a flexible trade-off in the convergence rates of the nuisance functions to satisfy the product rate condition in Assumption 3.1. Notably, this requirement is relaxed to $(1 - \gamma_h)/2$, which is strictly weaker than the $1/2$ threshold required for $\sqrt{T}$-consistent semiparametric estimators in discrete action spaces (Kallus et al., 2022). Moreover, it yields a more stable covariance structure $\Omega^*$ that depends on the conditional variance rather than the second moment as in the self-normalized IPW estimator of (2) (Leung et al., 2025), even when the logging policy is known.

Finally, we note that the asymptotic normality in Theorem 3.3 requires the bandwidth $h$ to satisfy an undersmoothing

condition relative to the smoothness of the noise distribution in Assumption 2.4. Specifically, the approximation bias introduced by kernel-smoothing is $\mathcal{O}_p(h^s)$ for $s = 1, 2$. To ensure this bias vanishes in the limiting distribution, one requires $\sqrt{T}h^{1+2s} \to 0$, leading to a regret rate slightly slower than $\tilde{\mathcal{O}}_p(T^{-s/(2s+1)})$ (see Theorem 4.4). This undersmoothing strategy is standard in the double machine learning literature for continuous treatment (Colangelo & Lee, 2026) as it enables the construction of valid frequentist confidence intervals without explicit bias correction.

# 4. Distributionally Robust Policy Learning

In this section, we now focus on learning a robust policy $\hat{\pi}$ that maximizes the empirical distributionally robust value defined in Definition 4.1, $\hat{\pi} \in \arg\max_{\pi \in \Pi} \hat{R}_\delta(\pi)$, and analyze how the empirically optimal policy performs out of sample by a finite-sample theoretical guarantee.

**Definition 4.1** (Doubly Robust Estimators). Let $K(\cdot)$ be a kernel function with bandwidth $h > 0$, and $\{(x_t, p_t, y_t)\}_{t=1}^T$ be a given dataset. Define the DR estimator for a continuous pricing problem as

$$\hat{W}_T(\pi, \alpha) = \frac{1}{L} \sum_{\ell=1}^L \hat{W}^{(\ell)}(\pi, \alpha), \qquad (8)$$

$$\hat{W}^{(\ell)}(\pi, \alpha) = \frac{1}{|\mathcal{I}_\ell|} \sum_{t \in \mathcal{I}_\ell} \hat{m}_0^{(\ell)}(x_t, \pi(x_t); \alpha) + \frac{1}{|\mathcal{I}_\ell|\hat{S}_\ell^\pi}$$

$$\cdot \sum_{t \in \mathcal{I}_\ell} \frac{K\left(\frac{p_t - \pi(x_t)}{h}\right)}{h\hat{\pi}_0^{(\ell)}(p_t|x_t)} \left(\exp(-p_t y_t/\alpha) - \hat{m}_0^{(\ell)}(x_t, p_t; \alpha)\right),$$

where $\hat{S}_\ell^\pi = \frac{1}{|\mathcal{I}_\ell|} \sum_{t \in \mathcal{I}_\ell} \frac{K\left(\frac{p_t - \pi(x_t)}{h}\right)}{h\hat{\pi}_0^{(\ell)}(p_t|x_t)}$.

Then, we define the distributionally robust value estimator $\hat{R}_\delta : \Pi \to \mathbb{R}$ as

$$\hat{R}_\delta(\pi) \triangleq \sup_{\alpha \geq 0}\{\hat{\varphi}_T(\pi, \alpha)\},$$

where $\hat{\varphi}_T(\pi, \alpha) \triangleq -\alpha \log \hat{W}_T(\pi, \alpha) - \alpha\delta$.

Different from distributionally robust policy evaluation at an initial guess $\alpha_{init}^{(\ell)}(\pi)$, distributionally robust policy learning needs to compute the distributionally robust estimator defined in Definition 4.1, and find the optimal $\hat{\alpha}^*(\pi)$ for maximizing $\hat{\varphi}_T(\pi, \alpha)$. Thus, policy learning inevitably requires $\hat{m}_0^{(\ell)}(x, p; \alpha)$ as a continuum estimate of the outcome regression functions $\{m_0(\cdot, \cdot; \alpha) : \mathcal{X} \times \mathcal{P} \to \mathbb{R} \mid \alpha \in (0, \bar{\alpha}]\}$.

## 4.1. Learning Algorithm

A significant challenge arises because $m_0$ depends on the dual variable $\alpha$. In the policy learning phase, $\alpha$ varies contin-

uously, and retraining a regression model for every possible value of $\alpha$ would be computationally prohibitive. Following the approach proposed by Kallus et al. (2022), we utilize a local weighting approach to estimate $m_0(x, p; \alpha)$. Specifically, given a dataset, we first learn a set of data-driven weight functions $\{\hat{\omega}_t^{(\ell)}(x, p)\}_{t \in \mathcal{I}_\ell^c}$. These weights represent the proximity or similarity of the observed covariate-action pair $(x_i, p_i)$ to a query point $(x, p)$. Common methods for constructing these weights include $k$-nearest neighbors, kernel regressions, and generalized random forests (Athey et al., 2019; Oprescu et al., 2019; Bertsimas & Kallus, 2020; Cevid et al., 2022; Khosravi et al., 2022). With these weights, the estimator for the regression continuum for any $\alpha > 0$ is

$$\hat{m}_0^{(\ell)}(x, p; \alpha) = \sum_{t \in \mathcal{I}_\ell^c} \hat{\omega}_t^{(\ell)}(x, p) \exp(-p_t y_t/\alpha). \quad (9)$$

In our experiments, we constructed the weights using generalized random forests (Athey et al., 2019). Specifically, we first train a regression forest targeting the reward $r$. For a given query point $(x, p)$, the forest-induced weights $\hat{\omega}_t(x, p)$ are computed by averaging the normalized leaf memberships across all $B$ trees,

$$\hat{\omega}_t^{(\ell)}(x, p) = \frac{1}{B} \sum_{b=1}^B \frac{\mathbf{1}(x_t, p_t) \in L_b(x, p)}{|L_b(x, p)|}, \quad (10)$$

where $L_b(x, p)$ denotes the set of training examples falling into the same leaf as the query point in the $b$-th tree. This method has been successfully applied in statistical estimation and decision making (Bertsimas & Kallus, 2020; Kallus et al., 2022; Kallus & Mao, 2023).

Since joint optimization over $(\pi, \alpha)$ is generally non-convex and computationally intractable, we adopt an alternating fashion (Kallus et al., 2022; Si et al., 2023). While this method guarantees convergence only to a local maximum, it is computationally efficient and empirically effective. The complete procedure, termed Continuum Doubly Robust DRO OPL for Continuous Pricing (CDR²O²PL-CP), is detailed in Algorithm 2.

## 4.2. Statistical Performance Guarantee

We establish the finite-sample statistical performance guarantee for the distributionally robust optimal policy $\hat{\pi}$. Under Assumption 3.2, the bias introduced by kernel density estimation is $\mathcal{O}(h^s)$ when Assumption 2.4 holds for $s$-order bounded derivative (see Lemma C.2), and depends on convergence rate of $\hat{m}_0^{(\ell)}(x, p; \alpha)$ (see Lemma C.5).

**Assumption 4.2.** Suppose $\left\{\hat{m}_0^{(\ell)}(\cdot; \alpha) : \alpha \in (0, \bar{\alpha}]\right\}_{\ell=1}^L$ and $\{\hat{\pi}_0^{(\ell)}(\cdot)\}_{\ell=1}^L$ are learned from a dataset of $\frac{T(L-1)}{L}$ points, satisfying the following conditions:

(i). There exists $\gamma_\pi, \gamma_m > 0$ such that

$$\mathbb{E}\left[\left\|\hat{\pi}_0^{(\ell)} - \pi_0\right\|_\infty^2\right] = o(T^{-2\gamma_\pi}), \text{ and}$$

$$\mathbb{E}\left[\left(\sup_{\alpha \in (0,\bar{\alpha}]}\left\|\hat{m}_0^{(\ell)}(\cdot,\cdot;\alpha) - m_0(\cdot,\cdot;\alpha)\right\|_\infty\right)^2\right] = o(T^{-2\gamma_m}).$$

(ii). $\hat{\pi}_0^{(\ell)} \in [\eta, 1]$, $\hat{m}_0^{(\ell)}(\cdot;\alpha) \in [0, 1]$, $\forall \ell \in [L]$, $\alpha \in (0, \bar{\alpha}]$.

(iii). $\{\hat{m}_0^{(\ell)}\}_{\ell=1}^L$ are $L_{\hat{m}}$-Lipschitz with respect to $(x, p) \in \mathcal{X} \times \mathcal{P}$ and monotonically non-decreasing in $\alpha$.

---

**Algorithm 2** Continuum Doubly Robust DRO OPL for Continuous Pricing (CDR$^2$O$^2$PL-CP)

1: **Input:** Dataset $\mathcal{D}$, logging policy $\pi_0$, uncertainty radius $\delta$, kernel function $K$ with bandwidth $h$, policy class $\Pi$.
2: **Output:** Distributionally robust optimal policy $\hat{\pi}$.
   *Phase 1: Continuum Regression*
3: Randomly evenly partition $\mathcal{D}$ into $L$ disjoint folds $\{\mathcal{I}_1, \ldots, \mathcal{I}_L\}$.
4: **for** $\ell = 1$ to $L$ **do**
5:     Let $\mathcal{I}_\ell^c = \mathcal{D} \setminus \mathcal{I}_\ell$ be the training data.
6:     Train $\hat{\pi}_0^{(\ell)}(p|x)$ to fit $\pi_0(p|x)$ using $\mathcal{I}_\ell^c$.
7:     Train $\hat{m}_0^{(\ell)}(x, p; \alpha)$ to fit $m_0(x, p; \alpha)$ for all $\alpha \in (0, \bar{\alpha}]$ using (9)–(10) with $\mathcal{I}_\ell^c$.
8: **end for**
   *Phase 2: Alternating Optimization*
9: Initialize $\hat{\pi}$.
10: **repeat**
11:     Compute $\hat{W}_T(\hat{\pi}, \alpha)$ using (8).
12:     Update $\hat{\alpha} \leftarrow \arg\max_{\alpha \geq 0} -\alpha \log \hat{W}_T(\hat{\pi}, \alpha) - \alpha\delta$.
13:     Update $\hat{\pi} \leftarrow \arg\min_{\pi \in \Pi} \hat{W}_T(\pi, \hat{\alpha})$.
14: **until** $\hat{\pi}$ converges.
15: **Return:** $\hat{\pi}$.

---

Assumption 4.2(i) imposes a uniform convergence rate ($L_\infty$ norm) for the nuisance estimators. Kallus et al. (2022) and Zhou et al. (2023) assume $L_2$ convergence rate for discrete treatment. In our continuous treatment setting, however, the $L_\infty$ requirement is a necessary theoretical cost to ensure that estimation remains robust across the entire continuous treatment manifold, which is consistent with Ai et al. (2026). We can construct a continuum estimator of $m_0$ by applying the GRF framework to (9)–(10). According to Theorem 5 in Athey et al. (2019), this yields the desired convergence rate of $o_p(T^{-2\gamma_m})$ under their regularity conditions, by setting the subsample size $s$ used to grow each tree to be $s = T^\beta$ with $\beta < 1 - 2\gamma_m$. To estimate the conditional probability $\pi_0$, one can use the methods in Colangelo & Lee (2026) and Cattaneo et al. (2024). Assumption 4.2 (ii) and (iii) share the same properties as true functions $\pi_0$ and $m_0$. Specifically,

we can employ a smoothed version of the forest (e.g., using a smooth kernel for leaf membership) to ensure it. We also discuss a corresponding second-order bounded derivative assumption when $s = 2$ in Assumption 2.4.

Before giving the theorem, we measure the complexity of the policy class $\Pi$ using pseudo-dimension. Pseudo dimension generalizes the classical VC-dimension from binary-valued functions to real-valued functions. Assumption 4.3 is mild and is satisfied by a vast range of parametric policy classes used in practice, such as linear policies, deep neural networks (Bartlett et al., 2019). This assumption is used to derive the Rademacher complexity bound, see Lemma C.6.

**Assumption 4.3** (Policy Class Complexity). The policy class $\Pi$ has a finite pseudo-dimension, denoted as Pdim($\Pi$).

**Theorem 4.4.** *Suppose Assumption 2.1-2.4, 3.2, 4.2, 4.3 hold, Given $\epsilon > 0$, $h > 0$, and a finite pseudo-dimension policy class $d_\Pi := Pdim(\Pi)$, we have with probability at least $1 - 7\epsilon$,*

$$\text{Reg}(\hat{\pi}) \lesssim \bar{\alpha}\Big(\log\Big(\frac{1}{h}\Big) \cdot \Big(\frac{d_\Pi}{Th/L} + \sqrt{\frac{d_\Pi}{Th/L}}$$

$$+ \sqrt{\frac{\log(1/\epsilon)}{Th/L}} + \frac{\log(1/\epsilon)}{Th/L}\Big) + T^{-\gamma_\pi - \gamma_m} + h^s\Big).$$

As shown in Theorem 4.4, with high probability, the distributionally robust regret of the learned policy is upper bounded by $\tilde{\mathcal{O}}_p(T^{-s/(2s+1)})$ with the optimal bandwidth $h = \Theta(T^{-1/(2s+1)})$, where $\tilde{\mathcal{O}}$ hides logarithmic factors in $T$. This rate only requires a not strong condition $\gamma_\pi + \gamma_m \geq s/(2s+1)$ in Assumption 4.2. A very recent study by Leung et al. (2025) also investigates the distributionally robust policy learning problem in continuous action space. They employ the IPW estimator (2) under $s = 2$ assumption, and the suboptimal regret bound they derived is $\mathcal{O}_p(T^{-1/4})$ with their optimal bandwidth $h = \Theta(T^{-1/8})$. While our DR estimator can naturally degenerate to their IPW estimator by setting the outcome regression $\hat{m}_0$ to zero. Hence, we establish a sharper regret bound not only for our DR but also for their IPW. This also aligns with rates achieved in non-robust continuous treatment settings (Ai et al., 2026).

## 5. Experiments

We evaluate our proposed OPE/L methods in both simulated and real-world contextual pricing environments.

### 5.1. Simulation Experiment

The simulated setup is as follows. Context vectors $X \in \mathbb{R}^3$ are sampled from $\mathcal{N}(\mathbf{0}, \mathbf{I}_3)$. Historical prices are generated according to a linear logging policy $\pi_0$, where $P \mid X \sim \mathcal{N}(\mu_0(X), \sigma_0^2)$ with $\sigma_0 = 1.0$, $\mu_0(X) =$

$\beta_0 + \beta_1^\top X$, $\beta_0 = 1.0$, $\beta_1 = [0.8, 1.0, 0.0]^\top$, clipped to $\mathcal{P} = [0.5, 3.0]$. The true customer valuation is $f(X) = \theta_0 + \theta_1^\top X + \theta_2 X_1 X_2$, $\theta_0 = 1.5$, $\theta_1 = [1.2, 1.8]^\top$, $\theta_2 = 0.3$, where $(X_1, X_2)$ are informative features and $X_3$ acts as a nuisance feature. The binary purchase outcomes are $Y = \mathbf{1}\{f(X) + Z \geq P\}$, where $Z \sim \text{Logistic}(0, 0.2)$ follows an infinitely differentiable logistic distribution with bounded derivatives. The reward is $R = P \cdot Y$.

First, we compare our proposed LDR$^2$O$^2$PE-CP (Algorithm 1) against the DRO-IPW baseline (2) from Leung et al. (2025). The target policy we seek to evaluate is a linear policy $\pi_{\text{target}}(X) = w_1^\top X + w_0$ with $w_{\text{target}} = (w_0, w_1)$, where $w_0$ captures the intercept. Policies are trained with uncertainty radius $\delta_{\text{train}} = 0.2$ and evaluated under test radii $\delta \in \{0.1, 0.2, 0.3\}$, and $T$ is the training sample size.. We use a Gaussian kernel $K(u) = \frac{1}{\sqrt{2\pi}} \exp(-u^2/2)$ with different bandwidth. We compare different bandwidths, (i) the OPE-optimal bandwidth $h = \Theta(T^{-1/5.0})$ under $s = 2$ in Kallus & Zhou (2018); Leung et al. (2025); (ii) our undersmoothing bandwidth $h = \Theta(T^{-1/(2s+1-0.1)})$ under $s = 1, 2$. Furthermore, we apply these choices with a Silverman-type scaling constant, i.e., $h = 1.06 \cdot \text{std}_p \cdot T^{-1/5.0}$ and $h = 1.06 \cdot \text{std}_p \cdot T^{-1/(2s+1-0.1)}$, where $\text{std}_p$ is the sample standard deviation of observed prices. Both methods use Ridge regression to estimate the logging policy by fitting $\mu_0(X)$ and using the residual standard deviation as $\sigma_0$. Outcome regressions are estimated by forest-induced weights (10) (via the `econml` package, 20 trees, max depth 10) to ensure consistency with the OPL setup. All models are fitted with $L = 5$ fold cross-fitting, and averaged over 20 random seeds. Shaded regions show 90% confidence intervals.

Figure 1 shows the mean squared error (MSE) of the estimated distributionally robust policy value $R_\delta$. We observe that LDR$^2$O$^2$PE-CP outperforms the DRO-IPW baseline, as long as the sample size $T$ is sufficiently large. However, in the non-asymptotic regime ($T < 1000$), LDR$^2$O$^2$PE-CP may exhibit less stability compared to DRO-IPW. Nevertheless, LDR$^2$O$^2$PE-CP exhibits a faster convergence rate of MSE decrease and offers a significant improvement over the baseline. Overall, LDR$^2$O$^2$PE-CP provides the efficacy of the localized technique even under varying adversarial environments and different bandwidth.

Next, we compare our proposed CDR$^2$O$^2$PL-CP (Algorithm 2) against the non-robust DR method from Ai et al. (2026) and the DRO-IPW learning method from Leung et al. (2025). The target policy class is $\{\pi(X) = w_1^\top X + w_0 : w = (w_0, w_1), \|w\|_\infty \leq 2\}$. The continuum of outcome regressions $\{\hat{m}_0(x, p; \alpha); \alpha\}$ is estimated according to (9)–(10) using the same `econml` setup as in OPE. Optimization uses Adam (learning rate $= 3 \times 10^{-3}$) in Line 13 and bounded optimization for $\alpha$ in Line 12.

Figure 2 reports the distributionally robust policy value $R_\delta$

of policies learned in the distributionally robust OPL task. For fair comparison, all methods, including the non-robust DR baseline from Ai et al. (2026), are evaluated using the worst-case robust metric $R_\delta(\pi)$, regardless of whether robustness was optimized during training. The non-robust DR baseline exhibits substantially wider confidence intervals, indicating high sensitivity to random seeds and brittleness under distributional shifts. In contrast, both robust methods produce much more stable estimates. Moreover, our CDR$^2$O$^2$PL-CP not only achieves the highest robust values but also yields slightly tighter confidence intervals than DRO-IPW for large sample sizes. This reflects the statistical efficiency gain of doubly robust estimation over IPW-based methods, even when propensity scores are estimated reliably (e.g., via Ridge regression). Overall, CDR$^2$O$^2$PL-CP delivers more stable learning and higher worst-case rewards across different levels of distributional robustness.

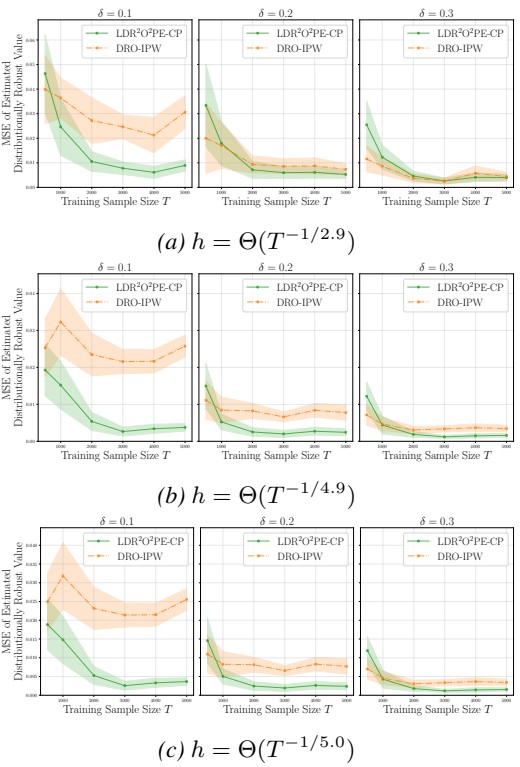

*Figure 1.* Comparison of LDR$^2$O$^2$PE-CP against DRO-IPW in the distributionally robust OPE task. Results are shown for different bandwidth choices and test environments $\delta \in \{0.1, 0.2, 0.3\}$ with $\delta_{\text{train}} = 0.2$ and 2500 test samples. The x-axis is the training sample size $T$, and the y-axis is the mean squared error (MSE) of the estimated distributionally robust value $R_\delta$.

## 5.2. Empirical Experiment - The Expedia Study

We study offline contextual pricing on the dataset of the Expedia Personalized Sort Competition (Adam et al., 2013) hosted by Kaggle in conjunction with Expedia.. Each record

is a search impression for a hotel listing, containing visitor and search context, hotel attributes, competitor prices, the logged list price, and whether the listing was booked. Following the offline pricing formulation in Section 2, we map each impression to a triplet $(x_t, p_t, y_t)$: $x_t \in \mathbb{R}^{45}$ collects numeric covariates (visitor history, search features, hotel characteristics, and competitor rates); $p_t$ is the logged price in USD; and $y_t$ is the booking indicator. To create distribution shifts, we split the training and test data based on hotel star rating prop_starrating $\in \{1, 2, 3, 4, 5\}$. We train using only prop_starrating $\in \{1, 2, 3\}$ and test only prop_starrating $\in \{4, 5\}$.

We learn the non-robust linear policy $\hat{\pi}_{\text{Lin}}$ (Ai et al., 2026), distributionally robust linear policy $\hat{\pi}_{\text{DRO}-\text{IPW}}$ (Leung et al., 2025) and our proposed $\hat{\pi}_{\text{CDR}^2\text{O}^2\text{PL}-\text{CP}}$ in the linear policy class $\Pi_{\text{lin}}$. Each policy $\pi_w(x) \in \Pi_{\text{lin}}$ is parameterized by a vector $w = (w_0, w_1) \in \mathbb{R}^{46}$, and the mapping $\pi : \mathcal{X} \to [p_{\min}, p_{\max}]$ is defined as

$$\Pi_{\text{lin}} \triangleq \left\{ \pi_w(x) = p_{\min} + (p_{\max} - p_{\min}) \, \sigma(w_1^\top x + w_0) \right\},$$

where $\sigma(\cdot)$ is the logistic link. We fix the uncertainty radius $\delta_{\text{train}} = 0.1$ used in the training procedure and the training size $T = 500$. To characterize adversarial performance on Expedia, we adopt an empirical KL-adversarial stress test following Si et al. (2023). We fix a deployment test pool of $T' = 2,500$ triplets from the training dataset and generate $M = 10$ replicates. For each $j \in [M]$, we draw a bootstrap batch $\{(x_t^{(j)}, p_t^{(j)}, y_t^{(j)})\}_{t=1}^{T'}$ with replacement from this pool, compute $\alpha_j^*(\pi)$ at test radius $\delta = 0.2$ by Lemma 2.8, and form weights $w_t^{(j)} \propto \frac{K((p_t^{(j)} - \pi(x_t^{(j)})/h))}{h \hat{\pi}_0(p_t^{(j)} | x_t^{(j)})} \exp(-p_t^{(j)} y_t^{(j)} / \alpha_j^*(\pi))$. We then obtain a new dataset $\{\tilde{x}_t^{(j)}, \tilde{p}_t^{(j)}, \tilde{y}_t^{(j)}\}_{t=1}^{T'}$ by resampling $T'$ triplets according to $\{w_t^{(j)}\}$ and evaluate

$$\hat{R}_{\text{adv}}(\pi) = \min_{1 \le j \le M} \frac{1}{T'} \sum_{t=1}^{T'} \hat{r}\big(\tilde{x}_t^{(j)}, \pi(\tilde{x}_t^{(j)})\big),$$

where $\hat{r}(x, p)$ is a linear outcome regression for $\mathbb{E}[r | X = x, P = p]$, fitted on the training set. Table 2 reports the mean and standard error over 20 random repeats. We find that our learned linear policy outperforms $\hat{\pi}_{\text{Lin}}$ and $\hat{\pi}_{\text{DRO}-\text{IPW}}$.

## 6. Conclusion

In this paper, we present a doubly robust framework for distributionally robust OPE/L in continuous action settings. Our framework protects against distributional shifts across environments. For evaluation, LDR$^2$O$^2$PE-CP uses localization to avoid the intractability of optimizing over a continuum of outcome functions. It fits only three regressions, as tractable as standard non-robust DR, while achieving semiparametric efficiency. For learning, CDR$^2$O$^2$PL-CP

fits a continuum of regressions with data-driven weights, smoothly adapting to continuous pricing. By leveraging the latent smoothness of demand noise, we overcome the discontinuity in purchase outcomes and derive a finite-sample regret bound of $\tilde{\mathcal{O}}_p(T^{-s/(2s+1)})$ for $s = 1, 2$ under mild product rate assumptions. This bound represents an improvement over the $\mathcal{O}_p(T^{-1/4})$ rate in prior work on DRO methods and aligns with rates in non-robust approaches.

Comprehensive simulations demonstrate that our methods exhibit strong robustness to distributional shifts. Although we adopt the KL-divergence, future research could explore other measures, such as Wasserstein distance, to offer better geometric protections in specific scenarios. Additionally, developing more scalable optimization routines to handle high-dimensional state spaces would further broaden the practical utility of our framework in large-scale applications.

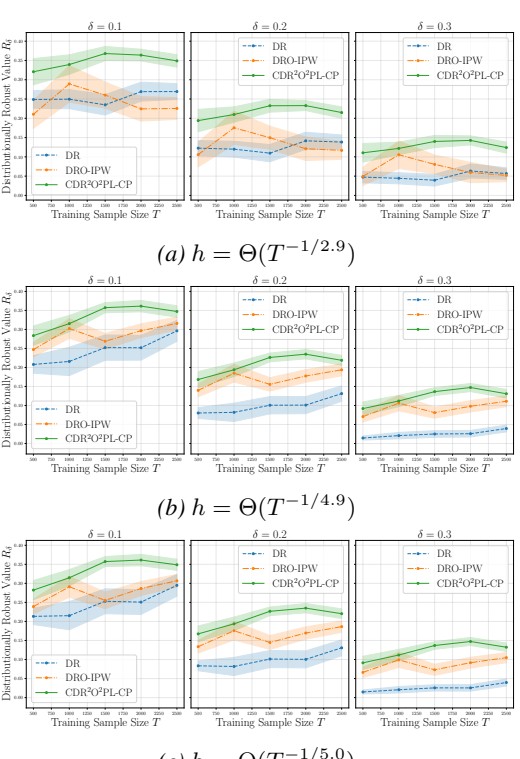

*(a)* $h = \Theta(T^{-1/2.9})$

*(b)* $h = \Theta(T^{-1/4.9})$

*(c)* $h = \Theta(T^{-1/5.0})$

*Figure 2.* Comparison of CDR$^2$O$^2$PL-CP against the non-robust DR and DRO-IPW in the distributionally robust OPL task. Results are shown for different bandwidth choices and test environments $\delta \in \{0.1, 0.2, 0.3\}$ with $\delta_{\text{train}} = 0.2$ and 2500 test samples. The x-axis is the training sample size $T$, and the y-axis is the distributionally robust policy value $R_\delta$.

*Table 2.* Comparison of robust performance in Expedia study.

| Method | $\hat{R}_{\text{adv}}$ |
|---|---|
| $\hat{\pi}_{\text{Lin}}$ | $6.201 \pm 0.683$ |
| $\hat{\pi}_{\text{DRO}-\text{IPW}}$ | $6.254 \pm 0.873$ |
| $\hat{\pi}_{\text{CDR}^2\text{O}^2\text{PL}-\text{CP}}$ | $\mathbf{6.350 \pm 0.598}$ |

## Acknowledgements

This work was supported in part by the National Natural Science Foundation of China [Grants 72394363/72394360, and 72571132], and the Postgraduate Research & Practice Innovation Program of Jiangsu Province [Grant KYCX25_0112].

## Impact Statement

Our research introduces a doubly robust framework for distributionally robust policy evaluation/learning in continuous action spaces. This work primarily aims to contribute to the methodological progress in machine learning and causal inference. It enhances the reliability of data-driven decision-making under environmental shifts, which is crucial for building safer and more dependable systems in domains like resource allocation and economic policy analysis. As with any policy learning algorithm, when applied to real-world scenarios such as dynamic pricing, the objective function should ensure that it does not inadvertently lead to discriminatory outcomes or unfair treatment of specific subpopulations. We encourage practitioners to conduct fairness audits when deploying such algorithms in socially sensitive contexts.

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

# A. Proof of Results in Section 2

*Proof of Lemma 2.8.* By Theorem 1 of Hu & Hong (2013), the strong duality holds:

$$\inf_{\mathbf{P} \in \mathcal{U}_{\mathbf{P}_0}(\delta)} \mathbb{E}_{\mathbf{P}}[r(X, \pi(X), Z)] = \sup_{\alpha \geq 0} \left\{ -\alpha \log \mathbb{E}_{\mathbf{P}_0} \left[ \exp \left( -\frac{r(X, \pi(X), Z)}{\alpha} \right) \right] - \alpha\delta \right\}.$$

For brevity, let $r \triangleq r(X, \pi(X), Z)$. To prove that $\varphi(\pi, \alpha)$ is strictly concave for $\alpha > 0$, we analyze its second derivative.

$$\frac{\partial^2}{\partial \alpha^2} \varphi(\pi, \alpha) = \frac{1}{\alpha^3 \mathbb{E}_{\mathbf{P}_0}[\exp(-r/\alpha)]} \left( \frac{(\mathbb{E}_{\mathbf{P}_0}[r \exp(-r/\alpha)])^2}{\mathbb{E}_{\mathbf{P}_0}[\exp(-r/\alpha)]} - \mathbb{E}_{\mathbf{P}_0} \left[ r^2 \exp(-r/\alpha) \right] \right).$$

By the Cauchy-Schwarz inequality,

$$\mathbb{E}_{\mathbf{P}_0}[r \exp(-r/\alpha)]^2 \leq \mathbb{E}_{\mathbf{P}_0}[r^2 \exp(-r/\alpha)] \mathbb{E}_{\mathbf{P}_0}[\exp(-r/\alpha)].$$

Since $\mathbb{E}_{\mathbf{P}_0}[\exp(-r/\alpha)] > 0$, we have $\frac{\partial^2}{\partial \alpha^2} \varphi(\pi, \alpha) \leq 0$. The Cauchy-Schwarz inequality holds strictly as an equality if and only if $r$ is almost surely constant. Under Assumption 2.3, $\mathbf{P}_0(Y = 0) \in [\underline{\rho}, \overline{\rho}]$, ensuring that $r$ is non-constant almost surely. Thus, $\frac{\partial^2}{\partial \alpha^2} \varphi(\pi, \alpha) < 0$, which guarantees strict concavity.

Because $r \in [0, 1]$, we obtain $R_\delta(\pi) \geq 0$. Furthermore, $r \leq 1$ implies $\mathbb{E}_{\mathbf{P}_0}[\exp(-r/\alpha)] \geq \exp(-1/\alpha)$, yielding the upper bound $\varphi(\pi, \alpha) \leq 1 - \alpha\delta$. By strong duality, the optimal dual solution $\alpha^*(\pi)$ must satisfy:

$$0 \leq R_\delta(\pi) = \varphi(\pi, \alpha^*(\pi)) \leq 1 - \alpha^*(\pi)\delta \implies \alpha^*(\pi) \leq \frac{1}{\delta} \triangleq \bar{\alpha}.$$

Note that $\lim_{\alpha \to 0} \varphi(\pi, \alpha) = 0$. If the optimal solution were $\alpha^*(\pi) = 0$, the maximum robust value would be exactly 0, which contradicts the assumption $R_\delta(\pi) > 0$. Therefore, $\alpha^*(\pi) \in (0, \bar{\alpha}]$.

We now prove that the condition $\delta < -\log \overline{\rho}$ is sufficient to guarantee $R_\delta(\pi) > 0$. Since $\lim_{\alpha \to 0^+} \varphi(\pi, \alpha) = 0$, it suffices to show that $\lim \inf_{\alpha \to 0^+} \frac{\partial}{\partial \alpha} \varphi(\pi, \alpha) > 0$. The first derivative of $\varphi(\pi, \alpha)$ for $\alpha > 0$ is given by:

$$\frac{\partial}{\partial \alpha} \varphi(\pi, \alpha) = -\log \mathbb{E}_{\mathbf{P}_0} \left[ \exp(-r/\alpha) \right] - \frac{\mathbb{E}_{\mathbf{P}_0} \left[ r/\alpha \exp(-r/\alpha) \right]}{\mathbb{E}_{\mathbf{P}_0} \left[ \exp(-r/\alpha) \right]} - \delta.$$

We evaluate the limits of the expectation terms as $\alpha \to 0^+$. For the first term, since $r \geq 0$, we have $\exp(-r/\alpha) \leq 1$. By the Dominated Convergence Theorem, we can interchange the limit and the expectation:

$$\lim_{\alpha \to 0^+} \mathbb{E}_{\mathbf{P}_0} \left[ \exp(-r/\alpha) \right] = \mathbb{E}_{\mathbf{P}_0} \left[ \lim_{\alpha \to 0^+} (\mathbf{1}\{Y = 0\} + \mathbf{1}\{Y = 1\} \exp(-r/\alpha)) \right] = \mathbf{P}_0(Y = 0).$$

For the second term, since $\lim_{x \to \infty} xe^{-x} = 0$, applying the Dominated Convergence Theorem yields $\lim_{\alpha \to 0^+} \mathbb{E}_{\mathbf{P}_0}[r/\alpha \exp(-r/\alpha)] = 0$. Substituting these limits back into the derivative gives

$$\lim_{\alpha \to 0^+} \frac{\partial}{\partial \alpha} \varphi(\pi, \alpha) = -\log(\mathbf{P}_0(Y = 0)) - \delta.$$

By Assumption 2.3, we know $-\log(\mathbf{P}_0(r = 0)) \geq -\log \overline{\rho}$. Therefore, if $\delta < -\log \overline{\rho}$, we obtain

$$\lim_{\alpha \to 0^+} \frac{\partial}{\partial \alpha} \varphi(\pi, \alpha) \geq -\log \overline{\rho} - \delta > 0,$$

which completes the proof. $\qquad \square$

# B. Proof of Results in Section 3

We first introduce the assumptions for the asymptotic analysis.

**Assumption B.1.** The oracle logging policy $\pi_0(p|x)$ is three-times differentiable with respect to $p \in \mathcal{P}$ for any $x \in \mathcal{X}$. Furthermore, there exists a compact neighborhood $\mathcal{A}$ of the true dual parameter $\alpha^*$, such that for all $\alpha \in \mathcal{A}$, the nuisance function function $\hat{\lambda}_1^{(\ell)}(x, p; \alpha)$ has bounded first-order and second-order derivative with respect to $p \in \mathcal{P}$ for any $x \in \mathcal{X}$ and $\ell \in [L]$, in probability, i.e.,

$$\sup_{x,p,\alpha \in \mathcal{A}} \left\| \frac{\partial^s}{\partial p^s} \hat{\lambda}_1^{(\ell)}(x, p; \alpha) \right\|_\infty = \mathcal{O}_p(1), \quad s = 1, 2.$$

And $\hat{\lambda}_2^{(\ell)}(x, p) \geq \eta$ for all $x, p \in \mathcal{X} \times \mathcal{P}, \ell \in [L]$.

**Assumption B.2.** There exist a compact neighborhood $\mathcal{B}_\theta$ of $\theta^*$ and a neighborhood $\mathcal{B}_\zeta$ of $\zeta^*$ such that for all $\theta \in \mathcal{B}_\theta$ and $\zeta \in \mathcal{B}_\zeta$
(i) $\nabla_\theta \psi(V; \theta, \zeta)$ is Lipschitz continuous in $\theta$ such that for any $\theta_1, \theta_2 \in \mathcal{B}_\theta$, $\|\nabla_\theta \psi(V; \theta_1, \zeta) - \nabla_\theta \psi(V; \theta_2, \zeta)\| \leq L(V, h)\|\theta_1 - \theta_2\|$, where the Lipschitz constant satisfies $\mathbb{E}[L(V, h)] = \mathcal{O}(1)$.
(ii) The conditional second moment is uniformly bounded, $\sup_{x,p} \sup_{(\theta,\zeta) \in \mathcal{B}_\theta \times \mathcal{B}_\zeta} \mathbb{E}[\|\nabla_\theta \psi(V; \theta, \zeta)\|^2 | X = x, P = p] \leq \mathcal{O}(1)$.
(iii) For any fixed $\alpha$ in a compact neighborhood $\mathcal{A}$ of $\alpha^*$, the map $\zeta \mapsto \mathbb{E}[\nabla_\theta \psi(V; \theta^*, \zeta)]$ is Lipschitz continuous in a neighborhood $\mathcal{B}_\zeta$ of $\zeta^*$, i.e., $\|\mathbb{E}[\nabla_\theta \psi(V; \theta^*, \zeta_1(x, p; \alpha))] - \mathbb{E}[\nabla_\theta \psi(V; \theta^*, \zeta_2(x, p; \alpha))]\| \leq C_\psi \|\zeta_1 - \zeta_2\|$ for some Lipschitz constant $C_\psi < \infty$.

Assumption B.2 is tailored to adapt Assumption 2 in Kallus et al. (2024) to the continuous pricing setting. Specifically, Assumption B.2 (i)-(ii) directly corresponds to Assumption 2 (ii) and (iv-v) in Kallus et al. (2024), ensuring the existence and stability of the Jacobian and covariance matrices for the joint moment equations. Assumption B.2 (iii) is a localized version of the Fréchet differentiability requirement discussed in Proposition 5 of Kallus et al. (2024), which is the fundamental pillar for the localization technique to achieve semiparametric efficiency using only a preliminary estimate $\hat{\alpha}_{init}$.

With the regressions $m_0$ and $m_1$, we can show that the Gateaux derivatives of $\mathbb{E}[\psi(V; \theta, \zeta)]$ are zero as $h \to 0$, when evaluated at $\theta^* = (\alpha^*, W_0^*, W_1^*, R_\delta(\pi))$, $\lambda_1(\cdot; \alpha) = \lambda_1^*(\cdot; \alpha^*)$ and $\lambda_2(\cdot) = \lambda_2^*(\cdot)$ respectively.

**Lemma B.3** (Neyman Orthogonality in S2 of Online supplement (Colangelo & Lee, 2026)). *Under Assumption B.1, B.2, the Gateaux derivative of the moment mapping at the oracle parameter $\theta^*$ and nuisance $\zeta^*$ in the direction $\Delta\zeta = \zeta - \zeta^*$ is given by*

$$|dM(\theta^*, \zeta^*; \Delta_\zeta)| = \left| \frac{\partial}{\partial \tau} \mathbb{E}[\psi(V; \theta^*, \zeta^* + \tau\Delta_\zeta)] \Big|_{\tau=0} \right| = \mathcal{O}_p(h^s), \tag{11}$$

*where $s = 1$ under Lipschitz continuity and $s = 2$ under second-order bounded derivative in Assumption B.1. Consequently, as $h \to 0$, the moment condition is first-order insensitive to the estimation errors of the nuisances.*

*Proof of Theorem 3.3.* We decompose the proof into three main components: (i) control of errors introduced by dual-variable localization and nuisance estimation; (ii) sensitivity and stability analysis of the joint moment equations (Jacobian consistency); and (iii) the linear representation via empirical process theory.

**Step 1: Nuisance Estimation and Localization Error.** We first examine the sensitivity of the outcome model $\lambda_1^*(x, p; \alpha)$ with respect to $\alpha$. The derivative of $\lambda_1^*(x, p; \alpha)$ is

$$\frac{\partial}{\partial \alpha} \lambda_1^*(x, p; \alpha) = \begin{pmatrix} \frac{1}{\alpha^2} \exp(-p/\alpha) \mathbf{P}(Y = 1 | X = x, P = p) \\ \frac{p}{\alpha^2} \exp(-p/\alpha) \mathbf{P}(Y = 1 | X = x, P = p) \\ 0 \\ 0 \end{pmatrix},$$

is uniformly bounded by $\frac{4}{p_{min}^2 e^2}$ for $p \in [p_{\min}, 1]$, $x \in \mathcal{X}$ and $\alpha > 0$, which implies that

$$|\lambda_1^*(x, p; \hat{\alpha}_{init}) - \lambda_1^*(x, p; \alpha^*)| \leq \frac{4}{p_{min}^2 e^2} |\hat{\alpha}_{init} - \alpha^*|. \tag{12}$$

Combining this with Assumption 3.1, the total error in the outcome nuisance function under the localization strategy is bounded by

$$|\lambda_1(x, p; \hat{\alpha}_{init}) - \lambda_1^*(x, p; \alpha^*)| \leq |\lambda_1(x, p; \hat{\alpha}_{init}) - \lambda_1^*(x, p; \hat{\alpha}_{init})| + |\lambda_1^*(x, p; \hat{\alpha}_{init}) - \lambda_1^*(x, p; \alpha^*)|$$

$$\leq o_p(T^{-\rho_m}) + \frac{4}{p_{min}^2 e^2}|\hat{\alpha}_{init} - \alpha^*|$$

$$\leq o_p(T^{-\rho_m} + T^{-\rho_\alpha}),$$

$$= o_p(T^{-\rho_m \wedge \rho_\alpha}), \tag{13}$$

and $$|\lambda_2(x,p) - \lambda_2^*(x,p)| \leq o_p(T^{-\rho_\pi}). \tag{14}$$

Therefore, we obtain $\|\zeta(\cdot; \hat{\alpha}_{init}) - \zeta^*(\cdot; \alpha^*)\|_\infty \leq o_p(T^{-\rho_m \wedge \rho_\alpha} + T^{-\rho_\pi})$.

**Step 2: Score Decomposition and Neyman Orthogonality.** For each fold $\ell \in [L]$, denote $\hat{\zeta}^{(\ell)}(x,p) = (\hat{\lambda}_1^{(\ell)}(x,p; \hat{\alpha}_{init}^{(\ell)}), \hat{\lambda}_2^{(\ell)}(x,p)), \Delta_{\lambda_1}^{(\ell)}(x,p) = \hat{\lambda}_1^{(\ell)}(x,p; \hat{\alpha}_{init}^{(\ell)}) - \lambda_1^*(x,p; \alpha^*)$ and $\Delta_{\lambda_2}^{(\ell)}(x,p) = \hat{\lambda}_2^{(\ell)}(x,p) - \lambda_2^*(x,p)$. Utilizing the cross-fitting structure, we decompose

$$\sqrt{n_\ell h} \frac{1}{n_\ell} \sum_{t \in \mathcal{I}_\ell} \left( \psi(v_t; \theta^*, \hat{\zeta}^{(\ell)}(v_t)) - \psi(v_t; \theta^*, \zeta^*) \right)$$

$$= \sqrt{\frac{h}{n_\ell}} \sum_{t \in \mathcal{I}_\ell} \left\{ \left[ \hat{\lambda}_1^{(\ell)}(x_t, \pi(x_t); \hat{\alpha}_{init}^{(\ell)}) - \lambda_1^*(x_t, \pi(x_t); \alpha^*) \right] \right.$$

$$\left. + \left[ \frac{K(\frac{p_t - \pi(x_t)}{h})}{h\hat{\lambda}_2^{(\ell)}(x_t, p_t)}(U(p_t y_t; \alpha^*) - \hat{\lambda}_1^{(\ell)}(x_t, p_t; \hat{\alpha}_{init}^{(\ell)})) - \frac{K(\frac{p_t - \pi(x_t)}{h})}{h\lambda_2^*(x_t, p_t)}(U(p_t y_t; \alpha^*) - \lambda_1^*(x_t, p_t; \alpha^*)) \right] \right\}$$

$$= \sqrt{\frac{h}{n_\ell}} \sum_{t \in \mathcal{I}_\ell} \left\{ \mathcal{T}_{1,t}^{(\ell)} + \mathcal{T}_{2,t}^{(\ell)} + \mathcal{T}_{DR,t}^{(\ell)} + \mathcal{T}_{prod,t}^{(\ell)} \right\},$$

where

$$\mathcal{T}_{1,t}^{(\ell)} = \left[ \Delta_{\lambda_1}^{(\ell)}(x_t, \pi(x_t)) - \frac{K(\frac{p_t - \pi(x_t)}{h})}{h\lambda_2^*(x_t, p_t)} \Delta_{\lambda_1}^{(\ell)}(x_t, p_t) \right] - \mathbb{E}\left[ \Delta_{\lambda_1}^{(\ell)}(X, \pi(X)) - \frac{K(\frac{P - \pi(X)}{h})}{h\lambda_2^*(X, P)} \Delta_{\lambda_1}^{(\ell)}(X, P) \Big| \mathcal{I}_\ell^c \right]$$

$$\mathcal{T}_{2,t}^{(\ell)} = \frac{K(\frac{p_t - \pi(x_t)}{h})}{h}(U(p_t y_t; \alpha^*) - \lambda_1^*(x_t, p_t; \alpha^*)) \left( \frac{1}{\hat{\lambda}_2^{(\ell)}(x_t, p_t)} - \frac{1}{\lambda_2^*(x_t, p_t)} \right)$$

$$- \mathbb{E}\left[ \frac{K(\frac{P - \pi(X)}{h})}{h}(U(PY; \alpha^*) - \lambda_1^*(X, P; \alpha^*)) \left( \frac{1}{\hat{\lambda}_2^{(\ell)}(X, P)} - \frac{1}{\lambda_2^*(X, P)} \right) \Big| \mathcal{I}_\ell^c \right]$$

$$\mathcal{T}_{DR}^{(\ell)} = \mathbb{E}\left[ \Delta_{\lambda_1}^{(\ell)}(X, \pi(X)) - \frac{K(\frac{P - \pi(X)}{h})}{h\lambda_2^*(X, P)} \Delta_{\lambda_1}^{(\ell)}(X, P) \Big| \mathcal{I}_\ell^c \right]$$

$$\mathcal{T}_{prod,t}^{(\ell)} = -\frac{K(\frac{p_t - \pi(x_t)}{h})}{h} \Delta_{\lambda_1}^{(\ell)}(x_t, p_t) \left( \frac{1}{\hat{\lambda}_2^{(\ell)}(x_t, p_t)} - \frac{1}{\lambda_2^*(x_t, p_t)} \right).$$

By cross-fitting, $\hat{\zeta}^{(\ell)} = (\hat{\lambda}_1^{(\ell)}, \hat{\lambda}_2^{(\ell)})$ are trained on the out-of-fold data $\mathcal{I}_\ell^c$ and are thus fixed functions. For the first term $\mathcal{T}_{1,t}^{(\ell)}$, since the summands are i.i.d. and mean-zero conditional on $\mathcal{I}_\ell^c$, the variance of the sum is

$$\text{Var}\left( \sqrt{\frac{h}{n_\ell}} \sum_{t \in \mathcal{I}_\ell} \mathcal{T}_{1,t}^{(\ell)} \Big| \mathcal{I}_\ell^c \right) = h \cdot \text{Var}\left( \mathcal{T}_{1,t}^{(\ell)} \Big| \mathcal{I}_\ell^c \right)$$

$$\leq 2h\mathbb{E}\left[ \left( \Delta_{\lambda_1}^{(\ell)}(X, \pi(X)) \right)^2 \Big| \mathcal{I}_\ell^c \right] + 2h\mathbb{E}\left[ \left( \frac{K(\frac{P - \pi(X)}{h})}{h\lambda_2^*(X, P)} \Delta_{\lambda_1}^{(\ell)}(X, P) \right)^2 \Big| \mathcal{I}_\ell^c \right]$$

$$\leq 2h\|\Delta_{\lambda_1}^{(\ell)}\|_\infty^2 + 2\|\Delta_{\lambda_1}^{(\ell)}\|_\infty^2 \cdot \mathbb{E}\left[ \int_{\mathcal{P}} \frac{K^2(\frac{P - \pi(X)}{h})}{h} dp \right]$$

$$= 2h\|\Delta_{\lambda_1}^{(\ell)}\|_\infty^2 + 2\|\Delta_{\lambda_1}^{(\ell)}\|_\infty^2 \cdot \mathbb{E}\left[ \int_{\mathcal{P}} K^2(u) du \right]$$

$$\leq 2h\|\Delta^{(\ell)}_{\lambda_1}\|^2_\infty + 2C_3\|\Delta^{(\ell)}_{\lambda_1}\|^2_\infty$$
$$= o_p(T^{-2\rho_m} + T^{-2\rho_\alpha}) = o_p(1),$$

where the last equality follows from (13). By Chebyshev's inequality, it follows that

$$\sqrt{\frac{h}{n_\ell}} \sum_{t \in \mathcal{I}_\ell} \mathcal{T}^{(\ell)}_{1,t} = o_p(1). \tag{15}$$

For the second term $\mathcal{T}^{(\ell)}_{2,t}$, since $\mathbb{E}[U(PY;\alpha^*)|X,P] = \lambda^*_1(X,P;\alpha^*)$, we get a similar variance bound as

$$\mathrm{Var}\left(\sqrt{\frac{h}{n_\ell}} \sum_{t \in \mathcal{I}_\ell} \mathcal{T}^{(\ell)}_{2,t} \Big| \mathcal{I}^c_\ell\right) = h \cdot \mathbb{E}\left[\frac{K^2(\frac{P-\pi(X)}{h})}{h^2}(U(PY;\alpha^*) - \lambda^*_1(X,P;\alpha^*))^2 \left(\frac{1}{\hat{\lambda}^{(\ell)}_2(x_t,p_t)} - \frac{1}{\lambda^*_2(x_t,p_t)}\right)^2 \Big| \mathcal{I}^c_\ell\right]$$

$$\leq \frac{C_3}{\eta^3}\|\Delta^{(\ell)}_{\lambda_2}\|^2_\infty = o_p(T^{-2\rho_\pi}),$$

where the last equality follows from (14). Then by Chebyshev's inequality, it follows that

$$\sqrt{\frac{h}{n_\ell}} \sum_{t \in \mathcal{I}_\ell} \mathcal{T}^{(\ell)}_{2,t} = o_p(1). \tag{16}$$

For the third term $\mathcal{T}^{(\ell)}_{DR}$, we discuss in two cases. Case 1. First-order bounded derivative in Assumption B.1.

$$\sqrt{n_\ell h}\left|\mathcal{T}^{(\ell)}_{DR}\right| = \sqrt{n_\ell h}\left|\mathbb{E}\left[\Delta^{(\ell)}_{\lambda_1}(X,\pi(X)) - \frac{K(\frac{P-\pi(X)}{h})}{h\lambda^*_2(X,P)}\Delta^{(\ell)}_{\lambda_1}(X,P)\Big|\mathcal{I}^c_\ell\right]\right|$$

$$= \sqrt{n_\ell h}\left|\mathbb{E}_X\left[\Delta^{(\ell)}_{\lambda_1}(X,\pi(X)) - \int_\mathcal{P} \frac{1}{h}K\left(\frac{p-\pi(X)}{h}\right)\Delta^{(\ell)}_{\lambda_1}(X,p)dp\right]\right|$$

$$\leq \sqrt{n_\ell h}\left|\mathbb{E}_X\left[\int K(u)\Delta^{(\ell)}_{\lambda_1}(X,\pi(X)) - \Delta^{(\ell)}_{\lambda_1}(X,\pi(X)+uh)\Big|du\right]\right.$$

$$= \sqrt{n_\ell h}\left|\mathbb{E}_X\left[\int K(u)uh\left\|\frac{\partial}{\partial p}\Delta^{(\ell)}_{\lambda_1}(X,\tilde{p})\right\|_\infty du\right]\right| \leq \mathcal{O}_p\left(\sqrt{n_\ell h^3}\right), \tag{17}$$

where we use Assumption 3.2, Lipschitz continuity of $\Delta^{(\ell)}_{\lambda_1}$ in Assumption B.1 and 2.4, and the initial estimate $\hat{\alpha}_{init} = \alpha^* + o_p(1)$. Case 2. Second-order bounded derivative in Assumption B.1.

$$\sqrt{n_\ell h}\left|\mathcal{T}^{(\ell)}_{DR}\right| = \sqrt{n_\ell h}\left|\mathbb{E}\left[\Delta^{(\ell)}_{\lambda_1}(X,\pi(X)) - \frac{K(\frac{P-\pi(X)}{h})}{h\lambda^*_2(X,P)}\Delta^{(\ell)}_{\lambda_1}(X,P)\Big|\mathcal{I}^c_\ell\right]\right|$$

$$= \sqrt{n_\ell h}\left|\mathbb{E}_X\left[\Delta^{(\ell)}_{\lambda_1}(X,\pi(X)) - \int_\mathcal{P} \frac{1}{h}K\left(\frac{p-\pi(X)}{h}\right)\Delta^{(\ell)}_{\lambda_1}(X,p)dp\right]\right|$$

$$\leq \sqrt{n_\ell h}\left|\mathbb{E}_X\left[\int K(u)\Delta^{(\ell)}_{\lambda_1}(X,\pi(X)) - \Delta^{(\ell)}_{\lambda_1}(X,\pi(X)+uh)\Big|du\right]\right.$$

$$= \sqrt{n_\ell h}\left|\mathbb{E}_X\left[\int K(u)\frac{(uh)^2}{2}\frac{\partial^2}{\partial p^2}\Delta^{(\ell)}_{\lambda_1}(X,\tilde{p})du\right]\right| \leq \mathcal{O}_p\left(\sqrt{n_\ell h^5}\right). \tag{18}$$

where we use Assumption 3.2 and a second-order Taylor expansion of the nuisance error $\Delta^{(\ell)}_{\lambda_1}$ around $\pi(x)$,

$$\Delta^{(\ell)}_{\lambda_1}(x,\pi(x)+uh) = \Delta^{(\ell)}_{\lambda_1}(x,\pi(x)) + (uh)\frac{\partial}{\partial p}\Delta^{(\ell)}_{\lambda_1}(x,\pi(x)) + \frac{(uh)^2}{2}\frac{\partial^2}{\partial p^2}\Delta^{(\ell)}_{\lambda_1}(x,\tilde{p}),$$

where $\tilde{p}$ lies between $\pi(x)$ and $\pi(x) + uh$. And the last inequality directly follows from Assumption B.1 and the initial estimate $\hat{\alpha}_{init} = \alpha^* + o_p(1)$.

For the fourth term $\mathcal{T}_{prod,t}^{(\ell)}$, by using the change of variables $u = (p - \pi(x))/h$ and (13)-(14), we reduce its expectation to

$$\sqrt{Th}\left|\mathbb{E}\left[\mathcal{T}_{prod,t}^{(\ell)}\Big|\mathcal{I}_\ell^c\right]\right| \le \mathcal{O}(\sqrt{Th}\cdot\|\Delta_{\lambda_1}^{(\ell)}\|_\infty\cdot\|\Delta_{\lambda_2}^{(\ell)}\|_\infty) \le o_p(T^{\frac12-\rho_\pi-\rho_\alpha\wedge\rho_m}\cdot h^{1/2}) = o_p(1),$$

and the variance is bounded by

$$\mathrm{Var}\left(\sqrt{\frac{h}{n_\ell}}\sum_{t\in\mathcal{I}_\ell}\mathcal{T}_{prod,t}^{(\ell)}\Big|\mathcal{I}_\ell^c\right)$$

$$= h\cdot\mathrm{Var}\left(\mathcal{T}_{prod,t}^{(\ell)}\Big|\mathcal{I}_\ell^c\right) \le h\cdot\mathbb{E}\left[\left(\mathcal{T}_{prod,t}^{(\ell)}\right)^2\Big|\mathcal{I}_\ell^c\right] \le h\int\frac{1}{h^2}K^2\left(\frac{p-\pi(x)}{h}\right)\frac{\|\Delta_{\lambda_1}^{(\ell)}\|_\infty^2\cdot\|\Delta_{\lambda_2}^{(\ell)}\|_\infty^2}{\eta^3}dpdF_X$$

$$= \frac{1}{\eta^3}\int K^2(u)\|\Delta_{\lambda_1}^{(\ell)}\|_\infty^2\cdot\|\Delta_{\lambda_2}^{(\ell)}\|_\infty^2 dudF_X \le \frac{C_3}{\eta^3}\|\Delta_{\lambda_1}^{(\ell)}\|_\infty^2\cdot\|\Delta_{\lambda_2}^{(\ell)}\|_\infty^2$$

$$\le o_p(T^{-2\rho_\pi-2\rho_\alpha} + T^{-2\rho_\pi-2\rho_m}) = o_p(1).$$

By applying Chebyshev's inequality conditionally on $\mathcal{I}_\ell^c$, we obtain $\sqrt{n_\ell h}\left|\frac{1}{n_\ell}\sum_{t\in\mathcal{I}_\ell}\mathcal{T}_{prod,t}^{(\ell)} - \mathbb{E}\left[\mathcal{T}_{prod,t}^{(\ell)}\Big|\mathcal{I}_\ell^c\right]\right| = o_p(1)$.
Furthermore, we get

$$\left|\sqrt{\frac{h}{n_\ell}}\sum_{t\in\mathcal{I}_\ell}\mathcal{T}_{prod,t}^{(\ell)}\right| \le \left|\sqrt{n_\ell h}\left(\frac{1}{n_\ell}\sum_{t\in\mathcal{I}_\ell}\mathcal{T}_{prod,t}^{(\ell)} - \mathbb{E}\left[\mathcal{T}_{prod,t}^{(\ell)}\Big|\mathcal{I}_\ell^c\right]\right)\right| + \sqrt{n_\ell h}\left|\mathbb{E}\left[\mathcal{T}_{prod,t}^{(\ell)}\Big|\mathcal{I}_\ell^c\right]\right| \le o_p(1). \quad (19)$$

To proceed, picking $\sqrt{Th^{1+2s}}\to 0$ and combining (13)-(19), we conclude that

$$\sqrt{n_\ell h}\frac{1}{n_\ell}\sum_{t\in\mathcal{I}_\ell}\left(\psi(v_t;\theta^*,\hat\zeta^{(\ell)}(v_t)) - \psi(v_t;\theta^*,\zeta^*)\right) = o_p(1). \quad (20)$$

**Step 3: Jacobian Stability and Asymptotic Distribution.** Denote $\theta^* = [\alpha^*, W_0^*, W_1^*, R_\delta(\pi)]^\top$. Since

$$\mathbb{E}[\psi(V;\theta,\zeta^*)] = \mathbb{E}\left[\lambda_1^*(X,\pi(X);\alpha) + \frac{K(\frac{P-\pi(X)}{h})}{h\pi_0(P|X)}\left(U(PY;\alpha) - \lambda_1^*(X,P;\alpha)\right) + G(\theta)\right]$$

$$= \mathbb{E}[\lambda_1^*(X,\pi(X);\alpha)] + G(\theta)$$

$$= \begin{pmatrix} \mathbb{E}[\exp(-\pi(X)Y(\pi(X))/\alpha)] - W_0 \\ \mathbb{E}[\pi(X)Y(\pi(X))\exp(-\pi(X)Y(\pi(X))/\alpha)] - W_1 \\ -\delta - \log W_0 - \frac{W_1}{\alpha W_0} \\ -R_\delta(\pi) - \alpha\log W_0 - \alpha\delta \end{pmatrix},$$

following the proof of Theorem 3.2 in Kallus et al. (2022), by replacing $r$ with $\pi(X)Y(\pi(X))$, the Jacobian is

$$J(\theta) = \begin{bmatrix} \frac{1}{\alpha^2}\mathbb{E}\left[\pi(X)Y(\pi(X))\exp(-\pi(X)Y(\pi(X))/\alpha)\right] & -1 & 0 & 0 \\ \frac{1}{\alpha^2}\mathbb{E}\left[\pi^2(X)Y^2(\pi(X))\exp(-\pi(X)Y(\pi(X))/\alpha)\right] & 0 & -1 & 0 \\ \frac{W_1}{\alpha^2 W_0} & -\frac{1}{W_0}+\frac{W_1}{\alpha W_0^2} & -\frac{1}{\alpha W_0} & 0 \\ -\log W_0 - \delta & -\frac{\alpha}{W_0} & 0 & -1 \end{bmatrix}, \quad (21)$$

and substituting in the optimal value, we have $J_{\theta^*} = \mathbb{E}[\nabla_\theta\psi(V;\theta,\zeta^*)]|_{\theta=\theta^*}$ is given by

$$J^* = J_{\theta^*} = \begin{pmatrix} W_1^*/\alpha^{*2} & -1 & 0 & 0 \\ W_2^*/\alpha^{*2} & 0 & -1 & 0 \\ \frac{W_1^*}{\alpha^{*2}W_0^*} & -\frac{1}{W_0^*}+\frac{W_1^*}{\alpha^*W_0^{*2}} & -\frac{1}{\alpha^*W_0^*} & 0 \\ -\log W_0^* - \delta & -\frac{\alpha^*}{W_0^*} & 0 & -1 \end{pmatrix},$$

and $J^*$ is invertible with $\det J^* = -\varphi''(\pi,\alpha^*) > 0$ from Lemma 2.8, and the inverse of $J^*$ is given by

$$J^{*-1} = -\frac{1}{\varphi''(\pi,\alpha^*)}\begin{pmatrix} -\frac{1}{W_0^*}+\frac{W_1^*}{\alpha^*W_0^{*2}} & -\frac{1}{\alpha^*W_0^*} & 1 & 0 \\ \frac{W_2^*/\alpha^*-W_1^*}{\alpha^{*2}W_0^*} & -\frac{W_1^*}{\alpha^{*3}W_0^*} & \frac{W_1^*}{\alpha^{*2}} & 0 \\ \frac{W_2^*}{\alpha^{*3}W_0^*}\left(\frac{W_1^*}{W_0^*}-\alpha^*\right) & -\frac{W_1^{*2}}{\alpha^{*3}W_0^{*2}} & \frac{W_2^*}{\alpha^{*2}} & 0 \\ X_1 & \frac{X_2}{\alpha^*W_0^*} & -X_2 & -\varphi''(\pi,\alpha^*) \end{pmatrix},$$

where $X_1 = \left(\frac{W_1^*}{\alpha^* W_0^{*2}} - \frac{1}{W_0^*}\right)\left(-\log W_0^* - \delta\right) - \frac{1}{\alpha^* W_0^{*2}}(W_1^* - W_2^*/\alpha^*)$ and $X_2 = W_1^*/(\alpha^* W_0^*) + \log W_0^* + \delta$.

Let $\hat{\theta} = [\hat{\alpha}, \hat{W}_0, \hat{W}_1, \hat{R}_{\mathrm{DRO}}]^\top$ be the solution to the empirical equation

$$\mathbb{E}_T[\psi(V; \theta, \hat{\zeta})] \triangleq \frac{1}{T}\sum_{\ell=1}^{L}\sum_{t\in\mathcal{I}_\ell}\psi(v_t; \theta, \hat{\zeta}^{(\ell)}) = \mathbf{0}. \tag{22}$$

By second order Taylor expansion around $(\theta^*, \zeta^*)$,

$$\mathbf{0} = \mathbb{E}_T[\psi(V; \hat{\theta}, \hat{\zeta})] = \mathbb{E}_T[\psi(V; \theta^*, \zeta^*)] + J^*(\hat{\theta} - \theta^*) + \mathbb{E}_T[\psi(V; \theta^*, \hat{\zeta}) - \psi(V; \theta^*, \zeta^*)] + \mathcal{R}_T, \tag{23}$$

where $\mathcal{R}_T$ represents high-order Taylor remainders. Specifically,

$$\mathcal{R}_T = (\hat{J}_T(\hat{\zeta}) - J^*)(\hat{\theta} - \theta^*) + \frac{1}{2}\nabla_\theta^2 \mathbb{E}_T[\psi(V; \tilde{\theta}, \hat{\zeta})](\hat{\theta} - \theta^*)^2,$$

where $\tilde{\theta}$ lies between $\theta^*$ and $\hat{\theta}$, $\hat{J}_T(\hat{\zeta}) := \nabla_\theta \mathbb{E}_T[\psi(V; \theta, \hat{\zeta})]|_{\theta^*}$ denotes the empirical Jacobian matrix evaluated with localized nuisance estimators.

Here we show the $\sqrt{Th}\mathcal{R}_T = o_p(1)$. First, we show the $\hat{J}_T(\hat{\zeta}) = J^* + o_p(1)$.

$$\begin{aligned}
|\hat{J}_T(\hat{\zeta}) - J^*| &= \left|\frac{1}{T}\sum_{\ell=1}^{L}\sum_{t\in\mathcal{I}_\ell}\left(\nabla_\theta\psi(v_t; \theta, \hat{\zeta}^{(\ell)}) - \mathbb{E}[\nabla_\theta\psi(V; \theta, \zeta^*)]\right)_{\theta=\theta^*}\right| \\
&\leq \frac{1}{L}\left|\frac{1}{|\mathcal{I}_\ell|}\sum_{\ell=1}^{L}\sum_{t\in\mathcal{I}_\ell}\nabla_\theta\psi(v_t; \theta^*, \hat{\zeta}^{(\ell)}) - \mathbb{E}[\nabla_\theta\psi(V; \theta^*, \hat{\zeta}^{(\ell)})]\right| + \left|\mathbb{E}[\nabla_\theta\psi(V; \theta^*, \hat{\zeta}^{(\ell)})] - \mathbb{E}[\nabla_\theta\psi(V; \theta^*, \zeta^*)]\right|
\end{aligned}$$

By cross-fitting, $\hat{\zeta}^{(\ell)}$ is fixed given the training data $\mathcal{I}_\ell^c$. Due to (13)-(14), $\hat{\zeta}^{(\ell)} = \zeta^* + o_p(1)$ and using Assumption B.2(ii), the variance of average over $\mathcal{I}_\ell$ samples scales as

$$\mathrm{Var}\left(\frac{1}{|\mathcal{I}_\ell|}\sum_{t\in\mathcal{I}_\ell}\nabla_\theta\psi(v_t; \theta^*, \hat{\zeta}^{(\ell)})\bigg|\mathcal{I}_\ell^c\right) = \frac{1}{|\mathcal{I}_\ell|}\mathrm{Var}\left(\nabla_\theta\psi(v_t; \theta^*, \hat{\zeta}^{(\ell)})|\mathcal{I}_\ell^c\right) \leq \mathcal{O}_p\left(\frac{1}{Th}\right)$$

where we use the fact that $\mathbb{E}\left[\frac{K^2(\frac{P-\pi(X)}{h})}{h^2}\right] = \mathcal{O}(\frac{1}{h})$. Then by Chebyshev's inequality, as $Th \to \infty$, we have

$$\left|\frac{1}{|\mathcal{I}_\ell|}\sum_{\ell=1}^{L}\sum_{t\in\mathcal{I}_\ell}\nabla_\theta\psi(v_t; \theta^*, \hat{\zeta}^{(\ell)}) - \mathbb{E}[\nabla_\theta\psi(V; \theta^*, \hat{\zeta}^{(\ell)})]\right| = o_p(1), \tag{24}$$

Consider the mapping $\alpha \mapsto \mathbb{E}[\nabla_\theta\psi(V; \theta^*, \zeta^*(x, p; \alpha))]$. Note that $\nabla_\theta\psi$ is composed of nuisance $\lambda_1^*$ and $\nabla_\theta G(\theta)$. $\zeta^*(x, p; \alpha)$ is Lipschitz in $\alpha$ from (12). And $W_0 \geq \rho/2$ by picking $h^s \leq \frac{\rho}{2C_1 L_m}$ in (36) and (43), then $\nabla_\theta G(\theta)$ is uniformly bounded in the compact neighborhood $\mathcal{A}$ of $\alpha^*$. Thus, the Jacobian of $\mathbb{E}[\nabla_\theta\psi(V; \theta^*, \zeta^*(x, p; \alpha))]$ is Lipschitz in $\alpha$ with a bounded constant $C_{loc}$. And using Assumption B.2 (iii), we have

$$\begin{aligned}
&\left|\mathbb{E}[\nabla_\theta\psi(V; \theta^*, \hat{\zeta}^{(\ell)})] - \mathbb{E}[\nabla_\theta\psi(V; \theta^*, \zeta^*)]\right| \\
&\leq \left|\mathbb{E}[\nabla_\theta\psi(V; \theta^*, \hat{\zeta}^{(\ell)}(x, p; \hat{\alpha}_{init}^{(\ell)}))] - \mathbb{E}[\nabla_\theta\psi(V; \theta^*, \zeta^*(x, p; \hat{\alpha}_{init}^{(\ell)}))]\right| \\
&\quad + \left|\mathbb{E}[\nabla_\theta\psi(V; \theta^*, \zeta^*(x, p; \hat{\alpha}_{init}^{(\ell)}))] - \mathbb{E}[\nabla_\theta\psi(V; \theta^*, \zeta^*(x, p; \alpha^*))]\right| \\
&\leq C_\Psi\|\hat{\zeta}^{(\ell)}(x, p; \hat{\alpha}_{init}^{(\ell)}) - \zeta^*(x, p; \hat{\alpha}_{init}^{(\ell)})\|_\infty + C_{loc}\|\hat{\alpha}_{init}^{(\ell)} - \alpha^*\|_\infty \leq o_p(1)
\end{aligned} \tag{25}$$

Combining (24) and (25), we have

$$|\hat{J}_T(\hat{\zeta}) - J^*| = o_p(1). \tag{26}$$

Second, by weak law of large numbers and Assumption B.2 (i), we have $\frac{1}{|\mathcal{I}_\ell|}\sum_{t\in\mathcal{I}_\ell}L(v_t,h)=\mathcal{O}_p(1)$. Thus,

$$
\left\|\nabla_\theta^2\mathbb{E}_T[\psi(V;\tilde\theta,\hat\zeta)]\right\| = \left\|\frac{1}{T}\sum_{\ell=1}^L\sum_{t\in\mathcal{I}_\ell}\nabla_\theta^2\psi(v_t;\tilde\theta,\hat\zeta^{(\ell)})\right\|
$$

$$
\leq \frac{1}{T}\sum_{\ell=1}^L\sum_{t\in\mathcal{I}_\ell}\left\|\nabla_\theta^2\psi(v_t;\tilde\theta,\hat\zeta^{(\ell)})\right\| \leq \frac{1}{T}\sum_{\ell=1}^L\sum_{t\in\mathcal{I}_\ell}L(v_t,h)=\mathcal{O}_p(1). \tag{27}
$$

Third, we show that $\|\hat\theta-\theta^*\|=\mathcal{O}_p((Th)^{-1/2})$. By a first order Taylor expansion in (22) around $\theta^*$, we have

$$
\mathbf{0}=\mathbb{E}_T[\psi(V;\theta^*,\hat\zeta)]+\bar{J}_T(\hat\theta-\theta^*),
$$

where $\bar{J}_T=\int_0^1\nabla_\theta\mathbb{E}_T[\psi(V;\theta^*+u(\hat\theta-\theta^*),\hat\zeta)]du$. Due to (26) and (27), we have $\bar{J}_T=J^*+o_p(1)$. Recalling that $J^*$ is invertible, then

$$
(\hat\theta-\theta^*)=-(J^*+o_p(1))^{-1}\mathbb{E}_T[\psi(V;\theta^*,\hat\zeta)]. \tag{28}
$$

We can decompose the empirical expectation as follows

$$
\mathbb{E}_T[\psi(V;\theta^*,\hat\zeta)]=\mathbb{E}_T[\psi(V;\theta^*,\zeta^*)]+(\mathbb{E}_T-\mathbb{E})[\psi(V;\theta^*,\hat\zeta)-\psi(V;\theta^*,\zeta^*)]+\mathbb{E}[\psi(V;\theta^*,\hat\zeta)-\psi(V;\theta^*,\zeta^*)].
$$

For the first term, using the fact that $\mathrm{Var}(\psi(V;\theta^*,\zeta^*))=\mathcal{O}(1/h)$, then $\mathrm{Var}(\mathbb{E}_T[\psi])=O(1/Th)$. By the Central Limit Theorem, we have

$$
\mathbb{E}_T[\psi(V;\theta^*,\zeta^*)]=\mathcal{O}_p((Th)^{-1/2}). \tag{29}
$$

For the second term, we evaluate the conditional variance $\mathrm{Var}\left((\mathbb{E}_T-\mathbb{E})[\psi(V;\theta^*,\hat\zeta^{(\ell)})-\psi(V;\theta^*,\zeta^*)]\mid\mathcal{I}_\ell^c\right)$. Following the variance scaling argument in Equation (A.5) in Colangelo & Lee (2026), we have

$$
\mathrm{Var}\left((\mathbb{E}_T-\mathbb{E})[\psi(V;\theta^*,\hat\zeta^{(\ell)})-\psi(V;\theta^*,\zeta^*)]\mid\mathcal{I}_\ell^c\right)=\mathcal{O}_p\left(\frac{\|\hat\zeta^{(\ell)}-\zeta^*\|_\infty^2}{Th}\right).
$$

Under the assumption that the nuisance estimators are consistent, $\|\hat\zeta^{(\ell)}-\zeta^*\|_\infty=o_p(1)$, and $Th\to\infty$, we have

$$
(\mathbb{E}_T-\mathbb{E})[\psi(V;\theta^*,\hat\zeta^{(\ell)})-\psi(V;\theta^*,\zeta^*)]=o_p((Th)^{-1/2}). \tag{30}
$$

For the third term, by the Neyman orthogonality B.3, we decompose the functional bias into

$$
\mathbb{E}[\psi(V;\theta^*,\hat\zeta)-\psi(V;\theta^*,\zeta^*)]=dM(\zeta^*;\hat\zeta-\zeta^*)+\frac{1}{2}d^2M(\bar\zeta;\hat\zeta-\zeta^*) \qquad (\bar\zeta\text{ lies between }\zeta^*\text{ and }\hat\zeta)
$$

$$
=\mathcal{O}_p(h^s\|\hat\zeta-\zeta^*\|_\infty)+\mathcal{O}_p(\|\hat\lambda_1-\lambda_1^*\|_\infty\|\hat\lambda_2-\lambda_2^*\|_\infty)
$$

$$
=\mathcal{O}_p(h^s(T^{-\rho_m\wedge\rho_\alpha}+T^{-\rho_\pi}))+T^{-(\rho_m\wedge\rho_\alpha)-\rho_\pi}). \qquad \text{(by (13), (14), (31))}
$$

where $\bar\lambda_1$ lies between $\lambda_1^*$ and $\hat\lambda_1^{(\ell)}$, satisfying $\bar\lambda_1\geq\eta$ and $\|\lambda_1^*-\bar\lambda_1\|_\infty\leq\|\hat\lambda_1-\lambda_1^*\|_\infty$,

$$
d^2M(\bar\zeta;\Delta_\zeta)=2\mathbb{E}\left[\frac{K\left(\frac{P-\pi(X)}{h}\right)}{h\bar\lambda_2^2(X,P)}\Delta_{\lambda_2}(X,P)\Delta_{\lambda_1}(X,P)\right]+2\mathbb{E}\left[\frac{K\left(\frac{P-\pi(X)}{h}\right)\Delta_{\lambda_2}^2(X,P)}{h\bar\lambda_2^3(X,P)}\left(U(PY;\alpha^*)-\bar\lambda_1(X,P;\alpha^*)\right)\right]
$$

$$
\leq\mathcal{O}\left(\|\Delta_{\lambda_1}\|_\infty\cdot\|\Delta_{\lambda_2}\|_\infty\right)+\mathcal{O}_p(\|\Delta_{\lambda_2}\|_\infty^2\cdot\|\Delta_{\lambda_1}\|_\infty). \tag{31}
$$

Under the bandwidth condition $\sqrt{Th^{1+2s}}\to0$, $s=1,2$, and the product rate condition in Assumption 3.1, we have

$$
\mathbb{E}[\psi(V;\theta^*,\hat\zeta)-\psi(V;\theta^*,\zeta^*)]=o_p((Th)^{-1/2}). \tag{32}
$$

Combining above (29)-(32), we have $\mathbb{E}_T[\psi(V; \theta^*, \hat{\zeta})] = O_p((Th)^{-1/2})$, then substituting (28), we have

$$\|\hat{\theta} - \theta^*\| = O_p((Th)^{-1/2}). \tag{33}$$

Hence, combining (26), (27), and (33), we have

$$\sqrt{Th}\mathcal{R}_T = o_p(1). \tag{34}$$

Based on (29) and (34), multiplying (23) by $\sqrt{Th}$ yields

$$\sqrt{Th}(\hat{\theta} - \theta^*) = -J^{*-1}\left(\sqrt{Th}\mathbb{E}_T[\psi(V; \theta^*, \zeta^*)]\right) + o_p(1),$$

which implies that $\hat{\theta}$ is asymptotically linear. Furthermore, by the central limit theorem, we have

$$\sqrt{Th}(\hat{\theta} - \theta^*) \xrightarrow{d} \mathcal{N}(\mathbf{0}, \Sigma^*), \quad \Sigma^* = J^{*-1}\Omega^* J^{*-\top},$$

where the asymptotic covariance matrix $\Omega^*$ is defined as $\lim_{h \to 0} \mathrm{Var}(\sqrt{h}\psi(V; \theta^*, \zeta^*))$. Since $\mathbb{E}[\psi(V; \theta^*, \zeta^*)] = 0$, the variance is given by

$$\Omega^* = \lim_{h \to 0} \mathbb{E}\left[(\sqrt{h}\psi(V; \theta^*, \zeta^*))^2\right] = \int K^2(u)du \cdot \mathrm{diag}(\omega_0, \omega_1, 0, 0),$$

$$\omega_j = \mathbb{E}\left[\frac{\mathrm{Var}((PY)^j \exp(-PY/\alpha^*)|X, P = \pi(X))}{\pi_0(\pi(X)|X)}\right], \; j = 0, 1,$$

where we use the change of variables formula $u = \frac{p - \pi(x)}{h}$ in integration, and only $K^2(\cdot)/h^2$ is non-zero when $h \to 0$.

Now, we establish that our proposed estimator $\hat{\theta}$ achieves the semiparametric efficiency lower bound. First, we show that we achieve semiparametric efficiency for the individual functionals $W_j(\pi, \alpha^*)$, $j = 0, 1$. Following the theoretical framework in Colangelo & Lee (2026) (see Supplement S2), the target parameters $\theta^*$ are defined via the functionals $W_j^* = \mathbb{E}[(PY)^j \exp(-PY/\alpha^*)|P = \pi(X)]$, $j = 0, 1$. Note that at the fixed optimal dual parameter $\alpha^*$ (determined by the first-order condition (4)), by treating the term $(PY)^j \exp(-PY/\alpha^*)$ as a square-integrable transformation on the outcome variable, each $W_j^*$ can be viewed as an average structural function. As established in Colangelo & Lee (2026) (see Remark S5), the kernel-smoothed structure $\frac{K(\frac{p - \pi(x)}{h})}{h\lambda_2}$ is the unique local Riesz representer for the evaluation of the conditional expectation at the point $p = \pi(x)$. This implies that the diagonal elements $\omega_j$ of $\Omega^*$ defined in the expression above achieve the semiparametric efficiency lower bound for estimating $W_j^*$.

Second, we demonstrate that this efficiency propagates to the parameter vector $\theta^*$ through Hadamard differentiability. The moment function $\psi$ is constructed as the doubly robust score. Specifically, the first two components of $\psi$ (corresponding to $W_0$ and $W_1$) use the local Riesz representer structure established in Colangelo & Lee (2026), which ensures that these components achieve semiparametric efficiency. The resulting influence function of the system is given by $-J^{*-1}\psi(V; \theta^*, \zeta^*)$. From the first-order condition (4), we can explicitly express $\alpha^*$ as $\alpha^* = \frac{W_1^*}{W_0^*(-\log W_0^* - \delta)}$, and $R_\delta^* = -\alpha^* \log W_0^* - \alpha^*\delta$ from (5). The mapping from $(W_0^*, W_1^*)$ to $(\alpha^*, R_\delta^*)$ is Hadamard differentiable. Specifically, since $W_1^*$ is bounded and the functions involved (logarithm and division) are smooth in the region where $W_0^* \geq \rho/2$ (by choosing $h$ such that $h^2 \leq \frac{\rho}{2C_1 L_m}$ in (43)), the mapping is Hadamard differentiable. By Theorems 25.20, 25.21, and 25.47 of van der Vaart (1998), the efficiency of $W_0^*$ and $W_1^*$ propagates to the efficiency of each component of $\theta^*$ through the delta method. Since the influence function $-J^{*-1}\psi$ has the structure of the efficient influence function for the parameter vector $\theta^*$, the asymptotic covariance matrix $\Sigma^* = J^{*-1}\Omega^* J^{*-\top}$ achieves the semiparametric efficiency lower bound for the entire parameter vector $\theta^*$, and in particular for each component $\alpha^*$, $W_0^*$, $W_1^*$, and $R_\delta^*$. $\qquad \square$

## C. Proof of Results in Section 4

To ease the presentation, we use notations $W$, $m$ and $\hat{m}^{(\ell)}$ to represent the true and estimated functions, $W_0$, $m_0$ and $\hat{m}_0^{(\ell)}$, respectively.

**Lemma C.1.** *Let $\hat{W}_T(\alpha)$ be defined as in* (8). *The first and second derivatives of the objective function $\hat{\varphi}_T(\alpha)$ required for the Newton update are:*

$$\frac{\partial}{\partial\alpha}\hat{\varphi}_T(\alpha) = -\log\hat{W}_T(\alpha) - \alpha\frac{\hat{W}_T'(\alpha)}{\hat{W}_T(\alpha)} - \delta,$$

$$\frac{\partial^2}{\partial\alpha^2}\hat{\varphi}_T(\alpha) = \frac{\alpha(\hat{W}_T'(\alpha))^2 - 2\hat{W}_T'(\alpha)\hat{W}_T(\alpha) - \alpha\hat{W}_T''(\alpha)\hat{W}_T(\alpha)}{(\hat{W}_T(\alpha))^2},$$

*where the derivatives of the weighted generating function are given by:*

$$\hat{W}_T'(\alpha) = \frac{1}{T}\sum_{\ell=1}^{L}\sum_{t\in\mathcal{I}_\ell}\frac{K\left(\frac{p_t-\pi(x_t)}{h}\right)}{\hat{S}_\ell^\pi h\hat{\pi}_0(p_t|x_t)}\exp\left(-\frac{p_ty_t}{\alpha}\right)\frac{p_ty_t}{\alpha^2},$$

$$\hat{W}_T''(\alpha) = \frac{1}{T}\sum_{\ell=1}^{L}\sum_{t\in\mathcal{I}_\ell}\frac{K\left(\frac{p_t-\pi(x_t)}{h}\right)}{\hat{S}_\ell^\pi h\hat{\pi}_0(p_t|x_t)}\exp\left(-\frac{p_ty_t}{\alpha}\right)\left(\frac{(p_ty_t)^2}{\alpha^4} - \frac{2p_ty_t}{\alpha^3}\right).$$

*Proof of Lemma C.1.* This proof is straightforward by taking the derivatives of the function $\hat{\varphi}_T(\alpha)$ with respect to $\alpha$. $\square$

*Proof of Theorem 4.4.* We can rewrite the DR estimator as follows,

$$\hat{W}^{(\ell)}(\pi,\alpha) = \frac{1}{|\mathcal{I}_\ell|\hat{S}_\ell^\pi}\sum_{t\in\mathcal{I}_\ell}\frac{K\left(\frac{p_t-\pi(x_t)}{h}\right)}{h\hat{\pi}_0(p_t|x_t)}\exp(-p_ty_t/\alpha) + \frac{1}{|\mathcal{I}_\ell|}\sum_{t\in\mathcal{I}_\ell}\left(\hat{m}^{(\ell)}(x_t,\pi(x_t);\alpha) - \frac{1}{\hat{S}_\ell^\pi}\hat{m}^{(\ell)}(x_t,p_t;\alpha)\right).$$

We also define an infeasible optimal dual objective function $\tilde{R}_\delta^h(\pi) \triangleq \sup_{\alpha\geq0}\{-\alpha\log\tilde{W}_h(\pi,\alpha) - \alpha\delta\}$, where

$$\tilde{W}_h(\pi,\alpha) \triangleq \mathbb{E}_{\mathbf{P}_0*\pi_0}\left[m(X,\pi(X);\alpha) + \frac{K\left(\frac{P-\pi(X)}{h}\right)}{h\pi_0(P|X)}\left(\exp(-r(X,P,Z)/\alpha) - m(X,P;\alpha)\right)\right],$$

the empirical dual objective function $\tilde{R}_\delta(\pi) \triangleq \sup_{\alpha\geq0}\{-\alpha\log\tilde{W}_T(\pi,\alpha) - \alpha\delta\}$, and the infeasible estimator is

$$\tilde{W}^{(\ell)}(\pi,\alpha) \triangleq \frac{1}{|\mathcal{I}_\ell|S_\ell^\pi}\sum_{t\in\mathcal{I}_\ell}\frac{K\left(\frac{p_t-\pi(x_t)}{h}\right)}{h\pi_0(p_t|x_t)}\exp(-p_ty_t/\alpha) + \frac{1}{|\mathcal{I}_\ell|}\sum_{t\in\mathcal{I}_\ell}\left(m(x_t,\pi(x_t);\alpha) - \frac{1}{S_\ell^\pi}\frac{K\left(\frac{p_t-\pi(x_t)}{h}\right)}{h\pi_0(p_t|x_t)}m(x_t,p_t;\alpha)\right),$$

$$S_\ell^\pi \triangleq \frac{1}{|\mathcal{I}_\ell|}\sum_{t\in\mathcal{I}_\ell}\frac{K\left(\frac{p_t-\pi(x_t)}{h}\right)}{h\pi_0(p_t|x_t)}, \quad \text{and} \quad \tilde{W}_T(\pi,\alpha) \triangleq \frac{1}{L}\sum_{\ell=1}^{L}\tilde{W}^{(\ell)}(\pi,\alpha).$$

Then we can decompose regret as

$$\begin{aligned}
\text{Reg}(\hat{\pi}) &= R_\delta(\pi^*) - \hat{R}_\delta(\hat{\pi}) + \hat{R}_\delta(\hat{\pi}) - R_\delta(\hat{\pi}) \\
&\leq \left(R_\delta(\pi^*) - \hat{R}_\delta(\pi^*)\right) + \left(\hat{R}_\delta(\hat{\pi}) - R_\delta(\hat{\pi})\right) \leq 2\sup_{\pi\in\Pi}|R_\delta(\pi) - \hat{R}_\delta(\pi)| \\
&\leq 2\sup_{\pi\in\Pi}|R_\delta(\pi) - \tilde{R}_\delta^h(\pi)| + 2\sup_{\pi\in\Pi}|\tilde{R}_\delta^h(\pi) - \tilde{R}_\delta(\pi)| + 2\sup_{\pi\in\Pi}|\tilde{R}_\delta(\pi) - \hat{R}_\delta(\pi)|.
\end{aligned}$$

By utilizing Lemmas C.2, C.4 and C.5, we obtain with probability at least $1 - 7\epsilon$,

$$\text{Reg}(\hat{\pi}) \lesssim \overline{\alpha}\left(\log\left(\frac{1}{h}\right)\cdot\left(\frac{d_\Pi}{Th/L} + \sqrt{\frac{d_\Pi}{Th/L}} + \sqrt{\frac{\log(1/\epsilon)}{Th/L}} + \frac{\log(1/\epsilon)}{Th/L}\right) + T^{-\gamma_\pi-\gamma_m} + h^s\right) + h^s,$$

for $s = 1, 2$ in Assumption 2.4, which completes the proof. $\square$

**Lemma C.2.** *Under Assumptions 2.1, 2.3, 2.4, 3.2, we have* $\sup_{\pi \in \Pi} |R_\delta(\pi) - \tilde{R}^h_\delta(\pi)| \lesssim h^s$ *for* $s = 1, 2$ *in Assumption 2.4.*

*Proof of Lemma C.2.* To ease the notation, let $W(\pi, \alpha) := \mathbb{E}_{\mathbf{P}_0} \left[ \exp\left(-r(X, \pi(X), Z)/\alpha\right) \right]$. Define an alternative approach using $m(X, \pi(X); \alpha)$ instead of $m(X, P; \alpha)$,

$$\tilde{W}'_h(\pi, \alpha) \triangleq \mathbb{E}_{\mathbf{P}_0 * \pi_0} \left[ m(X, \pi(X); \alpha) + \frac{K\left(\frac{P - \pi(X)}{h}\right)}{h\pi_0(P|X)} \left(\exp(-r(X, P, Z)/\alpha) - m(X, \pi(X); \alpha)\right) \right].$$

First, we show that $\sup_{\alpha > 0} \left\{ \alpha |\tilde{W}'_h(\pi, \alpha) - W(\pi, \alpha)| \right\} \le \mathcal{O}(h)$. Recalling that the conditional expectation of the transformed reward is $m(X, P; \alpha) := \mathbb{E}_{\mathbf{P}_0} \left[\exp(-r(X, P, Z)/\alpha) \mid X, P\right]$. Then we can rewrite as

$$
\begin{aligned}
W(\pi, \alpha) &= \mathbb{E}_{\mathbf{P}_0} \left[ \exp\left(-r(X, \pi(X), Z)/\alpha\right) \right] \\
&= \mathbb{E}_X \left[ \mathbb{E}_Z[\exp\left(-r(X, \pi(X), Z)/\alpha\right)|X] \right] \\
&= \mathbb{E}_X \left[ \mathbb{E}_Z[\exp\left(-r(X, P, Z)/\alpha\right)|X, P = \pi(X)] \right] \\
&= \mathbb{E}_X[m(X, \pi(X); \alpha)] \\
&= \mathbb{E}_X \left[ \mathbb{E}_Z[\exp\left(-r(P, Z)/\alpha\right)|X, P = p] \cdot \mathbf{1}\{\pi(X) = p\} \right] \\
&= \mathbb{E}_X \left[ \int_{p_{\min}}^1 \mathbb{E}_Z[\exp\left(-r(P, Z)/\alpha\right)|X, P = p] \cdot \mathbf{1}\{\pi(X) = p\} \mathrm{d}p \right],
\end{aligned}
$$

$$
\begin{aligned}
\tilde{W}'_h(\pi, \alpha) &= \mathbb{E}_{\mathbf{P}_0} \left[ m(X, \pi(X); \alpha) + \frac{1}{h} \int_{p_{\min}}^1 K\left(\frac{p - \pi(X)}{h}\right) \left(e^{-r(X, p, Z)/\alpha} - m(X, \pi(X); \alpha)\right) \mathrm{d}p \right] \\
&= \frac{1}{h} \int_{p_{\min}}^1 \mathbb{E}_{\mathbf{P}_0} \left[ K\left(\frac{p - \pi(X)}{h}\right) e^{-r(X, p, Z)/\alpha} \right] \mathrm{d}p \quad \text{Assumption 3.2 and Fubini-Tonelli} \\
&= \frac{1}{h} \int_{p_{\min}}^1 \mathbb{E}_X \left[ K\left(\frac{p - \pi(X)}{h}\right) \mathbb{E}_{\mathbf{P}_0|X} \left[ e^{-r(X, p, Z)/\alpha} | X \right] \right] \mathrm{d}p \quad (35) \\
&= \mathbb{E}_X \left[ \int_{p_{\min}}^1 \frac{K(\frac{p - \pi(X)}{h}) m(X, p; \alpha)}{h} \mathrm{d}p \right] = \mathbb{E}_X \left[ \int_{-\infty}^\infty m(X, th + \pi(X); \alpha) K(t) dt \right].
\end{aligned}
$$

Case 1: Second-order bounded derivative holds in Assumption 2.4. Suppose that the second-order derivative of $m(X, P; \alpha)$ is bounded by $L_m$, and we delay the justification of $L_m$ later. By the second-order Taylor expansion of $m(X, P; \alpha)$ around $P = \pi(X)$, we have

$$|\tilde{W}'_h(\pi, \alpha) - W(\pi, \alpha)| = \left| \mathbb{E}_{\mathbf{P}_0} \left[ \int m(X, th + \pi(X); \alpha) - m(X, \pi(X); \alpha) K(t) dt \right] \right| \le C_1 L_m h^2 \quad (36)$$

where we use $\int u K(u) du = 0$ and $\int u^2 K(u) du \le C_1 < \infty$ from Assumption 3.2. Now, we provide the specific $L_m$. We derive the partial derivative as

$$-1 \le \frac{\partial m(X, P; \alpha)}{\partial P} = -\frac{1}{\alpha} e^{-P/\alpha} \cdot \mathbb{P}(Z \ge P - f(X)|X) + \left(1 - e^{-P/\alpha}\right) f_Z(P - f(X)) \le L_Z, \quad (37)$$

$$
\begin{aligned}
0 \le \frac{\partial^2 m}{\partial P^2} &= f'_Z(P - f(X))(1 - e^{-P/\alpha}) + \frac{2}{\alpha} e^{-P/\alpha} f_Z(P - f(X)) + \frac{1}{\alpha^2} e^{-P/\alpha} \mathbb{P}(Z \ge P - f(X)|X) \\
&\le L_Z + \frac{2}{e p_{\min}} + \frac{4}{e^2 p_{\min}^2} \quad (38)
\end{aligned}
$$

then $L_m \equiv \max\{1, L_Z + \frac{2}{e p_{\min}} + \frac{4}{e^2 p_{\min}^2}\}$, where $f_Z, f'_Z$ are pdf and pdf derivative of $Z$, and $L_Z$-bounded from Assumption 2.4. Multiplying by $\alpha$, we get

$$\sup_{\alpha > 0} \left\{ \alpha |\tilde{W}'_h(\pi, \alpha) - W(\pi, \alpha)| \right\} \le \sup_{\alpha > 0} \alpha \left| \mathbb{E} \left[ \int m(X, th + \pi(X); \alpha) - m(X, \pi(X); \alpha) K(t) dt \right] \right| \le C_1 h^2 \cdot \sup_{\alpha > 0} \left\{ \left| \alpha \frac{\partial^2 m}{\partial P^2} \right| \right\}.$$

The following step is to show that $\sup_{\alpha>0}\{\alpha\frac{\partial^2 m}{\partial P^2}\}$ is a finite constant. Note that $p \in [p_{\min}, 1]$, then

$$0 \leq \alpha\frac{\partial^2 m}{\partial P^2} = \alpha f_Z'(P - f(X))(1 - e^{-P/\alpha}) + 2e^{-P/\alpha}f_Z(P - f(X)) + \frac{1}{\alpha}e^{-P/\alpha}\mathbb{P}(Z \geq P - f(X)|X)$$

$$\leq \alpha\left(1 - e^{-1/\alpha}\right)L_Z + 2L_Z + \frac{1}{\alpha}L_Z + \frac{1}{ep_{\min}} \leq 3L_Z + \frac{1}{ep_{\min}}, \quad \forall \alpha > 0. \tag{39}$$

Hence, we conclude that $\sup_{\alpha>0}\{\alpha\frac{\partial^2 m}{\partial P^2}\} = 3L_Z + \frac{1}{ep_{\min}}$. With some abuse of notation, let $L_m := 1 + 3L_Z + \frac{2}{ep_{\min}} + \frac{4}{e^2 p_{\min}^2}$. Then, we get

$$\sup_{\alpha>0}\left\{\alpha|\tilde{W}_h'(\pi,\alpha) - W(\pi,\alpha)|\right\} \leq C_1 L_m h^2. \tag{40}$$

By a similar process, we obtain

$$\left|\tilde{W}_h(\pi,\alpha) - \tilde{W}_h'(\pi,\alpha)\right| = \left|\mathbb{E}_{\mathbf{P}_0*\pi_0}\left[\frac{K\left(\frac{P-\pi(X)}{h}\right)}{h\pi_0(P|X)}(m(X,\pi(X);\alpha) - m(X,P;\alpha))\right]\right|$$

$$\leq \left|\frac{1}{h}\mathbb{E}_X\left[\int_{p_{\min}}^1 K\left(\frac{p-\pi(X)}{h}\right)m(X,\pi(X);\alpha) - m(X,p;\alpha)\,dp\Big|X\right]\right|$$

$$= \left|\mathbb{E}_X\left[\int K(u)m(X,\pi(X);\alpha) - m(X,\pi(X) + uh;\alpha)\,du\Big|X\right]\right| \leq C_1 L_m h^2, \quad \text{and} \tag{41}$$

$$\sup_{\alpha>0}\left\{\alpha|\tilde{W}_h'(\pi,\alpha) - \tilde{W}_h(\pi,\alpha)|\right\} \leq C_1 L_m h^2. \tag{42}$$

To proceed, we can pick $h^2 \leq \frac{\rho}{2C_1 L_m}$, the (36) and (41) imply that

$$\left|\tilde{W}_h'(\pi,\alpha) - W(\pi,\alpha)\right| \leq \frac{\rho}{2}, \quad \left|\tilde{W}_h(\pi,\alpha) - \tilde{W}_h'(\pi,\alpha)\right| \leq \frac{\rho}{2}.$$

Therefore, we have

$$|R_\delta(\pi) - \tilde{R}_\delta^h(\pi)| \leq \sup_{\alpha>0}\left|(-\alpha\log\tilde{W}_h(\pi,\alpha) - \alpha\delta) - (-\alpha\log W(\pi,\alpha) - \alpha\delta)\right|$$

$$= \sup_{\alpha>0}\left|\alpha\log\left(\frac{W(\pi,\alpha)}{\tilde{W}_h(\pi,\alpha)}\right)\right| \leq \sup_{\alpha>0}\left|\alpha\log\left(\frac{W(\pi,\alpha)}{\tilde{W}_h'(\pi,\alpha)}\right)\right| + \sup_{\alpha>0}\left|\alpha\log\left(\frac{\tilde{W}_h(\pi,\alpha)}{\tilde{W}_h'(\pi,\alpha)}\right)\right|.$$

For the first term,

$$\sup_{\alpha>0}\left|\alpha\log\left(\frac{W(\pi,\alpha)}{\tilde{W}_h'(\pi,\alpha)}\right)\right| = \sup_{\alpha>0}\left|\alpha\log\left(1 + \frac{W(\pi,\alpha) - \tilde{W}_h'(\pi,\alpha)}{\tilde{W}_h'(\pi,\alpha)}\right)\right|$$

$$\overset{(a)}{\leq} 2\sup_{\alpha>0}\left\{\alpha\left|\frac{W(\pi,\alpha) - \tilde{W}_h'(\pi,\alpha)}{\tilde{W}_h'(\pi,\alpha)}\right|\right\} \overset{(b)}{\leq} \frac{2}{\rho}\sup_{\alpha>0}\left\{\alpha\left|W(\pi,\alpha) - \tilde{W}_h'(\pi,\alpha)\right|\right\} \overset{(c)}{\leq} \frac{2}{\rho}C_1 L_m h^2,$$

where step (a)-(b) follow from the inequality $|\log(1 + u)| \leq 2|u|$ for any $|u| \leq 1/2$ and $\tilde{W}_h'(\pi,\alpha) \geq \rho$, which is continued from (35),

$$\tilde{W}_h'(\pi,\alpha) = \frac{1}{h}\int_{p_{\min}}^1 \mathbb{E}_X\left[K\left(\frac{p-\pi(X)}{h}\right)\mathbb{E}_{\mathbf{P}_0|X}\left[e^{-r(p,Z)/\alpha}|X\right]\right]dp$$

$$= \frac{1}{h}\int_{p_{\min}}^1 \mathbb{E}_X\left[K\left(\frac{p-\pi(X)}{h}\right)\left(e^{-p/\alpha}\cdot\mathbf{P_0}(v(X,Z) \geq p|X) + \mathbf{P_0}(v(X,Z) < p|X)\right)\right]dp$$

$$\geq \frac{\rho}{h}\int_{p_{\min}}^1 \mathbb{E}_X\left[K\left(\frac{p-\pi(X)}{h}\right)\right]dp \quad \text{by Assumption 2.3}$$

$$= \frac{\rho}{h} \mathbb{E}_X \left[ \int_{p_{\min}}^1 K \left( \frac{p - \pi(X)}{h} \right) \mathrm{d}p \right] \geq \underline{\rho}. \tag{43}$$

And step (c) directly follows from (40). So does the second item. Combining above two terms (40) and (42), we get $|R_\delta(\pi) - \tilde{R}_\delta^h(\pi)| \leq \frac{4}{\rho} C_1 L_m h^2$.

Case 2: First-order bounded derivative holds in Assumption 2.4. By a similar process, we have $|R_\delta(\pi) - \tilde{R}_\delta^h(\pi)| \leq \frac{4}{\rho} C_1 L_m h$, where we let $L_m \equiv \max\{1, L_Z\}$ from (37). $\qquad \square$

**Lemma C.3.** *Under Assumption 2.3, we have*

$$|\tilde{R}_{\mathrm{DRO}}^h(\pi) - \tilde{R}_{\mathrm{DRO}}(\pi)| \leq \frac{4}{\underline{\rho}} \frac{1}{L} \sum_{\ell=1}^L \Big( 2 d_{KS}(F_{\mathbb{Q}^\pi}, F_{\hat{\mathbb{Q}}_\ell^\pi}) + d_{KS}(F_{\mathbf{P}_0}, F_{\hat{\mathbb{P}}_\ell}) \Big),$$

*where $d_{\mathrm{KS}}(\cdot, \cdot)$ denotes the KS distance, defined as $d_{\mathrm{KS}}(F_1, F_2) \triangleq \sup_{s \in [0,1]} |F_1(s) - F_2(s)|$, $\hat{\mathbb{P}}_\ell$ is a uniform empirical measure on $\hat{\Omega}_\ell := \{(z_t, x_t, p_t)\}_{t \in \mathcal{I}_\ell}$, $\ell = 1, ..., L$, and the kernel-weighted measures are defined as*

$$\frac{\mathrm{d}\mathbb{Q}^\pi}{\mathrm{d}(\mathbf{P}_0 * \pi_0)} \triangleq \frac{K\left( \frac{P - \pi(X)}{h} \right)}{h \pi_0(P|X)}, \quad \text{and} \quad \frac{\mathrm{d}\hat{\mathbb{Q}}_\ell^\pi}{\mathrm{d}\hat{\mathbb{P}}_\ell}(s) \triangleq \frac{1}{S_\ell^\pi} \cdot \frac{K\left( \frac{p_t - \pi(x_t)}{h} \right)}{h \pi_0(p_t|x_t)}.$$

*Proof of Lemma C.3.* For simplicity, we denote the first term in $\tilde{W}_T(\pi, \alpha)$ as

$$\tilde{V}_1(\pi, \alpha) \triangleq \frac{1}{L} \sum_{\ell=1}^L \frac{1}{|\mathcal{I}_\ell| S_\ell^\pi} \sum_{t \in \mathcal{I}_\ell} \frac{K\left( \frac{p_t - \pi(x_t)}{h} \right)}{h \pi_0(p_t|x_t)} \exp(-p_t y_t / \alpha)$$

and the second term as

$$\tilde{V}_2(\pi, \alpha) \triangleq \frac{1}{L} \sum_{\ell=1}^L \frac{1}{|\mathcal{I}_\ell|} \sum_{t \in \mathcal{I}_\ell} \left( m(x_t, \pi(x_t); \alpha) - \frac{1}{S_\ell^\pi} \frac{K(\frac{p_t - \pi(x_t)}{h})}{h \pi_0(p_t|x_t)} m(x_t, p_t; \alpha) \right),$$

so that $\tilde{W}_T(\pi, \alpha) \equiv \tilde{V}_1(\pi, \alpha) + \tilde{V}_2(\pi, \alpha)$. Similarly, we write

$$\tilde{V}_1^h(\pi, \alpha) \triangleq \mathbb{E}_{\mathbf{P}_0 * \pi_0} \left[ \frac{K(\frac{P - \pi(X)}{h})}{h \pi_0(P|X)} \exp(-P \cdot Y / \alpha) \right], \quad \tilde{V}_2^h(\pi, \alpha) \triangleq \mathbb{E}_{\mathbf{P}_0 * \pi_0} \left[ m(X, \pi(X); \alpha) - \frac{K(\frac{P - \pi(X)}{h})}{h \pi_0(P|X)} m(X, P; \alpha) \right],$$

so that $\tilde{W}_h(\pi, \alpha) \equiv \tilde{V}_1^h(\pi, \alpha) + \tilde{V}_2^h(\pi, \alpha)$.

We decompose $\tilde{W}_T(\pi, \alpha)$ and $\tilde{W}_h(\pi, \alpha)$ into three distinct components based on the following measures. Denote the CDF of $r(X, P, Z)$ under the measure $\mathbb{Q}^\pi$ as $F_{\mathbb{Q}^\pi}(s)$,

$$F_{\mathbb{Q}^\pi}(s) \triangleq \mathbb{Q}^\pi(r(X, P, Z) \leq s) = \mathbb{E}_{\mathbb{Q}^\pi}[\mathbf{1}\{r(X, P, Z) \leq s\}] = \mathbb{E}_{\mathbf{P}_0 * \pi_0} \left[ \mathbf{1}\{r(X, P, Z) \leq s\} \cdot \frac{K\left( \frac{P - \pi(X)}{h} \right)}{h \pi_0(P|X)} \right].$$

Hence, we can calculate the expectation of the function $g(X, Z, P) = \exp(-r(X, P, Z) / \alpha)$ with respect to $\mathbb{Q}^\pi$ as follows:

$$\tilde{V}_1^h(\pi, \alpha) = \mathbb{E}_{\mathbb{Q}^\pi}[\exp(-r(X, P, Z) / \alpha)].$$

Similarly, denote the CDF of $r(X, P, Z)$ under the measure $\hat{\mathbb{Q}}_\ell^\pi$ as $F_{\hat{\mathbb{Q}}_\ell^\pi}(s)$,

$$F_{\hat{\mathbb{Q}}_\ell^\pi}(s) \triangleq \hat{\mathbb{Q}}_\ell^\pi(r \leq s) = \mathbb{E}_{\hat{\mathbb{Q}}_\ell^\pi}[\mathbf{1}\{r \leq s\}] = \mathbb{E}_{\hat{\mathbb{P}}_\ell} \left[ \mathbf{1}\{r \leq s\} \cdot \frac{\mathrm{d}\hat{\mathbb{Q}}_\ell^\pi}{\mathrm{d}\hat{\mathbb{P}}_\ell}(t) \right] = \frac{1}{|\mathcal{I}_\ell| S_\ell^\pi} \sum_{t \in \mathcal{I}_\ell} \frac{K\left( \frac{p_t - \pi(x_t)}{h} \right)}{h \pi_0(p_t|x_t)} \mathbf{1}\{r_t \leq s\},$$

which implies that

$$
\mathbb{E}_{\hat{\mathbb{Q}}_\ell^\pi}[\exp(-r/\alpha)] = \mathbb{E}_{\hat{\mathbb{P}}_\ell}\left[\exp(-r/\alpha)\frac{\mathrm{d}\hat{\mathbb{Q}}_\ell^\pi}{\mathrm{d}\hat{\mathbb{P}}_\ell}(t)\right] = \frac{1}{|\mathcal{I}_\ell|S_\ell^\pi}\sum_{t\in\mathcal{I}_\ell}\frac{K\left(\frac{p_t-\pi(x_t)}{h}\right)}{h\pi_0(p_t|x_t)}\exp(-r_t/\alpha),\ \tilde{V}_1(\pi,\alpha) = \frac{1}{L}\sum_{\ell=1}^L\mathbb{E}_{\hat{\mathbb{Q}}_\ell^\pi}[\exp(-r/\alpha)].
$$

Similarly, using the empirical measures $\hat{\mathbb{P}}_\ell$ and $\hat{\mathbb{Q}}_\ell^\pi$ defined above, we can rewrite

$$
\tilde{V}_2(\pi,\alpha) = \frac{1}{L}\sum_{\ell=1}^L\left(\mathbb{E}_{\hat{\mathbb{P}}_\ell}[m(X,\pi(X);\alpha)] - \mathbb{E}_{\hat{\mathbb{Q}}_\ell^\pi}[m(X,P;\alpha)]\right),\quad \tilde{V}_2^h(\pi,\alpha) = \mathbb{E}_{\mathbf{P}_0}[m(X,\pi(X);\alpha)] - \mathbb{E}_{\mathbb{Q}^\pi}[m(X,P;\alpha)].
$$

Picking $h^s \leq \frac{\rho}{2C_1 L_m}$ from (41), we have $\left|\tilde{W}_h(\pi,\alpha) - \tilde{W}_h'(\pi,\alpha)\right| \leq \underline{\rho}/2$, and from (43), it follows from $\tilde{W}_h(\pi,\alpha) \geq \tilde{W}_h'(\pi,\alpha) - \underline{\rho}/2 \geq \underline{\rho}/2$. Furthermore, by a similar process in (65), we can also get

$$
\left|\tilde{W}_T(\pi,\alpha) - \tilde{W}_h(\pi,\alpha)\right| = \mathcal{O}\left(\frac{d_\Pi\log(1/\epsilon)}{Th}\right),
$$

with the probability at least $1-\epsilon$. To proceed, we can pick $\Omega\left(\frac{d_\Pi\log(1/\epsilon)}{T\underline{\rho}}\right) \leq h \leq \left(\frac{\rho}{2C_1 L_m}\right)^{1/s}$ such that $\left|\tilde{W}_T(\pi,\alpha) - \tilde{W}_h(\pi,\alpha)\right| \leq \underline{\rho}/4$. By a similar argument in proof of Lemma C.2, we can partition $\alpha\left|\tilde{W}_T(\pi,\alpha) - \tilde{W}_h(\pi,\alpha)\right|$ into three terms.

$$
\begin{aligned}
|\tilde{R}_{\mathrm{DRO}}^h(\pi) - \tilde{R}_{\mathrm{DRO}}(\pi)| &= \left|\sup_{\alpha>0}\left\{-\alpha\log\tilde{W}_h(\pi,\alpha) - \alpha\delta\right\} - \sup_{\alpha>0}\left\{-\alpha\log\tilde{W}_T(\pi,\alpha) - \alpha\delta\right\}\right| \\
&\leq \sup_{\alpha>0}\left|\alpha\log\left(\frac{\tilde{W}_T(\pi,\alpha)}{\tilde{W}_h(\pi,\alpha)}\right)\right| \leq \frac{4}{\underline{\rho}}\sup_{\alpha>0}\left\{\alpha\left|\tilde{W}_T(\pi,\alpha) - \tilde{W}_h(\pi,\alpha)\right|\right\} \\
&\leq \frac{4}{\underline{\rho}}\sup_{\alpha>0}\left\{\alpha\left|\frac{1}{L}\sum_{\ell=1}^L(\mathbb{E}_{\hat{\mathbb{Q}}_T^\pi} - \mathbb{E}_{\mathbb{Q}^\pi})\left[\exp(-r(X,P,Z)/\alpha)\right]\right|\right\} + \frac{4}{\underline{\rho}}\sup_{\alpha>0}\left\{\alpha\left|\frac{1}{L}\sum_{\ell=1}^L(\mathbb{E}_{\hat{\mathbb{Q}}_\ell^\pi} - \mathbb{E}_{\mathbb{Q}^\pi})\left[m(X,P;\alpha)\right]\right|\right\} \\
&+ \frac{4}{\underline{\rho}}\sup_{\alpha>0}\left\{\alpha\left|\frac{1}{L}\sum_{\ell=1}^L(\mathbb{E}_{\hat{\mathbb{P}}_\ell} - \mathbb{E}_{\mathbf{P}_0})[m(X,\pi(X);\alpha)]\right|\right\}.
\end{aligned}
\tag{44}
$$

Note that for any non-negative random variable $S \in [0,1]$ with CDF $F_S$, and a differentiable function $g(s)$, we have $\mathbb{E}[g(S)] = g(0) + \int_0^1 g'(s)(1 - F_S(s))ds$. In our setting, $g(s) = \exp(-s/\alpha)$. Using the definition of the Kolmogorov-Smirnov distance $d_{KS}(F_1,F_2) = \sup_s|F_1(s) - F_2(s)|$, we bound the first term $\alpha(\mathbb{E}_{\hat{\mathbb{Q}}_\ell^\pi} - \mathbb{E}_{\mathbb{Q}^\pi})\left[\exp(-r(P,Z)/\alpha)\right]$.

$$
\begin{aligned}
\left|\alpha(\mathbb{E}_{\hat{\mathbb{Q}}_\ell^\pi} - \mathbb{E}_{\mathbb{Q}^\pi})[e^{-r/\alpha}]\right| &= \left|\alpha\int_0^1\left(-\frac{1}{\alpha}e^{-s/\alpha}\right)(F_{\mathbb{Q}^\pi}(s) - F_{\hat{\mathbb{Q}}_\ell^\pi}(s))ds\right| \\
&\leq d_{KS}(F_{\mathbb{Q}^\pi},F_{\hat{\mathbb{Q}}_\ell^\pi})\cdot\int_0^1 e^{-s/\alpha}ds = d_{KS}(F_{\mathbb{Q}^\pi},F_{\hat{\mathbb{Q}}_\ell^\pi})\cdot\alpha(1-e^{-1/\alpha}).
\end{aligned}
$$

Taking the limit by L'Hôpital, $\lim_{\alpha\to\infty}\frac{1-e^{-1/\alpha}}{1/\alpha} = 1$. Thus, we have $\sup_{\alpha>0}\left\{\alpha\left(1-e^{-1/\alpha}\right)\right\} = 1$ from the monotone increasing. Therefore,

$$
\sup_{\alpha>0}\left|\alpha(\mathbb{E}_{\hat{\mathbb{Q}}_\ell^\pi} - \mathbb{E}_{\mathbb{Q}^\pi})[e^{-r/\alpha}]\right| \leq d_{KS}(F_{\mathbb{Q}^\pi},F_{\hat{\mathbb{Q}}_\ell^\pi}).
$$

Applying the same logic to $\alpha(\mathbb{E}_{\hat{\mathbb{Q}}_\ell^\pi} - \mathbb{E}_{\mathbb{Q}^\pi})\left[m(X,P;\alpha)\right]$,

$$
\begin{aligned}
\left|\alpha(\mathbb{E}_{\hat{\mathbb{Q}}_\ell^\pi} - \mathbb{E}_{\mathbb{Q}^\pi})[m(X,P;\alpha)]\right| &= \left|\alpha\int_0^1\left(-\frac{1}{\alpha}e^{-s/\alpha}\right)(F_{\mathbb{Q}^\pi}(s) - F_{\hat{\mathbb{Q}}_\ell^\pi}(s))ds\right| \\
&\leq d_{KS}(F_{\mathbb{Q}^\pi},F_{\hat{\mathbb{Q}}_\ell^\pi})\cdot\int_0^1 e^{-s/\alpha}ds = d_{KS}(F_{\mathbb{Q}^\pi},F_{\hat{\mathbb{Q}}_\ell^\pi})\cdot\alpha(1-e^{-1/\alpha}).
\end{aligned}
$$

Taking the average over $\ell = 1, \ldots, L$, we have

$$|\tilde{R}^h_{\mathrm{DRO}}(\pi) - \tilde{R}_{\mathrm{DRO}}(\pi)| \le \frac{4}{\underline{\rho}} \frac{1}{L} \sum_{\ell=1}^{L} \left( 2d_{KS}(F_{\mathbb{Q}^\pi}, F_{\hat{\mathbb{Q}}^\pi_\ell}) + d_{KS}(F_{\mathbf{P}_0}, F_{\hat{\mathbb{P}}_\ell}) \right).$$

$\square$

**Lemma C.4.** *Under Assumptions 2.2 and 3.2, we have with probability at least $1 - 3\epsilon$,*

$$\sup_{\pi \in \Pi} |\tilde{R}^h_{\mathrm{DRO}}(\pi) - \tilde{R}_{\mathrm{DRO}}(\pi)| \lesssim \frac{d_\Pi L}{Th} + \sqrt{\frac{d_\Pi L}{Th}} + \sqrt{\frac{\log(1/\epsilon)L}{Th}} + \frac{\log(1/\epsilon)L}{Th},$$

*Proof of Lemma C.4.* We decompose the KS distance in Lemma C.3 using the triangle inequality,

$$d_{KS}(F_{\mathbb{Q}^\pi}, F_{\hat{\mathbb{Q}}^\pi_\ell}) = \sup_{s \in [0,1]} \left| \frac{1}{TS^\pi_\ell} \sum_{t \in \mathcal{I}_\ell} \frac{K\left(\frac{p_t - \pi(x_t)}{h}\right)}{h\pi_0(p_t|x_t)} \mathbf{1}\{r_t \le s\} - \mathbb{E}\left[ \frac{K\left(\frac{P - \pi(X)}{h}\right)}{h\pi_0(P|X)} \mathbf{1}\{r(X,P;\alpha) \le s\} \right] \right|$$

$$\le \sup_{s \in [0,1]} \left| \frac{1}{|\mathcal{I}_\ell|} \sum_{t \in \mathcal{I}_\ell} \frac{K\left(\frac{p_t - \pi(x_t)}{h}\right)}{h\pi_0(p_t|x_t)} \mathbf{1}\{r_t \le s\} - \mathbb{E}\left[ \frac{K\left(\frac{P - \pi(X)}{h}\right)}{h\pi_0(P|X)} \mathbf{1}\{r(X,P;\alpha) \le s\} \right] \right|$$

$$+ \sup_{s \in [0,1]} \left| \frac{S^\pi_\ell - 1}{|\mathcal{I}_\ell| S^\pi_\ell} \sum_{t \in \mathcal{I}_\ell} \frac{K\left(\frac{p_t - \pi(x_t)}{h}\right)}{h\pi_0(p_t|x_t)} \mathbf{1}\{r_t \le s\} \right|, \tag{45}$$

$$d_{KS}(F_{\mathbf{P}_0}, F_{\hat{\mathbb{P}}_\ell}) = \sup_{s \in [0,1]} \left| \frac{1}{|\mathcal{I}_\ell|} \sum_{t \in \mathcal{I}_\ell} \mathbf{1}\{r(x_t, \pi(x_t); \alpha) \le s\} - \mathbb{E}\left[\mathbf{1}\{r(X, \pi(X); \alpha) \le s\}\right] \right|. \tag{46}$$

Note that

$$\left| \frac{S^\pi_\ell - 1}{|\mathcal{I}_\ell| S^\pi_\ell} \sum_{t \in \mathcal{I}_\ell} \frac{K\left(\frac{p_t - \pi(x_t)}{h}\right)}{h\pi_0(p_t|x_t)} \mathbf{1}\{r_t \le s\} \right| \le |S^\pi_\ell - 1| \text{ and } \mathbb{E}[S^\pi_\ell] = 1. \tag{47}$$

Define

$$f_\pi(x,p) = \frac{1}{h\pi_0(p|x)} K\left(\frac{p - \pi(x)}{h}\right), \quad f^1_{\pi,s}(x,p,z) \triangleq \frac{1}{h\pi_0(p|x)} K\left(\frac{p - \pi(x)}{h}\right) \mathbf{1}\{r(x,p,z) \le s\}$$

$$f^2_{\pi,s}(x,z) \triangleq \mathbf{1}\{r(x, \pi(x), z) \le s\},$$

and the corresponding function classes

$$\mathcal{F}_\Pi := \{f_\pi | \pi \in \Pi\}, \quad \mathcal{G}_\Pi := \{g_\pi = f_\pi - \mathbb{E}[f_\pi] \big| \pi \in \Pi\},$$

$$\mathcal{F}^1_{\Pi,s} \triangleq \left\{f^1_{\pi,s} | \pi \in \Pi, s \in [0,1]\right\}, \quad \mathcal{G}^1_{\Pi,s} \triangleq \left\{g^1_{\pi,s} = f^1_{\pi,s} - \mathbb{E}[f^1_{\pi,s}] \big| \pi \in \Pi, s \in [0,1]\right\},$$

$$\mathcal{F}^2_{\Pi,s} \triangleq \left\{f^2_{\pi,s}(x,z) = f^2_{\pi,s} - \mathbb{E}[f^2_{\pi,s}] \big| \pi \in \Pi, s \in [0,1]\right\}, \quad \mathcal{G}^2_{\Pi,s} \triangleq \left\{g^2_{\pi,s} = f^2_{\pi,s} - \mathbb{E}[f^2_{\pi,s}] \big| \pi \in \Pi, s \in [0,1]\right\}.$$

Thus, we next provide the upper bounds for

$$\sup_{\pi \in \Pi} \left| \frac{1}{|\mathcal{I}_\ell|} \sum_{t \in \mathcal{I}_\ell} f_\pi(x_t, p_t) - \mathbb{E}[f_\pi] \right|, \quad \sup_{\pi \in \Pi, s \in [0,1]} \left| \frac{1}{|\mathcal{I}_\ell|} \sum_{t \in \mathcal{I}_\ell} f^1_{\pi,s}(x_t, p_t, z_t) - \mathbb{E}[f^1_{\pi,s}] \right|, \quad \sup_{\pi \in \Pi, s \in [0,1]} \left| \frac{1}{|\mathcal{I}_\ell|} \sum_{t \in \mathcal{I}_\ell} f^2_{\pi,s}(x_t) - \mathbb{E}[f^2_{\pi,s}] \right|.$$

**Step 1. Bound for** $\sup_{\pi \in \Pi} \left| \frac{1}{|\mathcal{I}_\ell|} \sum_{t \in \mathcal{I}_\ell} f_\pi(x_t, p_t) - \mathbb{E}[f_\pi] \right|$. Note that $f_\pi$ is uniformly bounded by a constant $B_\Pi := \frac{C_2}{h\eta}$ due to Assumption 3.2 and 2.2. Let $\{(x_t, p_t)\}_{t \in \mathcal{I}_\ell}$ be i.i.d. random variables. Our goal is to derive a high-probability upper bound for

$$\sup_{\pi \in \Pi} \left| \frac{1}{|\mathcal{I}_\ell|} \sum_{t \in \mathcal{I}_\ell} f_\pi(x_t, p_t) - \mathbb{E}[f_\pi] \right| = \frac{1}{|\mathcal{I}_\ell|} \sup_{g \in \mathcal{G}_\Pi} \left| \sum_{t \in \mathcal{I}_\ell} g(x_t, p_t) \right| \equiv \Delta. \tag{48}$$

First, we provide the second-moment bound.

$$\sup_{\pi \in \Pi} \mathbb{E}[f_\pi^2] = \sup_{\pi \in \Pi} \int_{\mathcal{X}} \int_{\mathcal{P}} \frac{1}{h^2 \pi_0(p|x)^2} K^2 \left( \frac{p - \pi(x)}{h} \right) \pi_0(p|x) dp dQ(x)$$

$$= \sup_{\pi \in \Pi} \int_{\mathcal{X}} \int_{\mathcal{P}} \frac{1}{h^2 \pi_0(p|x)} K^2 \left( \frac{p - \pi(x)}{h} \right) dp dQ(x) \leq \sup_{\pi \in \Pi} \frac{1}{\eta h} \int_{\mathcal{X}} \int_{-\infty}^{\infty} K^2(u) du \leq \frac{C_2}{h\eta} := \sigma^2 \qquad (49)$$

where the last inequality follows from $\int K^2(u) du < C_2$ in Assumption 3.2 and $\sigma^2 := \frac{C_2}{h\eta}$.

From the definition of covering number and uniform covering number (see Definition D.2), and Lemma D.4, we have

$$\mathcal{N}(\epsilon, \mathcal{F}_\Pi, L_2(P_{T'})) \leq \mathcal{N}_1 \left( \frac{\epsilon^2}{B_\Pi}, \mathcal{F}_\Pi, T' \right) \leq \mathcal{N}_1 \left( \frac{\epsilon^2}{B_\Pi L_K}, \Pi, T' \right),$$

which is directly by letting

$$\mathcal{F}_0 = \Pi, \ \mathcal{F}_1 = \left\{ (x, p) \mapsto \frac{p - \pi(x)}{h} | \pi \in \Pi \right\}, \ \mathcal{F}_\Pi = \left\{ f_\pi : (x, p) \mapsto \frac{K \left( \frac{p - \pi(x)}{h} \right)}{h \pi_0(p|x)} | \pi \in \Pi \right\} \qquad (50)$$

and $\text{Pdim}(\mathcal{F}_0) = \text{Pdim}(\Pi) := d_\Pi$. Combined with Lemma D.3(ii), we conclude that

$$\log \mathcal{N}(\epsilon, \mathcal{F}_\Pi, L_1(P_{T'})) \leq \log \mathcal{N}(\epsilon, \mathcal{F}_\Pi, L_2(P_{T'})) \lesssim d_\Pi \log \left( \frac{B_\Pi}{\epsilon} \right).$$

On the other hand, from Lemma D.3(iii), we have

$$\log \mathcal{M}(\epsilon, \mathcal{F}_\Pi, L_1(P_{T'})) \gtrsim \text{Pdim}(\mathcal{F}_\Pi) \log(1/\epsilon),$$

for $T' \geq \text{Pdim}(\mathcal{F}_\Pi)$. Combining the above two, which both hold for all sufficiently small $\epsilon > 0$, we can get

$$\text{Pdim}(\mathcal{F}_\Pi) \lesssim d_\Pi. \qquad (51)$$

By setting $T' = |\mathcal{I}_\ell|$ in Lemma C.6, we obtain

$$\mathbb{E}[\Delta] \equiv \frac{1}{|\mathcal{I}_\ell|} \mathbb{E} \left[ \sup_{g \in \mathcal{G}_\Pi} \left| \sum_{t \in \mathcal{I}_\ell} g(x_t, p_t) \right| \right] \lesssim \frac{B_\Pi d_\Pi}{|\mathcal{I}_\ell|} + \sqrt{\frac{\sigma^2 d_\Pi}{|\mathcal{I}_\ell|}}.$$

Now, we use Talagrand Inequality (see Lemma D.1) with deviation $\beta(\epsilon) > 0$,

$$\mathbb{P}(\Delta - \mathbb{E}[\Delta] \geq \beta(\epsilon)) \leq 2 \exp \left\{ -\frac{\beta(\epsilon)^2 |\mathcal{I}_\ell|}{8e\sigma^2 + 16eB_\Pi \mathbb{E}[\Delta] + 4B_\Pi \beta(\epsilon)} \right\}.$$

To get a bound holding with probability $1 - \epsilon$, we can pick $\beta(\epsilon) \lesssim \sqrt{\frac{\sigma^2 \log(1/\epsilon)}{|\mathcal{I}_\ell|}} + \frac{B_\Pi \log(1/\epsilon)}{|\mathcal{I}_\ell|}$. Hence, we have

$$\sup_{\pi \in \Pi} \left| \frac{1}{|\mathcal{I}_\ell|} \sum_{t \in \mathcal{I}_\ell} f_\pi(x_t, p_t) - \mathbb{E}[f_\pi] \right| \lesssim \frac{d_\Pi}{|\mathcal{I}_\ell|h} + \sqrt{\frac{d_\Pi}{|\mathcal{I}_\ell|h}} + \sqrt{\frac{\log(1/\epsilon)}{|\mathcal{I}_\ell|h}} + \frac{\log(1/\epsilon)}{|\mathcal{I}_\ell|h},$$

with probability at least $1 - \epsilon$.

**Step 2. Bound for** $\sup_{\pi \in \Pi, s \in [0,1]} \left| \frac{1}{|\mathcal{I}_\ell|} \sum_{t \in \mathcal{I}_\ell} f_{\pi,s}^1(x_t, p_t, z_t) - \mathbb{E}[f_{\pi,s}^1] \right|$. It is same as Step 1 that $\left| f_{\pi,s}^1 \right| \leq B_\Pi$, and the second-moment bound is bounded by $\sigma^2 := \frac{C_2}{h\eta}$. We first claim that

$$\mathcal{N}(\epsilon, \mathcal{F}_{\Pi,s}^1, L_2(P_{|\mathcal{I}_\ell|})) \leq \mathcal{N}_2(\epsilon/2, \mathcal{F}_\Pi, |\mathcal{I}_\ell|) \sup_{\mathbf{P}} \mathcal{N}(\eta\epsilon/(2C_2), \mathscr{E}, \| \cdot \|_{\mathbf{P}}),$$

where $\mathscr{E} = \{i_s(\epsilon) \triangleq \mathbf{1}\{\epsilon \le s\} | s \in [0,1]\}$. For ease of notation, let

$$N_{2,\mathcal{F}}(\epsilon) := \mathcal{N}_2(\epsilon/2, \mathcal{F}_\Pi, |\mathcal{I}_\ell|), \quad N_{\mathscr{E}}(\epsilon) := \sup_{\mathbf{P}} \mathcal{N}(\eta\epsilon/(2C_2), \mathscr{E}, \|\cdot\|_{\mathbf{P}}).$$

Suppose $\{f_{\pi_1}, ..., f_{\pi_{N_{2,\mathcal{F}}(\epsilon)}}\}$ is a $\epsilon/2$-cover for $\mathcal{F}_\Pi$, and $\{\mathbf{1}\{\epsilon \le s_1\}, ..., \mathbf{1}\{\epsilon \le s_{N_{\mathscr{E}}(\epsilon)}\}\}$ is a $\epsilon/2$-cover for $\mathscr{E}$ under the distance $\hat{\mathbb{P}}_\ell$, defined as

$$\hat{\mathbb{P}}_\ell \triangleq \frac{1}{|\mathcal{I}_\ell|} \sum_{t \in \mathcal{I}_\ell} \Delta(r(x_t, p_t, z_t)).$$

Then, we can prove that $\mathcal{F}_{\Pi,s}^{1,\epsilon}$ is a $\epsilon$-cover set for $\mathcal{F}_{\Pi,s}^1$, where $\mathcal{F}_{\Pi,s}^{1,\epsilon}$ is defined as

$$\mathcal{F}_{\Pi,s}^{1,\epsilon} \triangleq \left\{ f_{\pi_i,s_j}^1 = \frac{K\left(\frac{p-\pi_i(x)}{h}\right)}{h\pi_0(p|x)} \mathbf{1}\{r(x,p,z) \le s_j\} \, \middle| i \le N_{2,\mathcal{F}}(\epsilon), j \le N_{\mathscr{E}}(\epsilon) \right\}.$$

For $f_{\pi,s}^1(x,p,z) \in \mathcal{F}_{\Pi,s}^1$, we can pick $f_{\tilde{\pi},\tilde{s}}^1(x,p,z) \in \mathcal{F}_{\Pi,s}^{1,\epsilon}$,

$$\|f_\pi^1 - f_{\tilde{\pi}}^1\|_{L_2(P_{|\mathcal{I}_\ell|})} \le \epsilon/2 \quad \text{and} \quad \|\mathbf{1}\{r \le s\} - \mathbf{1}\{r \le \tilde{s}\}\|_{\hat{\mathbf{P}}_\ell} \le \eta\epsilon/(2C_2).$$

Then, we have

$$\|f_{\pi,s}^1 - f_{\tilde{\pi},\tilde{s}}^1\|_{L_2(P_{|\mathcal{I}_\ell|})} = \sqrt{\frac{1}{|\mathcal{I}_\ell|} \sum_{t \in \mathcal{I}_\ell} \left( f_{\pi,s}^1(x_t, p_t, z_t) - f_{\tilde{\pi},\tilde{s}}^1(x_t, p_t, z_t) \right)^2}$$

$$= \sqrt{\frac{1}{|\mathcal{I}_\ell|} \sum_{t \in \mathcal{I}_\ell} (f_\pi^1(x_t, p_t, z_t) \mathbf{1}\{r(x_t, p_t, z_t) \le s\} - f_{\tilde{\pi}}^1(x_t, p_t, z_t) \mathbf{1}\{r(x_t, p_t, z_t) \le \tilde{s}\})^2}$$

$$= \sqrt{\frac{1}{|\mathcal{I}_\ell|} \sum_{t \in \mathcal{I}_\ell} (f_{\pi,s}^1(x_t, p_t, z_t))^2 \left( \mathbf{1}\{r(x_t, p_t, z_t) \le s\} - \mathbf{1}\{r(x_t, p_t, z_t) \le \tilde{s}\} \right)^2 + \mathbf{1}\{r(x_t, p_t, z_t) \le \tilde{s}\} (f_\pi^1(x_t, p_t, z_t) - f_{\tilde{\pi}}^1(x_t, p_t, z_t))^2}$$

$$\le \frac{C_2}{\eta} \sqrt{\frac{1}{|\mathcal{I}_\ell|} \sum_{t \in \mathcal{I}_\ell} \left( \mathbf{1}\{r(x_t, p_t, z_t) \le s\} - \mathbf{1}\{r(x_t, p_t, z_t) \le \tilde{s}\} \right)^2} + \sqrt{\frac{1}{|\mathcal{I}_\ell|} \sum_{t \in \mathcal{I}_\ell} (f_\pi^1(x_t, p_t, z_t) - f_{\tilde{\pi}}^1(x_t, p_t, z_t))^2}$$

$$\le \epsilon$$

From Lemma 19.15 and Example 19.16 in van der Vaart (1998), we have $\sup_{\mathbf{P}} \mathcal{N}(\epsilon, \mathscr{E}, \|\cdot\|_{\mathbf{P}}) \lesssim \left(\frac{1}{\epsilon}\right)^2$. Hence, we obtain

$$\mathcal{N}(\epsilon, \mathcal{F}_{\Pi,s}^1, L_2(P_{|\mathcal{I}_\ell|})) \lesssim \mathcal{N}_1\left(\frac{\epsilon^2}{4B(\mathcal{F}_\Pi)}, \mathcal{F}_\Pi, |\mathcal{I}_\ell|\right) \cdot \left(\frac{1}{\epsilon}\right)^2, \tag{52}$$

which is directly from Lemma D.3(v), and $B(\mathcal{F}_\Pi)$ is the uniform maximum of $\mathcal{F}_\Pi$. Combined with Lemma D.3(ii), we have

$$\log \mathcal{N}(\epsilon, \mathcal{F}_{\Pi,s}^1, L_2(P_{|\mathcal{I}_\ell|})) \lesssim \text{Pdim}(\mathcal{F}_\Pi) \log\left(\frac{1}{\epsilon}\right).$$

On the other hand, from Lemma D.3(iii), we have

$$\log \mathcal{M}_1(\epsilon, \mathcal{F}_{\Pi,s}^1, |\mathcal{I}_\ell|) \gtrsim \text{Pdim}(\mathcal{F}_{\Pi,s}^1) \log(1/\epsilon),$$

for $\mathcal{I}_\ell \ge \text{Pdim}(\mathcal{F}_{\Pi,s}^1)$. Combining the above two inequalities, which both hold for all sufficiently small $\epsilon > 0$, we can get

$$\text{Pdim}(\mathcal{F}_{\Pi,s}^1) \le K \cdot \text{Pdim}(\mathcal{F}_\Pi) \tag{53}$$

for some positive constant $K$. Note that,

$$\mathcal{N}(\epsilon, \mathcal{G}_{\Pi,s}^1, L_2(P_{|\mathcal{I}_\ell|})) \le \mathcal{N}(\epsilon/2, \mathcal{F}_{\Pi,s}^1, L_2(P_{|\mathcal{I}_\ell|})) + N(\epsilon/2, \overline{\mathcal{F}}_{\Pi,s}^1, L_2(P_{|\mathcal{I}_\ell|})),$$

where
$$\overline{\mathcal{F}}^1_{\Pi,s} \triangleq \left\{ \overline{f}^1_{\pi,s} = \mathbb{E}_{\mathbf{P}_0 * \pi_0}[f^1_{\pi,s}(X, P, Z)] \,\big|\, \pi \in \Pi, s \in [0,1] \right\},$$

is a function class of constant functions, satisfying
$$\|\overline{f}^1_{\pi_1,s_1} - \overline{f}^1_{\pi_2,s_2}\|_{L_2(P_{|\mathcal{I}_\ell|})} = |\overline{f}^1_{\pi_1,s_1} - \overline{f}^1_{\pi_2,s_2}| \le \mathbb{E}_{\mathbf{P}_0 * \pi_0} \left| f^1_{\pi_1,s_1}(X, P, Z) - f^1_{\pi_2,s_2}(X, P, Z) \right|.$$

Consequently, we have
$$\mathcal{N}(\epsilon/2, \overline{\mathcal{F}}^1_{\Pi,s}, L_2(P_{|\mathcal{I}_\ell|})) \le \mathcal{N}(\epsilon/2, \mathcal{F}^1_{\Pi,s}, L_1(\mathbf{P}_0 * \pi_0))$$

Using Definition D.2 and Lemma D.3(iv), we have
$$\log \mathcal{N}(\epsilon/2, \mathcal{F}^1_{\Pi,s}, L_1(\mathbf{P}_0 * \pi_0)) \le \log \mathcal{M}(\epsilon/2, \mathcal{F}^1_{\Pi,s}, L_1(\mathbf{P}_0 * \pi_0))$$
$$\lesssim \text{Pdim}(\mathcal{F}^1_{\Pi,s}) \log \left( \frac{1}{\epsilon} \log \left( \frac{1}{\epsilon} \right) \right) \lesssim \text{Pdim}(\mathcal{F}_\Pi) \log \left( \frac{1}{\epsilon} \log \left( \frac{1}{\epsilon} \right) \right) \qquad \text{by (53)}. \quad (54)$$

Therefore, we conclude that
$$\log \mathcal{N}(\epsilon, \mathcal{G}^1_{\Pi,s}, L_2(P_{|\mathcal{I}_\ell|})) \le 2 \log \mathcal{N}(\epsilon/2, \mathcal{F}^1_{\Pi,s}, L_2(P_{|\mathcal{I}_\ell|})) + 2 \log N(\epsilon/2, \overline{\mathcal{F}}^1_{\Pi,s}, L_2(P_{|\mathcal{I}_\ell|}))$$
$$\lesssim \text{Pdim}(\mathcal{F}_\Pi) \log \left( \frac{1}{\epsilon} \right) \lesssim \text{Pdim}(\Pi) \log \left( \frac{1}{\epsilon} \right) \qquad \text{by (51)}.$$

Hence, we have an upper bound like (64), implying a Dudley's integral and a Rademacher complexity bound like the proof of Lemma C.6. That is,
$$\sup_{\pi \in \Pi, s \in [0,1]} \left| \frac{1}{|\mathcal{I}_\ell|} \sum_{t \in \mathcal{I}_\ell} f^1_{\pi,s}(x_t, p_t, z_t) - \mathbb{E}[f^1_{\pi,s}] \right| \lesssim \frac{d_\Pi}{|\mathcal{I}_\ell| h} + \sqrt{\frac{d_\Pi}{|\mathcal{I}_\ell| h}} + \sqrt{\frac{\log(1/\epsilon)}{|\mathcal{I}_\ell| h}} + \frac{\log(1/\epsilon)}{|\mathcal{I}_\ell| h},$$

with probability at least $1 - \epsilon$.

**Step 3. Bound for** $\sup_{\pi \in \Pi, s \in [0,1]} \left| \frac{1}{T} \sum_{t=1}^T f^2_{\pi,s}(x_t, z_t) - \mathbb{E}[f^2_{\pi,s}] \right|$. Using an analogous argument before, we can get
$$\sup_{\pi \in \Pi, s \in [0,1]} \left| \frac{1}{|\mathcal{I}_\ell|} \sum_{t \in \mathcal{I}_\ell} f^2_{\pi,s}(x_t, p_t) - \mathbb{E}[f^2_{\pi,s}] \right| \lesssim \frac{d_\Pi}{|\mathcal{I}_\ell|} + \sqrt{\frac{d_\Pi}{|\mathcal{I}_\ell|}} + \sqrt{\frac{\log(1/\epsilon)}{|\mathcal{I}_\ell|}} + \frac{\log(1/\epsilon)}{|\mathcal{I}_\ell|},$$

with probability at least $1 - \epsilon$. Specifically, $\left| f^2_{\pi,s} \right| \le 1$, and the second-moment bound is bounded by 1. For the indicator class $\mathcal{F}^2_{\Pi,s}$, we first note that for any binary-valued function class, its pseudo-dimension is identical to its VC-dimension (see Anthony & Bartlett, 2009). Note that
$$\mathcal{F}^2_{\Pi,s} = \{(x, z) \mapsto \mathbf{1}\{\pi(x) \cdot \mathbf{1}\{z \ge \pi(x) - f(x)\} \le s\} | \pi \in \Pi, s \in [0,1]\}$$
$$= \mathbf{1}\{(\pi(x) - f(x) > z) \text{ or } (\pi(x) - f(x) \le z \text{ and } \pi(x) \le s)\}$$

can be expressed as a finite Boolean combination of atomic level-set classes derived from $\Pi$. Since the VC-dimension of each atomic class is equivalent to $\text{Pdim}(\Pi) = d_\Pi$, it follows from the closure properties of VC-classes (Lemma 2.6.17 van der Vaart & Wellner, 1996) that
$$\text{Pdim}(\mathcal{F}^2_{\Pi,s}) = \text{VCdim}(\mathcal{F}^2_{\Pi,s}) \lesssim d_\Pi.$$

Consequently, we can now directly apply Lemma 19.15 in van der Vaart (1998) to obtain the uniform covering number bound (52), consistent with the methodology used in Step 1 and Step 2. Note that $|\mathcal{I}_\ell| = T/L$, we get
$$\sup_{\pi \in \Pi} |\tilde{R}^h_{\text{DRO}}(\pi) - \tilde{R}_{\text{DRO}}(\pi)| \lesssim \frac{4}{\rho} \frac{1}{L} \sum_{\ell=1}^L \frac{d_\Pi}{|\mathcal{I}_\ell| h} + \sqrt{\frac{d_\Pi}{Th}} + \sqrt{\frac{\log(1/\epsilon)}{|\mathcal{I}_\ell| h}} + \frac{\log(1/\epsilon)}{|\mathcal{I}_\ell| h} + \frac{d_\Pi}{|\mathcal{I}_\ell|} + \sqrt{\frac{d_\Pi}{|\mathcal{I}_\ell|}}$$
$$\lesssim \frac{d_\Pi L}{Th} + \sqrt{\frac{d_\Pi L}{Th}} + \sqrt{\frac{\log(1/\epsilon)L}{Th}} + \frac{\log(1/\epsilon)L}{Th}$$

which completes the proof. $\qquad \square$

**Lemma C.5.** *Let Assumptions 3.2, 2.2 and 4.2 hold. Picking $h \geq \Omega\left(\max\left(\frac{d_\Pi \log(1/\epsilon)}{T\rho^2}, \sqrt{\frac{d_\Pi + \log(1/\epsilon)}{2T/L}}\right)\right)$, we have*

$$\left|\tilde{R}_\delta(\pi) - \hat{R}_\delta(\pi)\right| \lesssim \overline{\alpha}\left(\log\left(\frac{1}{h}\right) \cdot \left(\frac{d_\Pi}{Th/L} + \sqrt{\frac{d_\Pi}{Th/L}} + \sqrt{\frac{\log(1/\epsilon)}{Th/L}} + \frac{\log(1/\epsilon)}{Th/L}\right) + T^{-\gamma_\pi - \gamma_m} + h^s\right),$$

*for $s = 1, 2$ in Assumption 2.4, which holds with probability at least $1 - 4\epsilon$.*

*Proof of Lemma C.5.* For each fold $\ell$, denote $\Delta_m^{(\ell)}(x, p; \alpha) = \hat{m}^{(\ell)}(x, p; \alpha) - m(x, p; \alpha)$, $v(x, p; \pi) = \frac{K\left(\frac{p - \pi(x)}{h}\right)}{h\pi_0(p|x)}/S_\ell^\pi$, and $\Delta_v^{(\ell)}(x, p; \pi) = \hat{v}^{(\ell)} - v$, where $\hat{v}^{(\ell)}$ uses the estimated propensity score $\hat{\pi}_0^{(\ell)}(p|x)$. We decompose $\hat{W}^{(\ell)}(\pi, \alpha) - \tilde{W}^{(\ell)}(\pi, \alpha)$ into three components

$$\hat{W}^{(\ell)}(\pi, \alpha) - \tilde{W}^{(\ell)}(\pi, \alpha) = \mathcal{E}_1^{(\ell)} + \mathcal{E}_2^{(\ell)} - \mathcal{E}_3^{(\ell)},$$

where

$$\mathcal{E}_1^{(\ell)} = \frac{1}{|\mathcal{I}_\ell|} \sum_{t \in \mathcal{I}_\ell} \left(\Delta_m^{(\ell)}(x_t, \pi(x_t); \alpha) - \frac{1}{S_\ell^\pi} \frac{K\left(\frac{p_t - \pi(x_t)}{h}\right)}{h\pi_0(p_t|x_t)} \Delta_m^{(\ell)}(x_t, p_t; \alpha)\right),$$

$$\mathcal{E}_2^{(\ell)} = \frac{1}{|\mathcal{I}_\ell|} \sum_{t \in \mathcal{I}_\ell} \left(\Delta_v^{(\ell)}(x_t, p_t; \pi)\left(\exp(-p_t y_t/\alpha) - m(x_t, p_t; \alpha)\right)\right),$$

$$\mathcal{E}_3^{(\ell)} = \frac{1}{|\mathcal{I}_\ell|} \sum_{t \in \mathcal{I}_\ell} \left(\Delta_v^{(\ell)}(x_t, p_t; \pi) \cdot \Delta_m^{(\ell)}(x_t, p_t; \alpha)\right).$$

Conditioning on dataset $\mathcal{I}_\ell^c$, the nuisance estimates $\hat{m}^{(\ell)}$ and $\hat{\pi}_0^{(\ell)}$ are fixed.

**Step 1. Bias of $\sup_{\pi \in \Pi, \alpha \in (0, \overline{\alpha}]} \left|\mathcal{E}_1^{(\ell)}\right|$, $\forall \ell \in [L]$.** Let $\ell \in [L]$ be fixed for now. We decompose the error $\mathcal{E}_1^{(\ell)}$ into two parts using the triangle inequality,

$$\left|\mathcal{E}_1^{(\ell)}\right| = \left|\frac{1}{|\mathcal{I}_\ell|} \sum_{t \in \mathcal{I}_\ell} \left(\Delta_m^{(\ell)}(x_t, \pi(x_t); \alpha) - \frac{1}{S_\ell^\pi} \frac{K\left(\frac{p_t - \pi(x_t)}{h}\right)}{h\pi_0(p_t|x_t)} \Delta_m^{(\ell)}(x_t, p_t; \alpha)\right)\right|$$

$$\leq \left|\frac{1}{|\mathcal{I}_\ell|} \sum_{t \in \mathcal{I}_\ell} \left(\Delta_m^{(\ell)}(x_t, \pi(x_t); \alpha) - \frac{K\left(\frac{p_t - \pi(x_t)}{h}\right)}{h\pi_0(p_t|x_t)} \Delta_m^{(\ell)}(x_t, p_t; \alpha)\right)\right| + \left|\frac{S_\ell^\pi - 1}{S_\ell^\pi |\mathcal{I}_\ell|} \sum_{t \in \mathcal{I}_\ell} \frac{K\left(\frac{p_t - \pi(x_t)}{h}\right)}{h\pi_0(p_t|x_t)} \Delta_m^{(\ell)}(x_t, p_t; \alpha)\right|$$

For the second term, using the fact that $m$ and $\hat{m}^{(\ell)}$ are bounded by 1, it is bounded by $|S_\ell^\pi - 1|$. From the concentration results in Lemma C.4, we have

$$\left|\frac{S_\ell^\pi - 1}{S_\ell^\pi |\mathcal{I}_\ell|} \sum_{t \in \mathcal{I}_\ell} \frac{K\left(\frac{p_t - \pi(x_t)}{h}\right)}{h\pi_0(p_t|x_t)} \Delta_m^{(\ell)}(x_t, p_t; \alpha)\right| \leq |S_\ell^\pi - 1| \lesssim \frac{d_\Pi}{|\mathcal{I}_\ell|h} + \sqrt{\frac{d_\Pi}{|\mathcal{I}_\ell|h}} + \sqrt{\frac{\log(1/\epsilon)}{|\mathcal{I}_\ell|h}} + \frac{\log(1/\epsilon)}{|\mathcal{I}_\ell|h},$$

with probability at least $1 - \epsilon$.

For the first term, we further split it into a zero-mean empirical process $\mathcal{E}_{1,1}^{(\ell)}$ and a conditional bias $\mathcal{E}_{1,2}^{(\ell)}$,

$$\phi_{\pi, \alpha}^{(\ell)}(x, p) = \Delta_m^{(\ell)}(x, \pi(x); \alpha) - \frac{K\left(\frac{p - \pi(x)}{h}\right)}{h\pi_0(p|x)} \Delta_m^{(\ell)}(x, p; \alpha),$$

$$\mathcal{E}_{1,1}^{(\ell)} = \frac{1}{|\mathcal{I}_\ell|} \sum_{t \in \mathcal{I}_\ell} \left(\phi_{\pi, \alpha}^{(\ell)}(\pi, \alpha) - \mathbb{E}\left[\phi_{\pi, \alpha}^{(\ell)}|\mathcal{I}_\ell^c\right]\right), \quad \mathcal{E}_{1,2}^{(\ell)} = \mathbb{E}\left[\phi_{\pi, \alpha}^{(\ell)}|\mathcal{I}_\ell^c\right].$$

For the conditional bias $\mathcal{E}_{1,2}^{(\ell)}$, we have

$$\left|\mathcal{E}_{1,2}^{(\ell)}\right| = \left|\mathbb{E}\left[\Delta_m^{(\ell)}(X, \pi(X); \alpha)\Big|\mathcal{I}_\ell^c\right] - \mathbb{E}\left[\mathbb{E}\left[\frac{K\left(\frac{P - \pi(X)}{h}\right)}{h\pi_0(P|X)} \Delta_m^{(\ell)}(X, P; \alpha)\Big| X, \mathcal{I}_\ell^c\right]\right]\right|$$

$$= \left| \mathbb{E}\left[ \Delta_m^{(\ell)}(X, \pi(X); \alpha) \middle| \mathcal{I}_\ell^c \right] - \mathbb{E}\left[ \int_{p_{\min}}^1 \frac{1}{h} K\left( \frac{p - \pi(X)}{h} \right) \Delta_m^{(\ell)}(X, p; \alpha) \mathrm{d}p \middle| \mathcal{I}_\ell^c \right] \right|$$

$$= \left| \mathbb{E}\left[ \Delta_m^{(\ell)}(X, \pi(X); \alpha) \middle| \mathcal{I}_\ell^c \right] - \mathbb{E}\left[ \int \Delta_m^{(\ell)}(X, \pi(X) + th; \alpha) K(t) \mathrm{d}t \middle| \mathcal{I}_\ell^c \right] \right|$$

$$= \left| \mathbb{E}\left[ \int \left( \Delta_m^{(\ell)}(X, \pi(X); \alpha) - \Delta_m^{(\ell)}(X, \pi(X) + th; \alpha) \right) K(t) \mathrm{d}t \right] \right|.$$

Under $s$-order bounded derivative in Assumptions 2.4 and 4.2, Following from the $s$-order Taylor expansion of $\Delta_m^{(\ell)}$ and Assumptions 3.2, 2.4,4.2, we obtain $\left| \mathcal{E}_{1,2}^{(\ell)} \right| \lesssim h^s$. To bound $\mathcal{E}_{1,1}^{(\ell)}$, we examine the Pseudo-dimension of the function class

$$\Phi_1 := \left\{ (x, p) \mapsto \phi_{\pi,\alpha}^{(\ell)} - \mathbb{E}\left[ \phi_{\pi,\alpha}^{(\ell)} \middle| \mathcal{I}_\ell^c \right] \;\middle|\; \pi \in \Pi, \alpha \in (0, \overline{\alpha}] \right\}.$$

Let $\mathbf{M} := \{ m_{\pi,\alpha} : x \mapsto m(x, \pi(x); \alpha) | \pi \in \Pi, \alpha \in (0, \overline{\alpha}] \}$. Note that for any fixed $(x, p)$, $m(x, p; \alpha)$ is a monotonic function of $\alpha$ on $(0, \overline{\alpha}]$, then the function class $\mathcal{F}_{\mathcal{A}} := \{ f_\alpha : (x, p) \mapsto m(x, p; \alpha) \mid \alpha \in (0, \overline{\alpha}] \}$ has a Pseudo-dimension of 1 (Theorem 11.3 in Anthony & Bartlett, 2009). By Lemma D.3(ii), the covering number of a one-parameter monotonic family satisfies $\log \mathcal{N}_2(\epsilon/2, \mathcal{F}_{\mathcal{A}}, |\mathcal{I}_\ell|) \lesssim \log(1/\epsilon)$. Crucially, because the monotonicity is uniform, we can select a grid of $\alpha$ values that forms an $\epsilon/2$-cover for all $x$. We first claim that

$$\mathcal{N}(\epsilon, \mathbf{M}, L_2(P_{|\mathcal{I}_\ell|})) \leq \mathcal{N}_2\left( \frac{\epsilon}{2L_m}, \Pi, |\mathcal{I}_\ell| \right) \times \mathcal{N}_2\left( \frac{\epsilon}{2}, \mathcal{F}_{\mathcal{A}}, |\mathcal{I}_\ell| \right).$$

For ease of notation, let

$$N_{2,\Pi}\left( \frac{\epsilon}{2L_m} \right) := \mathcal{N}_2\left( \frac{\epsilon}{2L_m}, \Pi, |\mathcal{I}_\ell| \right), \quad N_{2,\mathcal{A}}\left( \frac{\epsilon}{2} \right) := \mathcal{N}_2\left( \frac{\epsilon}{2}, \mathcal{F}_{\mathcal{A}}, |\mathcal{I}_\ell| \right).$$

Suppose $\{\pi_1, ..., \pi_{N_{2,\Pi}(\frac{\epsilon}{2L_m})}\}$ is a $\frac{\epsilon}{2L_m}$-cover for $\Pi$, and $f_{\alpha_1}, ..., f_{\alpha_{N_{2,\mathcal{A}}(\frac{\epsilon}{2})}}$ is a $\frac{\epsilon}{2}$-cover for $\mathcal{F}_{\mathcal{A}}$. Then, we can prove that $\mathbf{M}^\epsilon$ is a $\epsilon$-cover set for $\mathbf{M}$, where $\mathbf{M}^\epsilon$ is defined as

$$\mathbf{M}^\epsilon = \left\{ m_{\pi_i, \alpha_j} : x \mapsto m(x, \pi_i(x); \alpha_j) : i \leq N_{2,\Pi}\left( \frac{\epsilon}{2L_m} \right), j \leq N_{2,\mathcal{A}}\left( \frac{\epsilon}{2} \right) \right\}.$$

For $m(x, \pi(x); \alpha) \in \mathbf{M}$, we can pick $m(x, \pi'(x); \alpha') \in \mathbf{M}^\epsilon$ such that

$$\|\pi - \pi'\|_{L_2(P_{|\mathcal{I}_\ell|})} \leq \frac{\epsilon}{2L_m}, \quad \|f_\alpha - f_{\alpha'}\|_{L_2(P_{|\mathcal{I}_\ell|})} \leq \frac{\epsilon}{2}.$$

Then, we have

$$\|m_{\pi,\alpha} - m_{\pi',\alpha'}\|_{L_2(P_{|\mathcal{I}_\ell|})} = \sqrt{ \frac{1}{|\mathcal{I}_\ell|} \sum_{t \in \mathcal{I}_\ell} (m_{\pi,\alpha}(x_t) - m_{\pi',\alpha'}(x_t))^2 }$$

$$\leq \sqrt{ \frac{1}{|\mathcal{I}_\ell|} \sum_{t \in \mathcal{I}_\ell} (m(x_t, \pi(x_t); \alpha) - m(x_t, \pi'(x_t); \alpha))^2 + (m(x_t, \pi'(x_t); \alpha) - m(x_t, \pi'(x_t); \alpha'))^2 }$$

$$\leq \sqrt{ \frac{1}{|\mathcal{I}_\ell|} \sum_{t \in \mathcal{I}_\ell} L_m^2 (\pi(x_t) - \pi'(x_t))^2 + (f_\alpha(x_t, \pi'(x_t)) - f_{\alpha'}(x_t, \pi'(x_t)))^2 } \leq \epsilon.$$

Thus, by Lemma D.3, we obtain

$$\log \mathcal{N}\left( \epsilon, \mathbf{M}, L_2(P_{|\mathcal{I}_\ell|}) \right) \lesssim d_\Pi \log\left( \frac{1}{\epsilon} \right). \tag{55}$$

And let $\overline{\mathbf{M}} := \{ \mathbb{E}[m_{\pi,\alpha}(X)] | \pi \in \Pi, \alpha \in (0, \overline{\alpha}] \}$. Then

$$\|\mathbb{E}[m_{\pi,\alpha}] - \mathbb{E}[m_{\pi',\alpha'}]\|_{L_2(P_{|\mathcal{I}_\ell|})} = |\mathbb{E}[m_{\pi,\alpha}] - \mathbb{E}[m_{\pi',\alpha'}]| \leq \mathbb{E}|m_{\pi,\alpha} - m_{\pi',\alpha'}|$$

implies that $\mathcal{N}(\epsilon, \overline{\mathbf{M}}, L_2(P_{|\mathcal{I}_\ell})) \leq \mathcal{N}(\epsilon, \mathbf{M}, L_1(\mathbf{P}_0))$. Consequently, by Lemma D.3(iv), we have

$$\log \mathcal{N}\left(\epsilon, \overline{\mathbf{M}}, L_2(P_{|\mathcal{I}_\ell})\right) \lesssim d_\Pi \log\left(\frac{1}{\epsilon} \log\left(\frac{1}{\epsilon}\right)\right). \tag{56}$$

Conditional on $\mathcal{I}_\ell^c$, the estimated regression $\hat{m}^{(\ell)}(x, p; \alpha)$ is a fixed continuous function. Similarly, we obtain

$$\log \mathcal{N}\left(\epsilon, \hat{\mathbf{M}}, L_2(P_{|\mathcal{I}_\ell})\right) \lesssim d_\Pi \log\left(\frac{1}{\epsilon}\right), \quad \log \mathcal{N}\left(\epsilon, \overline{\hat{\mathbf{M}}}, L_2(P_{|\mathcal{I}_\ell})\right) \lesssim d_\Pi \log\left(\frac{1}{\epsilon} \log\left(\frac{1}{\epsilon}\right)\right). \tag{57}$$

for the classes

$$\hat{\mathbf{M}} := \left\{\hat{m}_{\pi,\alpha}^{(\ell)} : x \mapsto \hat{m}^{(\ell)}(x, \pi(x); \alpha)|\pi \in \Pi, \alpha \in (0, \overline{\alpha}]\right\} \quad \text{and} \quad \overline{\hat{\mathbf{M}}} := \left\{\mathbb{E}\left[\hat{m}_{\pi,\alpha}^{(\ell)}(X)\right]|\pi \in \Pi, \alpha \in (0, \overline{\alpha}]\right\}.$$

Let $\mathcal{K} := \left\{k_{\pi,\alpha} : (x, p) \mapsto \frac{K\left(\frac{p - \pi(x)}{h}\right)}{h\pi_0(p|x)} \Delta_m^{(\ell)}(x, p; \alpha)|\pi \in \Pi, \alpha \in (0, \overline{\alpha}]\right\}$. Then we can claim that

$$\mathcal{N}\left(\epsilon, \mathcal{K}, L_2(P_{|\mathcal{I}_\ell})\right) \leq \mathcal{N}_2\left(\epsilon/2, \mathcal{F}_\Pi, |\mathcal{I}_\ell|\right) \times \mathcal{N}_2\left(\frac{h\eta\epsilon}{4C_2}, \mathcal{F}_\mathcal{A}, |\mathcal{I}_\ell|\right) \times \mathcal{N}_2\left(\frac{h\eta\epsilon}{4C_2}, \hat{\mathcal{F}}_\mathcal{A}, |\mathcal{I}_\ell|\right),$$

where $\mathcal{F}_\Pi$ is defined in (50) and $\hat{\mathcal{F}}_\mathcal{A}^{(\ell)} := \left\{\hat{f}_\alpha^{(\ell)} : (x, p) \mapsto \hat{m}^{(\ell)}(x, p; \alpha) \mid \alpha \in (0, \overline{\alpha}]\right\}$. It follows from

$$\|k_{\pi,\alpha} - k_{\pi',\alpha'}\|_{L_2(P_{|\mathcal{I}_\ell})}$$

$$\leq \sqrt{\frac{1}{|\mathcal{I}_\ell|} \sum_{t \in \mathcal{I}_\ell} \left(\frac{K\left(\frac{p_t - \pi(x_t)}{h}\right)}{h\pi_0(p_t|x_t)} - \frac{K\left(\frac{p_t - \pi'(x_t)}{h}\right)}{h\pi_0(p_t|x_t)}\right)^2} + \frac{C_2}{h\eta} \sqrt{\frac{1}{|\mathcal{I}_\ell|} \sum_{t \in \mathcal{I}_\ell} (m(x_t, p_t; \alpha) - m(x_t, p_t; \alpha'))^2}$$

$$+ \frac{C_2}{h\eta} \sqrt{\frac{1}{|\mathcal{I}_\ell|} \sum_{t \in \mathcal{I}_\ell} \left(\hat{m}^{(\ell)}(x_t, p_t; \alpha) - \hat{m}^{(\ell)}(x_t, p_t; \alpha')\right)^2}.$$

Hence, we can bound the covering number as

$$\log \mathcal{N}\left(\epsilon, \mathcal{K}, L_2(P_{|\mathcal{I}_\ell})\right) \lesssim d_\Pi \log\left(\frac{1}{h\epsilon}\right). \tag{58}$$

Also for $\overline{\mathcal{K}} := \{\mathbb{E}[k_{\pi,\alpha}] | \pi \in \Pi, \alpha \in (0, \overline{\alpha}]\}$, we have

$$\log \mathcal{N}\left(\epsilon, \overline{\mathcal{K}}, L_2(P_{|\mathcal{I}_\ell})\right) \lesssim d_\Pi \log\left(\frac{1}{h\epsilon} \log\left(\frac{1}{h\epsilon}\right)\right). \tag{59}$$

Finally, we note that $\Phi_1$ is a subset of the vector space spanned by these classes $\mathbf{M}, \hat{\mathbf{M}}, \overline{\mathbf{M}}, \overline{\hat{\mathbf{M}}}, \mathcal{K}, \overline{\mathcal{K}}$. Combining (55)-(59), we obtain

$$\log \mathcal{N}\left(\epsilon, \Phi_1, L_2(P_{|\mathcal{I}_\ell})\right) \lesssim d_\Pi \log\left(\frac{1}{h\epsilon} \log\left(\frac{1}{h\epsilon}\right)\right),$$

which is a similar upper bound like (64). We further have the bound on the functions in $\Phi_1$,

$$\left|\phi_{\pi,\alpha}^{(\ell)}(x, p)\right| \leq 1 + \frac{C_2}{h\eta}, \quad \text{and} \quad \mathbb{E}\left[\left(\phi_{\pi,\alpha}^{(\ell)}(x, p)\right)^2\right] \leq 1 + \frac{C_2}{h\eta}.$$

It implies a similar Dudley's integral in Lemma C.6,

$$\mathbb{E}\left[\sup_{\pi \in \Pi, \alpha \in (0, \overline{\alpha}]} \left|\mathcal{E}_{1,1}^{(\ell)}\right|\right] \lesssim \log\left(\frac{1}{h}\right) \cdot \left(\frac{d_\Pi}{|\mathcal{I}_\ell|h} + \sqrt{\frac{d_\Pi}{|\mathcal{I}_\ell|h}} + \frac{d_\Pi}{|\mathcal{I}_\ell|} + \sqrt{\frac{d_\Pi}{|\mathcal{I}_\ell|}}\right).$$

Combining the above Rademacher complexity bound with Talagrand's Inequality (Lemma D.1), we obtain with probability at least $1 - \epsilon$

$$\sup_{\pi \in \Pi, \alpha \in (0, \overline{\alpha}]} \left| \mathcal{E}_{1,1}^{(\ell)} \right| \lesssim \mathbb{E} \left[ \sup_{\pi \in \Pi} \left| \mathcal{E}_{1,1}^{(\ell)} \right| \right] + \sqrt{\frac{\log(1/\epsilon)}{|\mathcal{I}_\ell|h}} + \frac{\log(1/\epsilon)}{|\mathcal{I}_\ell|h} + \sqrt{\frac{\log(1/\epsilon)}{|\mathcal{I}_\ell|}} + \frac{\log(1/\epsilon)}{|\mathcal{I}_\ell|},$$

Hence, for $s = 1, 2$ in Assumption 2.4, we have

$$\sup_{\pi \in \Pi, \alpha \in (0, \overline{\alpha}]} \left| \mathcal{E}_1^{(\ell)} \right| \lesssim \log \left( \frac{1}{h} \right) \left( \frac{d_\Pi}{|\mathcal{I}_\ell|h} + \sqrt{\frac{d_\Pi}{|\mathcal{I}_\ell|h}} + \sqrt{\frac{\log(1/\epsilon)}{|\mathcal{I}_\ell|h}} + \frac{\log(1/\epsilon)}{|\mathcal{I}_\ell|h} + \frac{d_\Pi}{|\mathcal{I}_\ell|} + \sqrt{\frac{d_\Pi}{|\mathcal{I}_\ell|}} + \sqrt{\frac{\log(1/\epsilon)}{|\mathcal{I}_\ell|}} + \frac{\log(1/\epsilon)}{|\mathcal{I}_\ell|} \right) + h^s.$$

**Step 2. Bound for** $\sup_{\pi \in \Pi, \alpha \in (0, \overline{\alpha}]} \left| \mathcal{E}_2^{(\ell)} \right|, \forall \ell \in [L]$. Let $\ell \in [L]$ be fixed for now. For any $\pi \in \Pi$ and $\alpha \in (0, \overline{\alpha}]$, we define

$$\psi_{\pi, \alpha}^{(\ell)}(x, p, y) = \Delta_v^{(\ell)}(x, p; \pi) \cdot (\exp(-py/\alpha) - m(x, p; \alpha)),$$

$$\Delta_v^{(\ell)}(x, p; \pi) = \frac{1}{h} K \left( \frac{p - \pi(x)}{h} \right) \left( \frac{1}{\hat{S}_\ell^\pi \hat{\pi}_0^{(\ell)}(p|x)} - \frac{1}{S_\ell^\pi \pi_0(p|x)} \right),$$

and the nuisance estimator $\hat{\pi}_0^{(\ell)}$ is fixed conditioned on the auxiliary dataset $\mathcal{I}_\ell^c$, We observe that each summand of $\mathcal{E}_2^{(\ell)}$ is zero-mean, by the definition of $m$. Now we analyze the Pseudo-dimension of the function class

$$\Psi_2 := \left\{ (x, p, y) \mapsto \psi_{\pi, \alpha}^{(\ell)}(x, p, y) | \pi \in \Pi, \alpha \in (0, \overline{\alpha}] \right\}.$$

We decompose $\psi_{\pi, \alpha}^{(\ell)}$ into three components involving $\pi$ and $\alpha$,

$$\left\| \psi_{\pi, \alpha}^{(\ell)} - \psi_{\pi', \alpha'}^{(\ell)} \right\|_{L_2(P_{|\mathcal{I}_\ell|})} = \sqrt{\frac{1}{|\mathcal{I}_\ell|} \sum_{t \in \mathcal{I}_\ell} \left( \psi_{\pi, \alpha}^{(\ell)}(x_t, p_t, y_t) - \psi_{\pi', \alpha'}^{(\ell)}(x_t, p_t, y_t) \right)^2}$$

$$\leq \sqrt{\frac{1}{|\mathcal{I}_\ell|} \sum_{t \in \mathcal{I}_\ell} \left( \Delta_v^{(\ell)}(x_t, p_t; \pi) - \Delta_v^{(\ell)}(x_t, p_t; \pi') \right)^2 (\exp(-py/\alpha) - m(x, p; \alpha))^2}$$

$$+ \sqrt{\frac{1}{|\mathcal{I}_\ell|} \sum_{t \in \mathcal{I}_\ell} \left( \Delta_v^{(\ell)}(x_t, p_t; \pi') \right)^2 (\exp(-p_t y_t/\alpha) - \exp(-p_t y_t/\alpha'))^2}$$

$$+ \sqrt{\frac{1}{|\mathcal{I}_\ell|} \sum_{t \in \mathcal{I}_\ell} \left( \Delta_v^{(\ell)}(x_t, p_t; \pi') \right)^2 (m(x_t, p_t; \alpha) - m(x_t, p_t; \alpha'))^2}.$$

For the first term, we use the fact that $(a + b)^2 \leq 2(a^2 + b^2)$ and $k_\pi(x, p) := \frac{1}{h} K \left( \frac{p - \pi(x)}{h} \right)$,

$$\sqrt{\frac{1}{|\mathcal{I}_\ell|} \sum_{t \in \mathcal{I}_\ell} \left( \Delta_v^{(\ell)}(x_t, p_t; \pi) - \Delta_v^{(\ell)}(x_t, p_t; \pi') \right)^2 (\exp(-p_t y_t/\alpha) - m(x_t, p_t; \alpha))^2}$$

$$\leq \sqrt{\frac{2}{|\mathcal{I}_\ell|} \sum_{t \in \mathcal{I}_\ell} \left( \frac{k_\pi(x_t, p_t)}{S_\ell^\pi \pi_0(p_t|x_t)} - \frac{k_{\pi'}(x_t, p_t)}{S_\ell^{\pi'} \pi_0(p_t|x_t)} \right)^2} + \sqrt{\frac{2}{|\mathcal{I}_\ell|} \sum_{t \in \mathcal{I}_\ell} \left( \frac{k_\pi(x_t, p_t)}{\hat{S}_\ell^\pi \hat{\pi}_0^{(\ell)}(p_t|x_t)} - \frac{k_{\pi'}(x_t, p_t)}{\hat{S}_\ell^{\pi'} \hat{\pi}_0^{(\ell)}(p_t|x_t)} \right)^2}$$

By Step 1 in Lemma C.4, we can pick $h \geq \sqrt{\frac{d_\Pi + \log(1/\epsilon)}{2|\mathcal{I}_\ell|}}$ such that $\sup_{\pi \in \Pi} |S_\ell^\pi| \leq \frac{3}{2}$ and $\inf_{\pi \in \Pi} |S_\ell^\pi| \geq 1/2$ with probability at least $1 - \epsilon$. Then, for any $\pi, \pi' \in \Pi$, we have

$$\sqrt{\frac{1}{|\mathcal{I}_\ell|} \sum_{t \in \mathcal{I}_\ell} \left( \frac{k_\pi(x_t, p_t)}{S_\ell^\pi \pi_0(p_t|x_t)} - \frac{k_{\pi'}(x_t, p_t)}{S_\ell^{\pi'} \pi_0(p_t|x_t)} \right)^2} = \sqrt{\frac{1}{|\mathcal{I}_\ell|} \sum_{t \in \mathcal{I}_\ell} \left( \frac{k_\pi(x_t, p_t) S_\ell^{\pi'} - k_{\pi'}(x_t, p_t) S_\ell^\pi}{S_\ell^\pi S_\ell^{\pi'} \pi_0(p_t|x_t)} \right)^2}$$

$$= \sqrt{\frac{1}{|\mathcal{I}_\ell|} \sum_{t \in \mathcal{I}_\ell} \left( \frac{(k_\pi(x_t, p_t) - k_{\pi'}(x_t, p_t)) S_\ell^{\pi'} + k_{\pi'}(x_t, p_t)(S_\ell^{\pi'} - S_\ell^{\pi})}{S_\ell^{\pi} S_\ell^{\pi'} \pi_0(p_t|x_t)} \right)^2}$$

$$\leq \frac{2}{\eta} \|k_\pi - k_{\pi'}\|_{L_2(P_{|\mathcal{I}_\ell|})} + 2|S_\ell^{\pi} - S_\ell^{\pi'}| \leq \frac{4}{\eta} \|k_\pi - k_{\pi'}\|_{L_2(P_{|\mathcal{I}_\ell|})},$$

where we use $\sum_t k_{\pi'}^2/(S_\ell^{\pi'} \pi_0)^2 \leq 1$, and

$$|S_\ell^{\pi} - S_\ell^{\pi'}| = \left| \frac{1}{|\mathcal{I}_\ell|} \sum_{t \in \mathcal{I}_\ell} \frac{1}{\pi_0(p_t|x_t)} (k_\pi(x_t, p_t) - k_{\pi'}(x_t, p_t)) \right| \leq \frac{1}{\eta} \sqrt{\frac{1}{|\mathcal{I}_\ell|} \sum_{t \in \mathcal{I}_\ell} (k_\pi(x_t, p_t) - k_{\pi'}(x_t, p_t))^2} = \frac{1}{\eta} \|k_\pi - k_{\pi'}\|_{L_2(P_{|\mathcal{I}_\ell|})}.$$

Furthermore, we note that

$$\sup_{\pi \in \Pi} |S_\ell^{\pi} - \hat{S}_\ell^{\pi}| = \sup_{\pi \in \Pi} \left| \frac{1}{|\mathcal{I}_\ell|} \sum_{t \in \mathcal{I}_\ell} \frac{\hat{\pi}_0^{(\ell)}(p_t|x_t) - \pi_0(p_t|x_t)}{\hat{\pi}_0^{(\ell)}(p_t|x_t)\pi_0(p_t|x_t)} k_\pi(x_t, p_t) \right| \leq \frac{1}{\eta} \|\hat{\pi}_0^{(\ell)} - \pi_0\|_\infty \cdot \sup_{\pi \in \Pi} \left( \frac{1}{|\mathcal{I}_\ell|} \sum_{t \in \mathcal{I}_\ell} \frac{k_\pi(x_t, p_t)}{\pi_0(p_t|x_t)} \right)$$

$$= \frac{1}{\eta} \|\hat{\pi}_0^{(\ell)} - \pi_0\|_\infty \cdot \sup_{\pi \in \Pi} S_\ell^{\pi} \leq \frac{3}{2\eta} \|\hat{\pi}_0^{(\ell)} - \pi_0\|_\infty, \tag{60}$$

by picking $h \geq \sqrt{\frac{d_\Pi + \log(1/\epsilon)}{2|\mathcal{I}_\ell|}}$, we have with probability $1 - \epsilon$

$$\sup_{\pi \in \Pi} \left| \mathbb{E}[S_\ell^{\pi}] - \mathbb{E}\left[\hat{S}_\ell^{\pi}\right] \right| \leq \mathbb{E}\left[ \sup_{\pi \in \Pi} \left| S_\ell^{\pi} - \hat{S}_\ell^{\pi} \right| \right] \leq \frac{3}{2\eta} \cdot \left( \mathbb{E}\left[ \left\| \hat{\pi}_0^{(\ell)}(p|x) - \pi_0(p|x) \right\|_\infty^2 \right] \right)^{1/2} \lesssim \frac{3T^{-\gamma_\pi}}{2\eta}.$$

Then, there exists a sufficiently large $T$ such that $\frac{3T^{-\gamma_\pi}}{2\eta} \leq 1/4$ such that $\inf_{\pi \in \Pi} \mathbb{E}\left[\hat{S}_\ell^{\pi}\right] \geq \inf_{\pi \in \Pi} \mathbb{E}[S_\ell^{\pi}] - 1/4 \geq 1/4$ with probability at least $1 - \epsilon$. Furthermore, we define

$$\hat{f}_\pi^{(\ell)}(x, p) = \frac{1}{h\hat{\pi}_0^{(\ell)}(p|x)} K\left( \frac{p - \pi(x)}{h} \right), \quad \hat{\mathcal{F}}_\Pi^{(\ell)} := \{f_\pi^{(\ell)} | \pi \in \Pi\}, \quad \hat{\mathcal{G}}_\Pi^{(\ell)} := \{\hat{g}_\pi^{(\ell)} = \hat{f}_\pi^{(\ell)} - \mathbb{E}[\hat{f}_\pi^{(\ell)}] | \pi \in \Pi\}.$$

Using Lemma C.6, we have

$$\mathbb{E}\left[ \sup_{\pi \in \Pi} \left| \frac{1}{|\mathcal{I}_\ell|} \sum_{t \in \mathcal{I}_\ell} \hat{f}_\pi^{(\ell)}(x_t, p_t) - \mathbb{E}[\hat{f}_\pi^{(\ell)}(x, p)] \right| \right] \lesssim \frac{d_\Pi}{|\mathcal{I}_\ell|h} + \sqrt{\frac{d_\Pi}{|\mathcal{I}_\ell|h}}.$$

By a similar argument in Step 1 of the proof of Lemma C.4, we have

$$\sup_{\pi \in \Pi} |\hat{f}_\pi^{(\ell)}(x, p)| \leq \frac{C_2}{h\eta}, \text{ and } \sup_{\pi \in \Pi} \mathbb{E}\left[ \left( \hat{f}_\pi^{(\ell)}(x, p) \right)^2 \right] \leq \frac{C_2}{h\eta^2}. \tag{61}$$

Recalling that $\hat{\pi}_0^{(\ell)}$ is trained on $T - |\mathcal{I}_\ell|$ data points which is fixed, and $\{\hat{f}_\pi^{(\ell)}(x_t, p_t)\}_{t \in \mathcal{I}_\ell}$ is i.i.d., we can get

$$\sup_{\pi \in \Pi} \left| \frac{1}{|\mathcal{I}_\ell|} \sum_{t \in \mathcal{I}_\ell} \hat{f}_\pi^{(\ell)}(x_t, p_t) - \mathbb{E}[\hat{f}_\pi^{(\ell)}(x, p)] \right| \lesssim \frac{d_\Pi}{|\mathcal{I}_\ell|h} + \sqrt{\frac{d_\Pi}{|\mathcal{I}_\ell|h}} + \sqrt{\frac{\log(1/\epsilon)}{|\mathcal{I}_\ell|h}} + \frac{\log(1/\epsilon)}{|\mathcal{I}_\ell|h}.$$

To proceed, picking $h \geq \sqrt{\frac{d_\Pi + \log(1/\epsilon)}{8|\mathcal{I}_\ell|}}$, we conclude $\inf_{\pi \in \Pi} \left| \hat{S}_\ell^{\pi} \right| \geq \inf_{\pi \in \Pi} \mathbb{E}[\hat{S}_\ell^{\pi}] - 1/8 \geq 1/8$ with probability at least $1 - 2\epsilon$. Thus, we reduce the second term to

$$\sqrt{\frac{1}{|\mathcal{I}_\ell|} \sum_{t \in \mathcal{I}_\ell} \left( \frac{k_\pi(x_t, p_t)}{\hat{S}_\ell^{\pi} \hat{\pi}_0^{(\ell)}(p_t|x_t)} - \frac{k_{\pi'}(x_t, p_t)}{\hat{S}_\ell^{\pi'} \hat{\pi}_0^{(\ell)}(p_t|x_t)} \right)^2} \leq \frac{8}{\eta} \|k_\pi - k_{\pi'}\|_{L_2(P_{|\mathcal{I}_\ell|})} + 8|\hat{S}_\ell^{\pi} - \hat{S}_\ell^{\pi'}| \leq \frac{16}{\eta} \|k_\pi - k_{\pi'}\|_{L_2(P_{|\mathcal{I}_\ell|})}.$$

For the second and third terms, using Assumption 3.2, there exists a constant $C > 0$ such that $|K(u)| \leq C$, above results $\inf_{\pi \in \Pi} |\hat{S}_\ell^\pi| \geq \frac{1}{8}$ and $\inf_{\pi \in \Pi} |S_\ell^\pi| \geq \frac{1}{2}$ satisfied with probability at least $1 - 2\epsilon$, and the assumption that $\hat{\pi}_0$ and $\pi_0$ are lower bounded by $\eta$,

$$\sup_{\pi \in \Pi} |\Delta_v^{(\ell)}(x_t, p_t; \pi)| \leq \frac{16C}{h\eta} \sup_{\pi \in \Pi} |S_\ell^\pi - \hat{S}_\ell^\pi| + \frac{2C}{h\eta^2}\|\pi_0 - \hat{\pi}_0\|_\infty \overset{\text{by (60)}}{\leq} \frac{26C}{h\eta^2}\|\hat{\pi}_0^{(\ell)} - \pi_0\|_\infty \leq \frac{26C}{h\eta^2}.$$

Thus, we reduce the second and third terms to

$$\sqrt{\frac{1}{|\mathcal{I}_\ell|} \sum_{t \in \mathcal{I}_\ell} \left(\Delta_v^{(\ell)}(x_t, p_t; \pi')\right)^2 (\exp(-p_t y_t/\alpha) - \exp(-p_t y_t/\alpha'))^2} \leq \frac{26C}{h\eta^2} \sqrt{\frac{1}{|\mathcal{I}_\ell|} \sum_{t \in \mathcal{I}_\ell} (\exp(-p_t y_t/\alpha) - \exp(-p_t y_t/\alpha'))^2},$$

$$\sqrt{\frac{1}{|\mathcal{I}_\ell|} \sum_{t \in \mathcal{I}_\ell} \left(\Delta_v^{(\ell)}(x_t, p_t; \pi')\right)^2 (m(x_t, p_t; \alpha) - m(x_t, p_t; \alpha'))^2} \leq \frac{26C}{h\eta^2} \sqrt{\frac{1}{|\mathcal{I}_\ell|} \sum_{t \in \mathcal{I}_\ell} (m(x_t, p_t; \alpha) - m(x_t, p_t; \alpha'))^2}.$$

Combined with the finite pseudo-dimension of $\mathcal{K}_\Pi = \left\{f_\pi : (x, p) \mapsto \frac{1}{h}K\left(\frac{p - \pi(x)}{h}\right) | \pi \in \Pi\right\}$ and the covering number of a one-parameter monotonic family, this inequality implies that

$$\log \mathcal{N}\left(\epsilon, \Psi_2, L_2(P_{|\mathcal{I}_\ell|})\right) \lesssim d_\Pi \log\left(\frac{1}{h\epsilon}\right).$$

By the similar argument in (61), we get with probability at least $1 - 3\epsilon$,

$$\sup_{\pi \in \Pi, \alpha \in (0, \overline{\alpha}]} \left|\mathcal{E}_2^{(\ell)}\right| \lesssim \log\left(\frac{1}{h}\right) \left(\frac{d_\Pi}{|\mathcal{I}_\ell|h} + \sqrt{\frac{d_\Pi}{|\mathcal{I}_\ell|h}} + \sqrt{\frac{\log(1/\epsilon)}{|\mathcal{I}_\ell|h}} + \frac{\log(1/\epsilon)}{|\mathcal{I}_\ell|h} + \frac{d_\Pi}{|\mathcal{I}_\ell|} + \sqrt{\frac{d_\Pi}{|\mathcal{I}_\ell|}} + \sqrt{\frac{\log(1/\epsilon)}{|\mathcal{I}_\ell|}} + \frac{\log(1/\epsilon)}{|\mathcal{I}_\ell|}\right)$$

$$\lesssim \log\left(\frac{1}{h}\right) \left(\frac{d_\Pi}{|\mathcal{I}_\ell|h} + \sqrt{\frac{d_\Pi}{|\mathcal{I}_\ell|h}} + \sqrt{\frac{\log(1/\epsilon)}{|\mathcal{I}_\ell|h}} + \frac{\log(1/\epsilon)}{|\mathcal{I}_\ell|h}\right).$$

**Step 3. Bound for** $\sup_{\pi \in \Pi, \alpha \in (0, \overline{\alpha}]} \left|\mathcal{E}_3^{(\ell)}\right|, \forall \ell \in [L]$. Let $\ell \in [L]$ be fixed for now. By previous step 2, picking $h \geq \sqrt{\frac{d_\Pi + \log(1/\epsilon)}{2|\mathcal{I}_\ell|}}$ we have $\sup_{\pi \in \Pi} |S_\ell^\pi| \leq \frac{3}{2}$ and $\inf_{\pi \in \Pi} |S_\ell^\pi| \geq 1/2$ with probability at least $1 - \epsilon$. Then, utilizing the normalization of $S_\ell^\pi$ and $\hat{S}_\ell^\pi$, we have

$$\frac{1}{|\mathcal{I}_\ell|} \sum_{t \in \mathcal{I}_\ell} \frac{K\left(\frac{p_t - \pi(x_t)}{h}\right)}{h} \left|\frac{1}{S_\ell^\pi \pi_0(p_t|x)} - \frac{1}{\hat{S}_\ell^\pi \hat{\pi}_0^{(\ell)}(p_t|x)}\right| = \frac{1}{|\mathcal{I}_\ell|} \sum_{t \in \mathcal{I}_\ell} \frac{K\left(\frac{p_t - \pi(x_t)}{h}\right)}{h} \left|\frac{\hat{S}_\ell^\pi(\hat{\pi}_0^{(\ell)} - \pi_0) + \pi_0(\hat{S}_\ell^\pi - S_\ell^\pi)}{S_\ell^\pi \hat{S}_\ell^\pi \pi_0 \hat{\pi}_0^{(\ell)}}\right|$$

$$\leq \frac{1}{\eta} \frac{1}{|\mathcal{I}_\ell|} \sum_{t \in \mathcal{I}_\ell} \frac{K\left(\frac{p_t - \pi(x_t)}{h}\right)}{h\pi_0(p_t|x_t)} \frac{1}{S_\ell^\pi} \|\hat{\pi}_0^{(\ell)} - \pi_0\|_\infty + 2\frac{1}{|\mathcal{I}_\ell|} \sum_{t \in \mathcal{I}_\ell} \frac{K\left(\frac{p_t - \pi(x_t)}{h}\right)}{h\hat{\pi}_0^{(\ell)}(p_t|x_t)} \frac{1}{\hat{S}_\ell^\pi} |S_\ell^\pi - \hat{S}_\ell^\pi|$$

$$\leq \frac{1}{\eta} \|\hat{\pi}_0^{(\ell)} - \pi_0\|_\infty + 2|S_\ell^\pi - \hat{S}_\ell^\pi|.$$

Furthermore, we reduce the expectation to

$$\mathbb{E}\left[\sup_{\pi \in \Pi, \alpha \in (0, \overline{\alpha}]} \left|\mathcal{E}_3^{(\ell)}\right|\right]$$

$$= \mathbb{E}\left[\sup_{\pi \in \Pi, \alpha \in (0, \overline{\alpha}]} \left|\frac{1}{|\mathcal{I}_\ell|} \sum_{t \in \mathcal{I}_\ell} \frac{K\left(\frac{p_t - \pi(x_t)}{h}\right)}{h} \left(\frac{1}{S_\ell^\pi \pi_0(p_t|x_t)} - \frac{1}{\hat{S}_\ell^\pi \hat{\pi}_0^{(\ell)}(p_t|x_t)}\right) \left(\hat{m}^{(\ell)}(x_t, p_t; \alpha) - m(x_t, p_t; \alpha)\right)\right|\right]$$

$$\leq E\left[\left(\frac{1}{\eta}\|\hat{\pi}_0^{(\ell)} - \pi_0\|_\infty + 2\sup_{\pi \in \Pi} |S_\ell^\pi - \hat{S}_\ell^\pi|\right) \cdot \sup_{\alpha \in (0, \overline{\alpha}]} \left\|\hat{m}^{(\ell)} - m\right\|_\infty\right]$$

$$\leq \left( \frac{1}{\eta} \left( \mathbb{E}[\|\hat{\pi}_0^{(\ell)} - \pi_0\|_\infty^2] \right)^{1/2} + 2\mathbb{E}\left[ \left( \sup_{\pi\in\Pi} |S_\ell^\pi - \hat{S}_\ell^\pi| \right)^2 \right]^{1/2} \right) \left( \mathbb{E}\left[ \left( \sup_{\alpha\in(0,\overline{\alpha}]} \left\| \hat{m}^{(\ell)} - m \right\|_\infty \right)^2 \right] \right)^{1/2}.$$

Using definition of estimation rate, and the fact that $\hat{\pi}_0^{(\ell)}(p|x), \hat{m}^{(\ell)}(x,p;\alpha)$ is trained on $T - |\mathcal{I}_\ell|$ data points (due to cross-fitting), we continue to bound (60)

$$\mathbb{E}\left[ \left( \sup_{\pi\in\Pi} |S_\ell^\pi - \hat{S}_\ell^\pi| \right)^2 \right] \leq \frac{1}{\eta^2} \mathbb{E}\left[ \left\| \hat{\pi}_0^{(\ell)} - \pi_0 \right\|_\infty^2 \right] \cdot \mathbb{E}\left[ \left( \sup_{\pi\in\Pi} S_\ell^\pi \right)^2 \right] \leq \frac{9}{4\eta^2} \mathbb{E}\left[ \left\| \hat{\pi}_0^{(\ell)} - \pi_0 \right\|_\infty^2 \right].$$

Thus, we have

$$\mathbb{E}\left[ \sup_{\pi\in\Pi,\alpha\in(0,\overline{\alpha}]} \left| \mathcal{E}_3^{(\ell)} \right| \right] \leq \frac{4}{\eta} \left( \mathbb{E}[\|\hat{\pi}_0^{(\ell)} - \pi_0\|_\infty^2] \right)^{1/2} \left( \mathbb{E}\left[ \left( \sup_{\alpha\in(0,\overline{\alpha}]} \left\| \hat{m}^{(\ell)} - m \right\|_\infty \right)^2 \right] \right)^{1/2} \lesssim \frac{4}{\eta} T^{-\gamma_\pi - \gamma_m}.$$

Following step (a)-(b) in the proof of Lemma C.2,

$$\left| \tilde{R}_\delta(\pi) - \hat{R}_\delta(\pi) \right| \leq \sup_{\alpha>0} \alpha \left| \log \frac{\frac{1}{L}\sum_{\ell=1}^{L} \hat{W}^{(\ell)}(\pi,\alpha)}{\frac{1}{L}\sum_{\ell=1}^{L} \tilde{W}^{(\ell)}(\pi,\alpha)} \right| \leq \frac{4}{\rho L} \sum_{\ell=1}^{L} \sup_{\alpha>0} \alpha \left| \hat{W}^{(\ell)}(\pi,\alpha) - \tilde{W}^{(\ell)}(\pi,\alpha) \right|,$$

and $\tilde{W}^{(\ell)}(\pi,\alpha) \geq \rho/2$ with probability at least $1 - \epsilon$ and $\Omega\left( \sqrt{\frac{d_\Pi + \log(1/\epsilon)}{T}} \right) \leq h \leq \mathcal{O}\left( \frac{\rho}{8 L_m C_1} \right)$ by Lemma C.7, we obtain with probability at least $1 - 4\epsilon$, by combining Steps 1-3,

$$\left| \tilde{R}_\delta(\pi) - \hat{R}_\delta(\pi) \right| \lesssim \overline{\alpha} \left( \log\left( \frac{1}{h} \right) \left( \frac{d_\Pi}{Th/L} + \sqrt{\frac{d_\Pi}{Th/L}} + \sqrt{\frac{\log(1/\epsilon)}{Th/L}} + \frac{\log(1/\epsilon)}{Th/L} \right) + T^{-\gamma_\pi - \gamma_m} + h^s \right),$$

for $s = 1, 2$ in Assumption 2.4, which completes the proof. $\qquad\square$

**Lemma C.6** (Rademacher Complexity Bound via Pseudo-dimension). *Define a function class $\mathcal{F}_\Pi = \{f_\pi | \pi \in \Pi\}$, which has finite pseudo dimension $Pdim(\mathcal{F}_\Pi) \leq C \cdot Pdim(\Pi)$ for some constant $C$, and is uniformly bounded by $B_\Pi$. Let its second-moment be bounded by $\sigma^2$. Under Assumption 4.3, the expectation of the Rademacher complexity is bounded as:*

$$\mathbb{E}\left[ \sup_{\pi\in\Pi} \left| \frac{1}{T'} \sum_{t=1}^{T'} f_\pi(x_t, p_t) - \mathbb{E}[f_\pi] \right| \right] \lesssim Pdim(\Pi) \frac{B_\Pi}{T'} + \sqrt{\frac{\sigma^2 Pdim(\Pi)}{T'}}, \quad \text{for any } T' \geq Pdim(\mathcal{F}_\Pi).$$

*Proof of Lemma C.6.* The proof structure closely follows that of Lemma A.5 in Kitagawa & Tetenov (2018) and Lemma B.8 in Ai et al. (2026). Kitagawa & Tetenov (2018) assumes that the function class $\mathcal{G}_\Pi$ has the VC dimension $V$, and Ai et al. (2026) relaxes this condition to the exponential bound on the covering number. In our case, we instead replace the VC-dimension-based covering number bound with the pseudo-dimension-based one. Here we provide the detailed proof.

We can bound this term by the standard symmetrization lemma (Proposition 4.11 in Wainwright, 2019). Define $\mathcal{G}_\Pi := \{g_\pi = f_\pi - \mathbb{E}[f_\pi] | \pi \in \Pi\}$ as a centered function class. Introduce i.i.d. Rademacher variables $\epsilon_t \in \{-1, +1\}$ to get

$$\mathbb{E}\left[ \sup_{g\in\mathcal{G}_\Pi} \left| \sum_{t=1}^{T'} g(x_t, p_t) \right| \right] \leq 2\mathbb{E}\left[ \sup_{g\in\mathcal{G}_\Pi} \left| \sum_{t=1}^{T'} \epsilon_t g(x_t, p_t) \right| \right] \tag{62}$$

The expectation of the supremum of the Rademacher process is bounded by Dudley Theorem (Equation (5.48) in Wainwright, 2019):

$$\mathbb{E}\left[ \sup_{g\in\mathcal{G}_\Pi} \left| \sum_{t=1}^{T'} \epsilon_t g(x_t, p_t) \right| \right] \lesssim \mathbb{E}\left[ \sqrt{T'} \int_0^{2B_\Pi} \sqrt{\log N(\epsilon, \mathcal{G}_\Pi, L_2(P_{T'}))} \, d\epsilon \right] \tag{63}$$

where $L_2(P_{T'})$ is the empirical norm, $\mathcal{N}(\epsilon, \mathcal{G}_\Pi, L_2(P_{T'}))$ is the $L_2(P_{T'})$-covering number. Note that

$$\mathcal{N}(\epsilon, \mathcal{G}_\Pi, L_2(P_{T'})) \leq \mathcal{N}(\epsilon/2, \mathcal{F}_\Pi, L_2(P_{T'})) + N(\epsilon/2, \overline{\mathcal{F}}_\Pi, L_2(P_{T'})),$$

where

$$\overline{\mathcal{F}}_\Pi \triangleq \left\{ \overline{f}_\pi = \mathbb{E}[f_\pi(X, P)] \, \big| \, \pi \in \Pi \right\}$$

is a function class of constant functions, implying

$$\|\overline{f}_{\pi_1} - \overline{f}_{\pi_2}\|_{L_2(P_{T'})} = |\overline{f}_{\pi_1}, - \overline{f}_{\pi_2}| \leq \mathbb{E}_{\mathbf{P}_0 * \pi_0} |f_{\pi_1}, (X, P) - f_{\pi_2}(X, P)|.$$

Consequently, we have

$$\mathcal{N}(\epsilon/2, \overline{\mathcal{F}}_\Pi, L_2(P_{T'})) \leq \mathcal{N}(\epsilon/2, \mathcal{F}_\Pi, L_1(\mathbf{P}_0 * \pi_0))$$

To proceed, we use Definition D.2 and Lemma D.3,

$$\log \mathcal{N}(\epsilon/2, \mathcal{F}_\Pi, L_1(\mathbf{P}_0 * \pi_0)) \leq \log \mathcal{M}(\epsilon/2, \mathcal{F}_\Pi, L_1(\mathbf{P}_0 * \pi_0)) \lesssim \mathrm{Pdim}(\mathcal{F}_\Pi) \log \left( \frac{1}{\epsilon} \log \left( \frac{1}{\epsilon} \right) \right) \lesssim d_\Pi \log \left( \frac{1}{\epsilon} \log \left( \frac{1}{\epsilon} \right) \right).$$

Hence, we conclude that

$$\log \mathcal{N}(\epsilon, \mathcal{G}_\Pi, L_2(P_{T'})) \leq 2 \log \mathcal{N}(\epsilon/2, \mathcal{F}_\Pi, L_2(P_{T'})) + 2 \log N(\epsilon/2, \overline{\mathcal{F}}_\Pi, L_2(P_{T'})) \lesssim d_\Pi \log \left( \frac{B_\Pi}{\epsilon} \right), \tag{64}$$

where we use $\log a + \log b \leq 2 \log(a + b)$ and $\log(x \log x) \leq 2 \log x$.

The following is slightly different from the last inequality in the proof of Lemma A.5 in Kitagawa & Tetenov (2018). In particular, replacing (A.5) by (64) gives

$$\mathbb{E}\left[ \sup_{g \in \mathcal{G}_\Pi} \sum_{t=1}^{T'} g(x_t, p_t) \right] \lesssim B_\Pi \sqrt{T'} \sqrt{d_{\mathcal{G}_\Pi}} \left( \sqrt{\frac{d_\Pi}{T'}} + \sqrt{\frac{d_\Pi}{T'} + \frac{\sigma^2}{B_{\mathcal{G}_\Pi}^2}} \right) \lesssim B_\Pi d_\Pi + \sqrt{T'} \sqrt{\sigma^2 d_\Pi},$$

which completes the proof. □

**Lemma C.7.** *Under Assumptions 3.2 and 2.2, picking* $\Omega\left(\sqrt{\frac{d_\Pi + \log(1/\epsilon)}{T}}\right) \leq h \leq \mathcal{O}\left(\frac{\rho}{8 L_m C_1}\right)$, *then we have* $\tilde{W}^{(\ell)}(\pi, \alpha) \geq \rho/2$ *with probability at least* $1 - \epsilon$.

*Proof of Lemma C.7.* Define

$$\tilde{W}^{(\ell)'}(\pi, \alpha) \triangleq \frac{1}{|\mathcal{I}_\ell| S_\ell^\pi} \sum_{t \in \mathcal{I}_\ell} \frac{K(\frac{p_t - \pi(x_t)}{h})}{h \pi_0(p_t | x_t)} \exp(-p_t y_t / \alpha) + \frac{1}{|\mathcal{I}_\ell|} \sum_{t \in \mathcal{I}_\ell} \left( m(x_t, \pi(x_t); \alpha) - \frac{1}{S_T^\pi} \frac{K(\frac{p_t - \pi(x_t)}{h})}{h \pi_0(p_t | x_t)} m(x_t, \pi(x_t); \alpha) \right),$$

Then, we can decompose as

$$|\tilde{W}^{(\ell)}(\pi, \alpha) - \tilde{W}'_h(\pi, \alpha)| \leq |\tilde{W}^{(\ell)}(\pi, \alpha) - \tilde{W}^{(\ell)'}(\pi, \alpha)| + |\tilde{W}^{(\ell)'}(\pi, \alpha) - \tilde{W}'_h(\pi, \alpha)|.$$

Define

$$A_\ell(\pi) \triangleq \frac{1}{|\mathcal{I}_\ell|} \sum_{t \in \mathcal{I}_\ell} \frac{K(\frac{p_t - \pi(x_t)}{h})}{h \pi_0(p_t | x_t)} (e^{-p_t y_t / \alpha} - m(x_t, \pi(x_t); \alpha)) \quad \text{and} \quad B_\ell(\pi) \triangleq \frac{1}{|\mathcal{I}_\ell|} \sum_{t \in \mathcal{I}_\ell} m(x_t, \pi(x_t); \alpha).$$

Then, we rewrite as $\tilde{W}^{(\ell)'}(\pi, \alpha) = \frac{A_\ell}{S_\ell^\pi} + B_\ell$. First, we provide the high probability upper bound for $\sup_{\pi \in \Pi, \alpha \in (0, \bar{\alpha}]} |A_\ell(\pi) - \mathbb{E}[A_\ell(\pi)]|$. Define the function class $\mathcal{M}_A = \{m_{\pi, \alpha}^A(x, p, y) : \pi \in \Pi, \alpha \in (0, \bar{\alpha}]\}$, where

$$m_{\pi, \alpha}^A(x, p, y) = \frac{1}{h \pi_0(p | x)} K\left( \frac{p - \pi(x)}{h} \right) \left[ e^{-py/\alpha} - m(x, \pi(x); \alpha) \right].$$

Recalling that $|e^{-py/\alpha} - m(x, \pi(x); \alpha)| \leq 1$, we have $\sup_{m_{\pi,\alpha}^A \in \mathcal{M}_A} \|m_\pi^A\|_\infty \leq \frac{C_2}{h\eta} := B_\Pi$ by Assumption 3.2 and 2.2. Also, for any $m_{\pi,\alpha}^A \in \mathcal{M}_A$, we have the second-moment bound $\mathbb{E}[(m_{\pi,\alpha}^A)^2] \leq \frac{C_2}{h\eta} := \sigma^2$ by (49). Following the similar argument of (58), we have $\text{Pdim}(\mathcal{M}_A) \lesssim d_\Pi$. Utilizing Talagrand Inequality (Lemma D.1) and setting $\beta(\epsilon) \lesssim \sqrt{\frac{\sigma^2 \log(1/\epsilon)}{|\mathcal{I}_\ell|}} + \frac{B_\Pi \log(1/\epsilon)}{|\mathcal{I}_\ell|}$, we obtain

$$\sup_{\pi \in \Pi, \alpha \in (0,\bar\alpha]} |A_\ell(\pi) - \mathbb{E}[A_\ell(\pi)]| \leq \mathbb{E}\Big[\sup_{\pi \in \Pi, \alpha \in (0,\bar\alpha]} |A_\ell(\pi) - \mathbb{E}[A_\ell(\pi)]|\Big] + \beta(\epsilon),$$

with probability at least $1 - \epsilon/3$. Substituting $\mathbb{E}\Big[\sup_{\pi \in \Pi, \alpha \in (0,\bar\alpha]} |A_\ell(\pi) - \mathbb{E}[A_\ell(\pi)]|\Big] \lesssim \frac{B_\Pi d_\Pi}{|\mathcal{I}_\ell|} + \sqrt{\frac{\sigma^2 d_\Pi}{|\mathcal{I}_\ell|}}$ by Lemma C.6, we get the upper bound for

$$\sup_{\pi \in \Pi, \alpha \in (0,\bar\alpha]} |A_\ell(\pi) - \mathbb{E}[A_\ell(\pi)]| \lesssim \frac{d_\Pi}{|\mathcal{I}_\ell|h} + \sqrt{\frac{d_\Pi}{|\mathcal{I}_\ell|h}} + \sqrt{\frac{\log(1/\epsilon)}{|\mathcal{I}_\ell|h}} + \frac{\log(1/\epsilon)}{|\mathcal{I}_\ell|h}.$$

Second, we provide the high probability upper bound for $\sup_{\pi \in \Pi, \alpha \in (0,\bar\alpha]} |B_\ell(\pi) - \mathbb{E}[B_\ell(\pi)]|$. Recalling $\mathbf{M} = \{m_{\pi,\alpha} : x \mapsto m(x, \pi(x); \alpha) : \pi \in \Pi, \alpha \in (0, \bar\alpha]\}$ in the proof of Lemma C.5, we also have

$$\sup_{\pi \in \Pi, \alpha \in (0,\bar\alpha]} |B_\ell(\pi) - \mathbb{E}[B_\ell(\pi)]| \lesssim \frac{d_\Pi}{|\mathcal{I}_\ell|} + \sqrt{\frac{d_\Pi}{|\mathcal{I}_\ell|}} + \sqrt{\frac{\log(1/\epsilon)}{|\mathcal{I}_\ell|}} + \frac{\log(1/\epsilon)}{|\mathcal{I}_\ell|},$$

with probability at least $1 - \epsilon/3$. Thus, combined with Step 1 in Lemma C.4, with the probability at least $1 - 2\epsilon/3$, we have

$$|\tilde{W}^{(\ell)'}(\pi, \alpha) - \tilde{W}_h'(\pi, \alpha)| = \left|\left(\frac{A_\ell}{S_\ell} + B_\ell\right) - (\mathbb{E}[A_\ell] + \mathbb{E}[B_\ell])\right| = \mathcal{O}\Big(\sqrt{\frac{d_\Pi \log(1/\epsilon)}{|\mathcal{I}_\ell|h}}\Big). \tag{65}$$

To proceed, we pick $h \geq \Omega\left(\frac{d_\Pi \log(1/\epsilon)}{|\mathcal{I}_\ell|\underline{\rho}^2}\right)$ such that $|\tilde{W}^{(\ell)'}(\pi, \alpha) - \tilde{W}_h'(\pi, \alpha)| \leq \underline{\rho}/4$.

On the other hand, , from the definitions of $\tilde{W}^{(\ell)}(\pi, \alpha), \tilde{W}^{(\ell)'}(\pi, \alpha)$, we have

$$|\tilde{W}^{(\ell)}(\pi, \alpha) - \tilde{W}^{(\ell)'}(\pi, \alpha)| = \left|\frac{1}{|\mathcal{I}_\ell|S_\ell^\pi} \sum_{t \in \mathcal{I}_\ell} \frac{K\left(\frac{p_t - \pi(x_t)}{h}\right)}{h\pi_0(p_t|x_t)}\Big(m(x_t, \pi(x_t); \alpha) - m(x_t, p_t; \alpha)\Big)\right|$$

$$\leq \frac{1}{|\mathcal{I}_\ell|} \sum_{t \in \mathcal{I}_\ell} \frac{K\left(\frac{p_t - \pi(x_t)}{h}\right)}{h\pi_0(p_t|x_t)} \cdot L_m |p_t - \pi(x_t)| + |S_\ell^\pi - 1| \quad \text{(by } L_m\text{-Lipschitz of } m(x, \cdot; \alpha) \text{ in (37) and (47))}$$

Consider $\tilde{S}_\ell^\pi \triangleq \frac{1}{|\mathcal{I}_\ell|} \sum_{t \in \mathcal{I}_\ell} \frac{K\left(\frac{p_t - \pi(x_t)}{h}\right)}{h\pi_0(p_t|x_t)} \frac{|p_t - \pi(x_t)|}{h}$.

Define a function class $\tilde{\mathcal{K}} = \left\{k_\pi(x, p) = \frac{1}{h\pi_0(p|x)} K\left(\frac{p - \pi(x)}{h}\right) \left|\frac{p - \pi(x)}{h}\right| : \pi \in \Pi\right\}$. Under Assumption 3.2, it follows that $\sup_u |uK(u)| \leq C_1$, $\int u^2 K^2(u)du < C_2$ and $\int |uK(u)|du < C_3$ for some constants $C_1, C_2, C_3 < \infty$. This ensures the uniform boundedness $\sup_{k_\pi \in \tilde{\mathcal{K}}} |k_\pi| \leq \frac{C_1}{h\eta}$ and $\mathbb{E}[k_\pi^2] \leq \frac{C_2}{h\eta}$. Thus, by a similar process, we have with probability at least $1 - \epsilon/6$,

$$\sup_{\pi \in \Pi} \left|\tilde{S}_T^\pi\right| \lesssim |\mathbb{E}[k_\pi]| + \frac{d_\Pi}{|\mathcal{I}_\ell|h} + \sqrt{\frac{d_\Pi}{|\mathcal{I}_\ell|h}} + \sqrt{\frac{\log(1/\epsilon)}{|\mathcal{I}_\ell|h}} + \frac{\log(1/\epsilon)}{|\mathcal{I}_\ell|h} \leq \frac{C_3}{\eta} + \frac{d_\Pi}{|\mathcal{I}_\ell|h} + \sqrt{\frac{d_\Pi}{|\mathcal{I}_\ell|h}} + \sqrt{\frac{\log(1/\epsilon)}{|\mathcal{I}_\ell|h}} + \frac{\log(1/\epsilon)}{|\mathcal{I}_\ell|h}$$

Picking $h \geq \Omega\left(\max\{\sqrt{\frac{d_\Pi + \log(1/\epsilon)}{2C_3^2|\mathcal{I}_\ell|/\eta^2}}, \sqrt{\frac{d_\Pi + \log(1/\epsilon)}{2|\mathcal{I}_\ell|}}\}\right)$, we have with probability at least $1 - \epsilon/3$, $\sup_{\pi \in \Pi} |\tilde{S}_T^\pi| \leq 2C_3/\eta$, $\inf_{\pi \in \Pi} |S_T^\pi| \geq 1/2$ by Step 1 in Lemma C.4, then $\sup_{\pi \in \Pi} |\tilde{S}_T^\pi/S_T^\pi| \leq 4C_3/\eta$. To proceed, we pick $\Omega\left(\sqrt{\frac{d_\Pi + \log(1/\epsilon)}{2|\mathcal{I}_\ell|\underline{\rho}^2}}\right) \leq h \leq \frac{\underline{\rho}\eta}{8L_m C_3}$ such that $|\tilde{W}_T'(\pi, \alpha) - \tilde{W}_h'(\pi, \alpha)| \leq \underline{\rho}/4$.

To conclude, we obtain $|\tilde{W}^{(\ell)}(\pi, \alpha) - \tilde{W}_h'(\pi, \alpha)| \leq \underline{\rho}/2$, with probability at least $1 - \epsilon$, which implies that $\tilde{W}_T(\pi, \alpha) \geq \underline{\rho} - \frac{\rho}{2} \geq \frac{\rho}{2}$, where we use (43). □

## D. Proof of Auxiliary Results

**Lemma D.1** (Talagrand's Inequality, Lemma B.9 in Ai et al. (2026)). *Let $Z_1, ..., Z_T$ be i.i.d. $\mathcal{Z}$-random variables and $\mathcal{F}$ be a class of functions where each $f : \mathcal{Z} \to \mathbb{R}$ in $\mathcal{F}$ satisfies $\sup_{f \in \mathcal{F}} \sup_{z \in \mathcal{Z}} |f(z)| \le B(\mathcal{F})$ and $\sup_{f \in \mathcal{F}} \mathbb{E}[f^2(Z)] \le \sigma^2(\mathcal{F})$.*
*Denote $\Delta \equiv \sup_{f \in \mathcal{F}} \left| \frac{1}{T} \sum_{t=1}^T (f(Z_t) - \mathbb{E}[f(Z_t)]) \right|$. Then, the following right and left tail bounds hold for all $\epsilon > 0$*

$$\mathbb{P}\left(\Delta - \mathbb{E}[\Delta] > \epsilon\right) \le 2 \exp\left(-\frac{T\epsilon^2}{8e^2\sigma(\mathcal{F}) + 16eB(\mathcal{F})\mathbb{E}[\Delta] + 4B(\mathcal{F})\epsilon}\right),$$

$$\mathbb{P}\left(-(\Delta - \mathbb{E}[\Delta]) > \epsilon\right) \le 2 \exp\left(-\frac{T\epsilon^2}{8e^2\sigma(\mathcal{F}) + 16eB(\mathcal{F})\mathbb{E}[\Delta] + 4B(\mathcal{F})\epsilon}\right).$$

Next, we introduce some related results for the pseudo-dimension.

**Definition D.2.** For a function class $\mathcal{H}$ and domain $\mathcal{X}$,
(i) (Theorem 12.1 in Anthony & Bartlett, 2009). For a set $A$, the covering number $\mathcal{N}(\epsilon, A, d)$ is the size of the smallest $\epsilon$-cover of $A$ under metric $d$. Packing number $\mathcal{M}(\epsilon, A, d)$ is the maximum cardinality of an $\epsilon$-separated subset of $A$. For any subset $W \subset A$, it satisfies $\mathcal{M}(2\epsilon, W, d) \le \mathcal{N}(\epsilon, W, d) \le \mathcal{M}(\epsilon, W, d)$.
(iii) Uniform covering number is defined as $\mathcal{N}_p(\epsilon, \mathcal{H}, m) := \max_{|C|=m} \mathcal{N}(\epsilon, \mathcal{H}|_C, d_p)$, and uniform packing number is $\mathcal{M}_p(\epsilon, \mathcal{H}, m) := \max_{|C|=m} \mathcal{M}(\epsilon, \mathcal{H}|_C, d_p)$, where $d_p$ is the empirical distance, $d_2(x,y) = \sqrt{\frac{1}{m}\sum_{i=1}^m (x_i - y_i)^2}$, $d_1(x,y) = \frac{1}{m}\sum_{i=1}^m |x_i - y_i|$, and $d_\infty(x,y) = \max_{i \in [m]} |x_i - y_i|$.
(iv) Pseudo-dimension $Pdim(\mathcal{H})$ is the cardinality of the largest set $(w_1, \ldots, w_m)$ that can be pseudo-shattered, i.e., there exists a witness $u \in \mathbb{R}^m$ such that for all $b \in \{\pm 1\}^m$, there is $h_b \in \mathcal{H}$ satisfying $\mathrm{sgn}(h_b(w_i) - u_i) = b_i$.

**Lemma D.3** (Theorem 1,2 in Haussler (1995), Theorem 11.4, 18.5 in Anthony & Bartlett (2009)). *Let $\mathcal{H} : \mathcal{X} \to [0, B]$ with $Pdim(\mathcal{H}) = d$. The following hold:*
*(i) Define $B_{\mathcal{H}} = \{B_h(x, r) = \mathrm{sgn}(h(x) - r) : h \in \mathcal{H}\}$, then VC dimension satisfies $VC(B_{\mathcal{H}}) = Pdim(\mathcal{H})$.*
*(ii) Covering Number satisfies $\mathcal{N}_1(\epsilon, \mathcal{H}, m) \le e(d+1)(2e/\epsilon)^d$.*
*(iii) For all natural numbers $d \ge 1$ and $s \ge 1$, let $n = sd$. There exists a subset $V \subset \{0,1\}^n$ with VC dimension $d$ such that for each integer $k \in \{1, \ldots, n\}$, the $k/n$-packing number of $V$ with respect to the $d_1$ metric satisfies $\mathcal{M}(k/n, V, d_1) \ge \left(\frac{n}{2e(k+d)}\right)^d$.*
*(iv) Suppose $\mathcal{H} : \mathcal{X} \to [0, 1]$, and has finite pseudo-dimension $d$. Then $\mathcal{M}(\epsilon, \mathcal{H}, L_1(P)) \le 2\left(\frac{2e}{\epsilon} \log\left(\frac{8e}{\epsilon}\right)\right)^d$ for any probability distribution $P$ on $\mathcal{X}$, and for all $0 < \epsilon \le 1$.*
*(v) $\mathcal{N}(\epsilon, \mathcal{H}, L_2(P_n)) \le \mathcal{N}(\epsilon^2/(2B), \mathcal{H}, L_1(P_n))$, where $\| \cdot \|_{L_2(P_n)} := \sqrt{\frac{1}{n}\sum_{i=1}^n (f(z_i) - g(z_i))^2}$, and $\| \cdot \|_{L_1(P_n)} := \frac{1}{n}\sum_{i=1}^n |f(z_i) - g(z_i)|$.*

**Lemma D.4** (Stability Properties of Pseudo-dimension). *Let $\mathcal{H}$ be a class of real functions. Then*
*(i) $Pdim(a\mathcal{H} + z) = Pdim(\mathcal{H})$ for any constants $a, z \in \mathbb{R}$ with $a \ne 0$;*
*(ii) $\mathcal{N}_p(L_\sigma \epsilon, \sigma \circ \mathcal{H}, n) \le \mathcal{N}_p(\epsilon, \mathcal{H}, n)$ and $Pdim(\sigma \circ \mathcal{H}) \lesssim Pdim(\mathcal{H})$ for any $L_\sigma$-Lipschitz function $\sigma$;*
*(iii) $Pdim(g_0 \cdot \mathcal{H}) \le 2Pdim(\mathcal{H})$ for fixed function $g_0$.*

*Proof of Lemma D.4.* (i) Affine Transformation: If $\mathcal{H}$ shatters $C$ with witness $r_H$, then $a\mathcal{H} + z$ shatters $C$ with witness $ar_H + z$. If $a < 0$, the required labels are simply flipped, not affecting shatterability.

(ii) Composition: Let $\mathcal{C}$ be an $\epsilon$-cover for $\mathcal{H}$. For any $h \in \mathcal{H}$, $\exists c \in \mathcal{C}$ s.t. $d_p(h, c) \le \epsilon$. By Lipschitzness, $|\sigma(h(x)) - \sigma(c(x))| \le L_\sigma |h(x) - c(x)|$, thus $\sigma \circ \mathcal{C}$ is an $(L_\sigma \epsilon)$-cover for $\sigma \circ \mathcal{H}$, which implies $\mathcal{N}_p(L_\sigma \epsilon, \sigma \circ \mathcal{H}, n) \le \mathcal{N}_p(\epsilon, \mathcal{H}, n)$. Applying Definition D.2(i), Lemma D.3(ii,iii), we obtain $Pdim(\sigma \circ \mathcal{H}) \log(1/\epsilon) \lesssim \log \mathcal{N}_1(\epsilon, \mathcal{H}, n) \lesssim Pdim(\mathcal{H}) \log(L_\sigma/\epsilon)$, yielding the result for small $\epsilon$. Finally, we get $Pdim(\sigma \circ \mathcal{H}) \lesssim Pdim(\mathcal{H})$.

(iii) Multiplication: Let $\mathcal{G}_\Pi = g_0 \cdot \mathcal{H}$ and assume $\mathcal{G}_\Pi$ pseudo-shatters a set $C$ of size $m$ with witness $r$. Partition $C$ into $C^+ = \{x \in C | g_0(x) > 0\}$ and $C^- = \{x \in C | g_0(x) < 0\}$. On $C^+$, $\mathrm{sgn}(g_0 h - r) = \mathrm{sgn}(h - r/g_0)$; on $C^-$, it is $\mathrm{sgn}(h - r/g_0) \cdot (-1)$. It follows that $\mathcal{H}$ can pseudo-shatter the subsets $C^+$ and $C^-$ separately. Thus $|C| = |C^+| + |C^-| \le 2Pdim(\mathcal{H})$. $\qquad\square$

