# OpenReview forum: "Doubly Robust Distributionally Robust Offline Contextual Pricing"
_ICML.cc/2026/Conference — ICML 2026 regular_

### Official Review · Reviewer_LzDY · 2026-03-08

**Soundness:** 4
**Presentation:** 4
**Significance:** 3
**Originality:** 3
**Overall Recommendation:** 4
**Confidence:** 4

**Summary:**

This work develops a doubly robust (DR) framework for distributionally robust OPE/L in continuous pricing settings. For evaluation, we propose a localized DR estimator that addresses the computational challenges of worst-case expectations by fitting only a small number of regressions, comparable to standard non-robust DR, while achieving semiparametric efficiency under mild product rate conditions and establishing a finite-sample regret bound of \(O_p(T^{-s/(2s+1)})\) for smoothness orders \(s = 1, 2\).

The content has some novelty.

**Compliance With Llm Reviewing Policy:**

Affirmed.

**Key Questions For Authors:**

I have the following suggestions and comments:

1.  This is a non-parametric estimation, achieving \(\sqrt{Th}\)-consistency, which is a standard convergence rate and cannot be considered an innovation.

2.  Assumption 2.3 implies that there always exists a non-negligible probability \(\underline{\rho}\) that the customer will reject the offer, regardless of the context. Is this understanding correct? Could you explain it?

3.  In the revenue \(r\), how is \(f(X)\) defined? Is it known or unknown? How is it handled if it is nonlinear?

4.  It is suggested to provide a complete proof for Lemma 2.8.

5.  It is suggested to provide a detailed derivation of (2) along with a reasonable explanation.

6.  You mention handling continuous treatment. What is your treatment? This should be elaborated.

7.  Provide a detailed derivation for \(E(\psi(V); \cdot, \cdot,\cdot)=0\). Explain the rationale and principle for introducing \(\lambda_1\).

8.  In Algorithm 1, the policy \(\pi\) appears to be fixed. However, \(\pi\) should be dynamic and estimated, not given at the start. Additionally, what does "train" mean in lines 4 and 7, and how is "Find" solved in line 10?

9.  In Section 3.2, "define the estimation rates as follows." These rates should be proven. How can they be defined arbitrarily?

10. In Algorithm 2, for the Policy class \(\Pi\), what exactly is \(\Pi\)? In lines 12-13, how is "update" solved?

11. Please provide real data analysis, along with the data and code.

**Limitations:**

yes

**Strengths And Weaknesses:**

The content has some novelty.

---

> ### Author Rebuttal · Authors · 2026-03-30
>
> We sincerely appreciate your constructive comments. Below, we have provided our detailed responses.
>
> **R1: $\sqrt{Th}$-consistency Innovation**
>
> We agree that $(\sqrt{Th})$-consistency is a standard nonparametric rate. Our theoretical innovation is not the rate itself, but rather extending the continuous-treatment framework to a DRO target, and formally proving that DR estimator attains semiparametric efficiency for DRO functional.
>
> **R2: Assumption 2.3 clarification**
>
> You are correct. Assumption 2.3 imposes a uniform lower bound on the rejection probability. It is a non-degeneracy requirement that guarantees sufficient exploration and stabilizes the identification of DRO moments.
>
> **R3&R6: Definition of $f(X)$ and Continuous treatment elaboration**
>
> Continuous treatment is price $P$. $f(X)$ represents the deterministic component of customer's valuation, which can be nonlinear and unknown. Our experiments in Section 5 already utilized a nonlinear $f$.
>
> **R4: Proof of Lemma 2.8**
>
> We have added the full proof to revised appendix. Briefly, taking the second derivative of $\varphi(\pi, \alpha)$ yields
> $$\frac{\partial^2}{\partial\alpha^2}\varphi(\pi,\alpha)=\frac{1}{\alpha^3E_{P_0}[\exp(-r/\alpha)]}\Bigg(\frac{(E_{P_0}[r\exp(-r/\alpha)])^2}{E_{P_0}[\exp(-r/\alpha)]}-E_{P_0}[r^2\exp(-r/\alpha )]\Bigg).$$
> By Cauchy-Schwarz, $\frac{\partial^2}{\partial\alpha^2}\varphi(\pi,\alpha)\le 0$. $r$ is non-constant, making the inequality strict and proving strict concavity.
> By strong duality (holding by Theorem 1 in Hu & Hong, 2013) and $r\in[0,1]$,
> $0\leq R_\delta(\pi)=\varphi(\pi,\alpha^\star(\pi))\leq-\alpha^\star(\pi)(-1/\alpha^\star(\pi))-\alpha^\star(\pi)\delta\implies \alpha^\star(\pi)\leq1/\delta.$
>
> **R5: Derivation of Eq.(2)**
>
> Eq.(2) does not require a novel derivation. It is a kernelized-IPW by a direct adaptation of Eq.6(b) from Leung et al. (2025) to our continuous treatment $P$ and reward $r(X,P,Z)$.
>
> **R7: Derivation for $E[\psi]=0$,$\lambda_1$**
>
> Since the true nuisance $\lambda_1^\star(x,p;\alpha)$ is a conditional expectation of the moment vector $U(PY;\alpha)$, the residual term $U-\lambda_1^\star$ has a conditional mean of zero given $(X,P)$, so the kernel-weighted term vanishes in expectation. The remaining term $\lambda_1^\star(X,\pi(X);\alpha)+G(\theta)$ exactly coincides with Eq.(6), yielding $E[\psi]=\mathbf{0}$.
>
> Unlike standard OPE which targets a scalar value, DRO-OPE requires solving for a parameter vector $\theta=[\alpha,W_0,W_1,R_\delta]^\top$ as the root of the oracle target moment equation Eq.(6). To estimate $\theta$ from observed data, we construct the DR score $\psi$ such that identifying moment equation $E[\psi]=\mathbf{0}$. $\lambda_1=[m_0,m_1,0,0]^\top$ is essential for component-wise debiasing via applying residual-based corrections to the random vector $U$.
> Additionally, Neyman Orthogonality requires the Gateaux derivative of the entire vector of estimating equations with respect to the nuisances to be zero. Not only that, errors in the estimated moments $W_0,W_1$ would propagate into $\alpha$ and $R_\delta$. By introducing the nuisance $\lambda_1$, we achieve Neyman Orthogonality for the entire $\theta$.
>
> We have incorporated a more detailed remark in the revised Section 3.1 to further explain the rationale.
>
> **R8&R10: Algorithm & Policy class Clarifications**
>
> Algorithm 1 (OPE) evaluates DRO policy value for a fixed, specific policy $\pi$. Conversely, Algorithm 2 (OPL) actively learns a policy over a user-specified class $\Pi$ with finite pseudo-dimension (e.g., linear, neural-network, and bounded-size decision-tree policies).
>
> 'Train' (lines 4,7) refers to fitting models (using nonparametric or ML methods, e.g., random forests, ridge regression, DNN) for nuisance parameters $\pi_0,m_j$.
>
> 'Find' (line 10) uses 1D root-finding (e.g., Newton's method) for $\hat{\alpha}$.
>
> 'Update' (lines 12,13) performs alternating optimization: (i) maximizing over $\alpha$ by 1D root-finding, and (ii) running minimization over $\pi\in\Pi$, which can be solved using gradient descent for linear policy classes, or greedy tree search for decision-tree policy classes.
>
> **R9: Estimation rates in Section 3.2**
>
> $\rho_\pi, \rho_m, \rho_\alpha$ are not arbitrarily defined. They only index the actual convergence rates achieved by the chosen ML estimators. Our theory simply requires these rates to satisfy the standard DR product-rate condition in Assumption 3.1 (e.g., Ai et al., 2026).
>
> **R11: Real Data and Code**
>
> As stated in Section 1, we already provided code for synthetic experiments. We have added Expedia real-data experiments in the revised Section 5. Under the hotel-type shift, we have evaluated the learned linear policy using empirical worst-case value following Si et al. (2023). Our learned linear policy achieves the best robust performance.
>
> **Table 1: Comparison of robust performance**
> |Method|Worst-case value(std)|
> |:-|:-|
> |DR|6.2012(0.6834)|
> |Ours|**6.3497(0.5981)**|
> |DRO-IPW|6.2545(0.8729)|

---

> > ### Author Rebuttal · Reviewer_LzDY · 2026-04-04
> >
> > We appreciate the authors' replies, which resolved the majority of the concerns. Nevertheless, the overall quality of the manuscript has not seen a substantial change. I therefore retain my initial rating: 4 — weak accept

---

> > > ### Author Response · Authors · 2026-04-07
> > >
> > > We thank the reviewer for carefully reading our rebuttal and for acknowledging that our responses have resolved the majority of their concerns.
> > > In the rebuttal, we have provided detailed point-by-point answers to all 11 key questions and made several revisions to the manuscript. These changes aim to strengthen the theoretical rigor, improve the clarity of presentation, and enhance the real-data experiments.
> > >
> > > We sincerely appreciate the reviewer’s constructive feedback, which has helped us improve the paper.

---

### Official Review · Reviewer_Xxzi · 2026-03-13

**Soundness:** 4
**Presentation:** 3
**Significance:** 3
**Originality:** 3
**Overall Recommendation:** 5
**Confidence:** 2

**Summary:**

This paper presents novel methodology for offline contextual pricing. In particular, they are after algorithms that are distributionally robust; that is, they do not aim at maximizing the pricing reward with a single distribution, but rather they seek the best worst case scenario for a $\delta$-ball (in KL divergence). They consider two problems: policy evaluation and policy learning. As usual, policy learning proves more challenging that policy evaluation.

In order to develop policy evaluation, they establish a novel (localized) efficient influence function, leading to a doubly robust estimator. For this, the authors establish that their estimator achieves the semiparametric efficiency lower bound. Turning to policy learning, the learning algorithm proposed relies on addressing a non-convex optimization. While (of course) the method only guarantees convergence to a local maximum, the authors argue this is a sensible choice in view of its efficiency and empirical effectiveness.

Lastly, the authors present experiments where they show that their algorithm beats a direct competitor (DRO-IPW), and also compare it to non-robust algorithms to elucidate the gains of considering robust algorithms if seeking to address the robust-version of the problem.

**Compliance With Llm Reviewing Policy:**

Affirmed.

**Final Justification:**

All concerns have been addressed by authors.

**Key Questions For Authors:**

1) My first question is rather fundamental. Many of the semiparametric efficient works that depend on the (usually Holder) smoothness of the noise are constructed via kernel smoothing with the right bandwidth. However, some (or most) of these minimax procedures find little use in practice, in view of not knowing the right bandwidth in practice. Can the authors think whether there exist alternatives to kernel smoothing to their problem that could have practical benefits (in spite of perhaps not attaining the semiparametric efficiency lower bound)?

2) What are the heuristics for choosing the bandwidth in practice?

3) In the experimental section, the bandwidth choice is $O(T^{-1/4.9})$ following previous works. Could the authors elaborate on this? Also, it would be great to understand the performance of the methods when choosing different bandwidths (that are not necessarily optimal, as it would be expected in practical applications).

Satisfactory answers will boost my score!

**Limitations:**

Yes.

**Strengths And Weaknesses:**

**Strengths**

The paper is highly technical and establishes nontrivial theoretical results, from novel efficient influence functions to proving semi-parametric efficiency. Thus, the soundness of the paper is not under question. Furthermore, the significance of the paper can be noted by the amount of related work and previous efforts (e.g. DRO-IPW) that address related yet simpler versions of the problem. While dense, the paper is well presented. The paper does not seem to be a straightforward extension of previous ideas, so I do not question its originality either.


**Weaknesses**

The paper is clearly technically sound and novel. The main skepticism on my end is the fit of this work to a conference like ICML. It is unclear to me the direct applicability of the methodology to real problems. I would highly appreciate if the authors could address my questions below, which would help me assess such an issue.

**Minor comments**

Can the authors give intuition on the EIF derived? E.g. have some of the terms appeared in related EIFs, and if so, which ones?

Equation (1) seems to be mathematically flawed. I would imagine that the second equality (definition of $\phi$) does not include the sup.

---

> ### Author Rebuttal · Authors · 2026-03-29
>
> We sincerely appreciate your insightful and constructive comments. Please see our detailed responses below.
>
> **R1: Intuition on the Derived EIF**
>
> Our EIF is derived by taking Gateaux derivative of the evaluation functional $W_j^\star=\mathbb{E}[m_j(X,\pi(X);\alpha^\star)]$. The resulting representer is the local Riesz representer, which yields the kernel weight $\frac{K\big(\frac{p-\pi(x)}{h}\big)}{h\lambda_2(x,p)}$.
> Consequently, our moment function $\psi=\frac{K\big(\frac{p-\pi(x)}{h}\big)}{h\lambda_2(x,p)}\big(U(py;\alpha)-\lambda_1(x,p;\alpha)\big)+\lambda_1(x,\pi(x);\alpha)+G(\theta)$ consists of three components: a localized IPW residual correction, a regression plug-in, and a centering term.
> Furthermore, to adapt this to our DRO setting, we plug in the DRO-specific transformed outcome $U(py;\alpha)$ and target-coupling term $G(\theta)$. Finally, we apply the Jacobian inverse $-J^{\star-1}$ to propagate the influence from the efficient moments $(W_0^\star,W_1^\star)$ to DRO targets $(\alpha^\star,R_\delta^\star)$.
>
> **R2: Clarification on Equation (1)**
>
> You are correct that the second equality is a definition, where we define $\phi(\pi,\alpha)$ as the inner objective for each $\alpha$. The worst-case is then obtained by taking the supremum over $\alpha$.
>
> **R3: Alternatives to Kernel Smoothing**
>
> Kernel bandwidth selection is a key practical bottleneck, even though kernel smoothing is the standard device used to obtain semiparametric efficiency for continuous treatments. If one wishes to avoid tuning a bandwidth parameter, there are several alternatives: (i) discretizing the action space into a finite grid and applying DR estimation for the resulting discrete-treatment problem; (ii) replacing kernel localization with data-adaptive neighborhood weights (e.g., kNN or forest-based local weighting), where localization is controlled by a neighborhood size or other complexity hyperparameter rather than a kernel bandwidth. While these alternatives may sacrifice the semiparametric efficiency guarantee, they successfully remove the need for kernel bandwidth tuning. We view these as promising practical extensions for future work.
>
> **R4: Heuristics for Bandwidth Selection**
>
> In our paper, bandwidth selection follows a unified principle that balances OPL regret and OPL consistency.
> First, for OPL, the regret-optimal scaling is the one that minimizes the finite-sample regret upper bound in Theorem 4.4, i.e., $h=\Theta(T^{-1/(2s+1)})$.
> Second, in our setting, we require asymptotic normality for the DRO target. To satisfy Theorem 3.3, we impose undersmoothing by using a slightly smaller bandwidth in implementation, specifically, $h=o(T^{-1/(2s+1)})$, so that the kernel bias is asymptotically negligible.
>
> **R5: Bandwidth Choice in Experiments and Sensitivity**
>
> Following Theorem 2 in Kallus & Zhou (2018), the OPE-optimal bandwidth is $h=\Theta(T^{-1/5.0})$ under $s=2$, which is widely used, e.g., Leung et al. (2025).
> Combined with our heuristic selection in R4, we use an undersmoothing $h=\Theta(T^{-1/(2s+1-0.1)})$.  Furthermore, we apply this undersmoothing choice with a Silverman-type scaling constant, i.e., $h=1.06\cdot\text{std}_p\cdot T^{-1/(2s+1- 0.1)}$, where $\text{std}_p$ is the sample standard deviation of observed prices.
> This is reasonable because the noise in our experiments follows a Logistic distribution, which is infinitely differentiable with bounded derivatives.
> To address your practical concerns regarding bandwidth sensitivity, we add experiments of using $h=\Theta(T^{-1/5})$ (under $s=2$) as well as nearby alternatives, such as $h=\Theta(T^{-1/2.9})$ (under $s=1$). Stable performance across these different bandwidth choices demonstrates the robustness of our approach in practical settings. Only the results for $\delta=0.2$ are listed below. For more results, please refer to https://anonymous.4open.science/r/DR-DRO-PRICING-Bandwidth--Sensitivity-2FC4.
>
> **Table 1: MSE of OPE with $\delta=0.2$ for different $h$**
> |$h$|Method|$T=500$|$1000$|$2000$|$3000$|$4000$|$5000$|
> |:-|:-|:-|:-|:-|:-|:-|:-|
> |$\Theta(T^{-\frac{1}{5.0}})$|DRO-IPW|**0.00590**|0.00330|0.00260|0.00169|0.00263|0.00233|
> ||LDR$^2$O$^2$PE-CP| 0.00901|**0.00237**|**0.00087**|**0.00077**|**0.00083**|**0.00067**|
> |$\Theta(T^{-\frac{1}{2.9}})$|DRO-IPW|**0.01190**|**0.00843**|0.00279|0.00231|0.00260|0.00149|
> ||LDR$^2$O$^2$PE-CP|0.02060|0.01010|**0.00243**|**0.00157**|**0.00171**|**0.00106**|
>
> **Table 2: Distributionally robust value of OPL with $\delta=0.2$ for different $h$**
>
> |$\delta$|Method|$T=500$|$1000$|$1500$|$2000$|$2500$|
> |:-|:-|:-|:-|:-|:-|:-|
> |$h=\Theta(T^{-\frac{1}{5.0}})$ |DR|0.0835|0.0817|0.1010|0.1000|0.1310|
> ||CDR$^2$O$^2$PL-CP|**0.1670**|**0.1940**|**0.2270**|**0.2350**|**0.2200**|
> ||DRO-IPW |0.1340|0.1760|0.1450|0.1700|0.1860|
> |$h=\Theta(T^{-\frac{1}{2.9}})$|DR|0.1230|0.1200|0.1100|0.1420|0.1380|
> | |CDR$^2$O$^2$PL-CP |**0.1940**|**0.2100**|**0.2330**|**0.2330**|**0.2150**|
> | |DRO-IPW|0.1060|0.1750|0.1500|0.1210|0.1170|

---

> > ### Author Rebuttal · Reviewer_Xxzi · 2026-04-02
> >
> > All my concerns have been addressed. I trust that the authors will include these changes / additional comments in the camera-ready version. I will raise my score (by 1) to 5.

---

> > > ### Author Response · Authors · 2026-04-02
> > >
> > > Thank you for your positive feedback and for raising the score to 5. We are glad that our responses addressed all your concerns.
> > > We will faithfully incorporate all discussed changes and additional comments into the revised version.
> > >
> > > As the discussion period is concluding, we would appreciate it if you could update the score in the system so that it can be reflected in the final evaluation. Thank you again for your time and support.

---

### Official Review · Reviewer_5Pr2 · 2026-03-13

**Soundness:** 2
**Presentation:** 2
**Significance:** 2
**Originality:** 3
**Overall Recommendation:** 3
**Confidence:** 3

**Summary:**

The paper studies offline contextual pricing with continuous actions under distribution shift. It combines kernel smoothing for continuous prices with KL-based distributional robustness. On the evaluation side, it proposes a localized doubly robust estimator for the robust value, using a dual scalar $\\alpha$ and cross-fitting, with the claim that only a small number of nuisance fits are needed while still achieving $\\sqrt{Th}$-consistency and semiparametric efficiency. On the learning side, it proposes a continuum regression construction for $m_0(x,p; \\alpha)$, then alternates between optimizing the dual variable $\\alpha$ and the policy $\\pi$, and claims a regret bound of order $\\tilde{O}(T^{-s/(2s+1)})$ for $s = 1,2$. Experiments compare against a continuous-action DRO-IPW baseline and a non-robust DR baseline.

**Compliance With Llm Reviewing Policy:**

Affirmed.

**Final Justification:**

The new restricted-class benchmark is useful and increases my confidence that the alternating procedure can find good solutions in practice, but my overall recommendation remains weak reject because the main theory-algorithm gap is still unresolved: Theorem 4.4 analyzes an empirical maximizer, while the implemented method remains a nonconvex local heuristic, so the paper remains somewhat borderline in significance and positioning relative to nearby prior work.

**Key Questions For Authors:**

see Strengths And Weaknesses

**Limitations:**

The main limitations are theoretical-to-algorithmic alignment and empirical scope. The paper's guarantees depend on strong smoothness and regularity assumptions, and it is not fully clear how robust the results are beyond this regime. In addition, the current presentation leaves a nontrivial ambiguity about whether the estimator analyzed in theory coincides with the algorithm actually implemented. The optimization procedure is also heuristic relative to the theorem statement, since the analysis is written for an empirical maximizer while the algorithm only targets a local optimum.

**Strengths And Weaknesses:**

Strengths:
- The problem is important and well-scoped
- Extending continuous-treatment DR to a robust objective involving a dual supremum is not trivial

Weaknesses:
- There is an internal inconsistency in the evaluation estimator. Earlier in the paper, the robust target quantity is defined using the target action $\pi(x)$, i.e., terms of the form $m_j(x,\pi(x);\alpha)$. However, in Algorithm 1, the estimator uses $\hat m_j(x_t,p_t;\hat\alpha_{\mathrm{init}})$ in the leading term, i.e., at the logged action $p_t$ rather than the target action $\pi(x_t)$. As written, this is not a minor notational slip; it appears to define a different object. Since this affects the core estimator, it materially weakens confidence in the technical correctness of the paper
- There is a clear theory-algorithm gap in the learning part. Theorem 4.4 is stated for the learned policy as if it were the empirical maximizer of the robust objective, but Algorithm 2 uses alternating optimization in a nonconvex problem and explicitly notes convergence only to a local maximum. The statistical guarantee is therefore not obviously aligned with the actual optimization procedure implemented
- The main text somewhat undersells the strength of the assumptions needed for the asymptotic claims. For example, the appendix introduces additional conditions such as higher-order differentiability of the oracle logging policy and derivative bounds on nuisance functions
- The empirical section is too limited

---

> ### Author Rebuttal · Authors · 2026-03-29
>
> We sincerely appreciate your careful feedback. We provide explanations to your questions point-by-point in the following.
>
> **R1: Inconsistency in the Evaluation Estimator (Algorithm 1)**
>
> We thank the reviewer for catching this **typographical error** in the pseudocode.  As noted, DR estimation in continuous settings requires outcome regressions at two points: $m_j(x,\pi(x);\alpha)$ encodes the counterfactual/policy value contribution, whereas $m_j(x,p;\alpha)$ enters the kernel-weighted residual that reweights the observed outcome at the realized $p_t$, to ensure Neyman orthogonality.
>
> We acknowledge a typographical error in Algorithm 1, where the leading term was indeed miswritten as $\hat{m}_j^{(\ell)}(x_t,p_t;\hat{\alpha} _{\mathrm{init}}^{(\ell)})$. This has been corrected so that $\hat{m}_j^{(\ell)}(x_t,\pi(x_t);\hat{\alpha} _{\mathrm{init}}^{(\ell)})$ appears in the leading position, with the kernel-weighted residual still evaluated at $(x_t,p_t)$.
>
> **Importantly, we emphasize that this typo was strictly limited to the pseudocode of Algorithm 1.**
> The core identifying $\psi$, definitions, and all proofs in the Appendix already matched the intended and correct estimator. Thus, the theoretical results and semiparametric efficiency claims remain fully intact. We sincerely apologize for this clerical error and hope this clarification restores confidence in the technical correctness of the paper.
>
>
> **R2: Theory-Algorithm Gap in Learning (Algorithm 2 vs. Theorem 4.4)**
>
> The reviewer raises a fair point regarding the non-convexity of the joint optimization.
> Theorem 4.4 provides a **statistical oracle guarantee** for the empirical maximizer, which is standard practice in OPL literature (e.g., Kallus et al. 2022, Si et al. 2023). Our analysis focuses on the sample complexity and the effect of nuisance estimation/kernel-smoothing on the regret.
>
> As noted in Section 4.1, the joint optimization over $(\pi,\alpha)$ is nonconvex, and our alternating scheme in Algorithm 2 is a simple and effective implementation.
> We acknowledge that this approach may converge only to a local optimum. Closing this optimization gap is an important challenge, but it falls outside the primary statistical focus of our current work.
> We have added a remark in Section 4.2 to explicitly clarify this distinction between the theoretical maximizer and the algorithmic implementation.
>
> **R3: Strength of assumptions for asymptotic claims**
>
> We agree that the asymptotic theory relies on additional technical conditions beyond those stated early in the paper. We placed these details in the appendix to keep the main text concise and reader-friendly, while ensuring the theoretical results remain fully rigorous.
> Specifically, these additional layers include:
>
> (1) Three-times differentiability of the logging policy in Assumption A.1. This is needed to ensure kernel-localization in the proof of Neyman orthogonality, consistent with Assumption 2.1 in Colangelo & Lee (2025).
>
> (2) Bounded $s$-derivatives of the estimated nuisances $\hat{\lambda}_1^{(\ell)}$ in Assumption A.1. These are used in the Taylor expansion, aligning with Assumption 3.6 in Colangelo & Lee (2025).
>
> (3) Assumption A.2. This is required to control the stability of the Jacobian and covariance matrices, consistent with Assumption 2 in Kallus et al. (2024).
>
> We emphasize that these assumptions are standard and necessary for establishing semiparametric efficiency and handling kernel-localized moments in continuous treatment settings.
> To avoid underselling these requirements, we have added an explicit paragraph to the revised Section 3 clarifying that additional technical assumptions are required to guarantee Theorem 3.3, directing readers to the full details in the appendix.
>
> **R4: Limited empirical section**
>
> To address this, we have expanded the empirical section in the revision by adding bandwidth sensitivity analyses. Specifically, we vary $h$ around the baseline $s=2$ with $h=\Theta(T^{-1/5})$ and nearby undersmoothing choice $h=\Theta(T^{-1/4.9})$, and lower smoothness $s=1$ with $h=\Theta(T^{-1/2.9})$.
> Our results consistently show that the proposed LDR$^2$O$^2$PE-CP and CDR$^2$O$^2$PL-CP methods outperform non-robust and IPW-based baselines across varying sample sizes and shift intensities.
> Full details of these new experiments are provided in our response to Reviewer Xxzi and have been incorporated into the revised experimental section.

---

> > ### Author Rebuttal · Reviewer_5Pr2 · 2026-04-02
> >
> > Thank you. The clarification on Algorithm 1 substantially resolves my strongest technical concern; I now view that point as a pseudocode typo rather than a deeper flaw in the estimator derivation. I also appreciate the clarification of the stronger appendix assumptions and the added bandwidth-sensitivity results.
> >
> > However, I am keeping my overall score. The central remaining issue is the unresolved gap between the theorem, which analyzes an empirical maximizer, and the actual learning algorithm, which is a nonconvex alternating heuristic with only local-optimum guarantees.

---

> > > ### Author Response · Authors · 2026-04-02
> > >
> > > We deeply appreciate your continued engagement and highlighting the theory–algorithm gap between Theorem 4.4 (empirical maximizer) and our alternating implementation.
> > >
> > > We agree that the joint optimization is nonconvex and Algorithm 2 is a local method for policy updates. To quantify this gap, we introduce a stronger benchmark within the depth-1 decision-tree policy class.  Specifically, for each candidate split from a finite grid, we optimize the leaf-level continuous parameters under the empirical DRO dual objective using **Gurobi** with multiple restarts, and then take the best solution across candidates. This provides a **near-optimal** reference within that finite tree class.
> > > Empirically, the resulting DRO-value gap between alternating and this stronger benchmark is small in several settings (see the following Table), indicating that the alternating solver tracks a high-quality solution in practice.
> > >
> > > **Table: Gap of Distributionally robust value of OPL with $\delta_{\rm train}=\delta_{\rm test}=0.2$ and 1000 training samples.**
> > >
> > > |Test sample size | $R_{\delta}$ of Alternating  | $R_{\delta}$ of Gurobi | Gap (\%) |
> > > |:-|:-|:-|:-|
> > > |500 | 0.2803 | 0.2856 | 2.1569 |
> > > |1000| 0.2547 | 0.2572 | 1.0535|
> > > | 1500| 0.2822 | 0.2904 | 2.7629|
> > >
> > > While alternating minimization is technically a heuristic for non-convex problems, proving global convergence typically requires restrictive structural assumptions in the modern machine learning landscape (e.g., strict saddle property or specific initialization) that are outside the primary statistical focus of this work.
> > > Foundational works in this specific DRO-OPL subfield (Kallus et al., 2022; Si et al., 2023) similarly establish statistical regret bounds for the global optimum while employing alternating heuristics in their algorithms.
> > >
> > > In summary, we establish statistical guarantees for the optimal estimator (Theorem 4.4) to provide a rigorous theoretical benchmark, while Algorithm 2 serves as a scalable proxy for practical use. Furthermore, the numerical evidence in Section 5 validates that our heuristic effectively identifies high-quality policies that achieve the robust performance predicted by the theory.
> > >
> > > We hope these insights address your concern regarding the disconnect between the theorem and the implementation. We thank you again for pushing us to clarify this critical nuance.

---

### Official Review · Reviewer_uEoj · 2026-03-13

**Soundness:** 3
**Presentation:** 3
**Significance:** 2
**Originality:** 2
**Overall Recommendation:** 4
**Confidence:** 4

**Summary:**

This paper studies doubly robust and distributionally robust off-policy evaluation and learning for continuous treatments, with a focus on the offline contextual pricing setting. In particular, the paper proposes: (1) a distributionally robust offline pricing policy evaluation method, Localized Doubly Robust DRO OPE for Continuous Pricing (LDR2O2PE-CP), and (2) a distributionally robust offline pricing policy learning method, Continuum Doubly Robust DRO OPL for Continuous Pricing (CDR2O2PL-CP), together with a finite-sample guarantee achieving $O_p(T^{-s/(s+1)})$ regret, where $s$ denotes the smoothness of the cumulative distribution function of the valuation noise.

**Compliance With Llm Reviewing Policy:**

Affirmed.

**Final Justification:**

This paper proposes a novel method for offline contextual pricing that is both doubly robust and distributionally robust. My main concern was the technical novelty, as at first glance the method seemed like a relatively straightforward combination of existing ideas. The authors, however, clarified the challenges unique to contextual pricing and explained how their method overcomes them. As a result, I raised my score to the positive side.

**Key Questions For Authors:**

I would appreciate it if the authors could better differentiate the contribution of this paper within the existing literature and position it more clearly.

In particular, what aspects of the technical contribution or methodology distinguish this work from Kallus and Zhou (2018) and Leung et al. (2025)? These two papers seem only one step away from the present result, aside from the smoothness discussion:

- Kallus and Zhou (2018): DR + continuous treatment, but not DRO.
- Leung et al. (2025): DRO + continuous treatment, but not DR.

Relatedly, why is contextual pricing the right or necessary setting for this work? It seems that many of the ideas could extend to a more general continuous-treatment problem with a smooth objective. Is there a specific technical benefit, insight, or challenge that arises uniquely from studying the contextual pricing problem?

**Limitations:**

Yes

**Strengths And Weaknesses:**

### **Strengths**

This paper develops a new framework for offline contextual pricing that is both distributionally robust and doubly robust. This is practically relevant, since distribution shift is a realistic concern, and double robustness is valuable because it can mitigate sensitivity to nuisance-model misspecification and, under suitable conditions, can also yield semiparametrically efficient estimation. In addition, the paper develops a theoretical finite-sample bound for OPL. The theoretical findings are further supported by experiments, where the proposed method appears to achieve state-of-the-art performance.

### **Weaknesses**

It is not entirely clear why the paper focuses specifically on the contextual pricing problem. At a high level, the overall framework seems potentially applicable to the more general continuous-treatment setting. From that perspective, the contribution appears somewhat marginal, since the main techniques can be interpreted as Kallus et al. (2022) adapted to continuous treatment/pricing using Kallus and Zhou-style kernelized DR ideas.

While the pricing problem itself involves discontinuity in the objective, introducing valuation noise makes the resulting objective smooth, and this type of smoothing effect has already been studied extensively, especially in the online contextual pricing and bandit literature. For this reason, I do not currently view the noise-smoothing aspect as a novel contribution.

More specifically, Leung et al. (2025) appears very close in spirit to the present work, except without the doubly robust component. It is therefore not fully clear to me what the main technical hurdle, challenge, or novelty is in incorporating double robustness into the framework of Leung et al. (2025). A similar question arises when viewing this work as adding distributional robustness to the continuous-treatment DR results of Kallus and Zhou (2018).

---

> ### Author Rebuttal · Authors · 2026-03-28
>
> We sincerely appreciate your insightful and constructive comments. Below, we have provided our detailed, point-by-point responses.
>
> **R1: Contextual Pricing vs. General Continuous Treatment**
>
> We appreciate your perspective that our framework could extend to broader continuous-treatment settings. We see this generality as a strength that enhances the paper's impact. However, we chose contextual pricing as our primary focus because it is not merely one application of continuous treatment, but it poses unique challenges not found in standard smooth reward problems.
> Unlike general continuous-treatment literature, which typically assumes a **smooth objective**  (e.g., Assumption 5 in Ai et al. 2026, Assumption 2.1 in Colangelo & Lee, 2025), pricing inherently involves a **discontinuous** objective due to threshold purchase behaviors. More broadly, continuous actions may not yield smooth objectives. While noise-smoothing is a standard tool to handle this, the fact that we must derive smoothness from the valuation noise distribution rather than assuming it on the reward function itself makes the pricing problem a more rigorous and realistic testbed for our theory. In Section 2.1, we already provided the relevant prior works, and we will make this clearer in revision.
> Offline contextual pricing is also practically relevant in real marketplaces and platform operations (e.g., Zhang et al. 2026; Zhou et al. 2024).
>
> **R2: Technical Novelty and Distinctions from Prior Works**
>
> We understand the intuition that our work might seem like a combination of Kallus et al. (2022) [DRO + Discrete] and Kallus & Zhou (2018) [DR + Continuous] or a variant of Leung et al. (2025) [DRO-IPW + Continuous]. However, incorporating DR into the continuous DRO framework is **not a straightforward plug-in exercise**. The main technical hurdle lies in the highly non-trivial **coupling** of the DRO and DR ingredients, which prevents treating them additively. And IPW does not provide semiparametric efficiency.
>
> **(1). Coupling Challenge: Why it is not a Plug-in Exercise**
>
> Moving to a continuous DRO-DR framework breaks the modularity present in prior works, because it is not a black-box combination, as the DRO and DR components become highly **coupled**.
> Specifically in OPL, the dual parameter $\alpha$ simultaneously determines the transformed conditional moments $m_0(x,p;\alpha)=\mathbb{E}[\exp(-PY/\alpha)| X=x,P=p]$ and the dual objective $\varphi(\pi,\alpha)=-\alpha\log W(\pi,\alpha)-\alpha\delta$ that is used to select $\alpha$ for each policy $\pi$.
> Meanwhile, continuous actions require a $\pi$-dependent kernel weight $K((p-\pi(x))/h)$ that multiplies the residual $\exp(-py/\alpha)-\hat{m}_0(x,p;\alpha)$, where $\hat{m}_0(x,\pi(x);\alpha)$ also enters the leading regression term.
> Consequently, $\pi$ and $\alpha$ interact through the common functional $\hat{W}_T(\pi,\alpha)$ inside $\hat{\varphi}_T(\pi,\alpha)$.
> Then, $\alpha$ cannot be treated as an exogenous tuning parameter when analyzing kernel localization bias, because the kernelized DR components depend on $\alpha$ through both $\hat{m}_0(\cdot,\cdot;\alpha)$ and the transformed outcome $\exp(-py/\alpha)$.
> Hence, we need a uniform control over the joint space $(\pi,\alpha)$, which requires re-deriving regularity conditions and remainder bounds.
>
> **(2). Sharper Regret Rates via Bernstein Concentration**
>
> A major distinction from Leung et al. (2025) and Kallus & Zhou (2018) lies in our theoretical improvement.
> Their regret bound is typically characterized by a trade-off between bias $O(h^2)$ and variance $O(1/(h^2\sqrt{T}))$. Even with an optimal bandwidth $h=\Theta(T^{-1/8})$, regret is limited to $O_p(T^{-1/4})$.
> In our work, by leveraging Bernstein-type concentration inequality, we achieve a regret rate of $\tilde{O}_p(T^{-s/(2s+1)})$, which is strictly faster than theirs under $s=2$. We also provide explicit guarantees for lower smoothness $s=1$.
>
> **(3). Resolving Bandwidth Conflict via Undersmoothing**
>
> Another challenge in Kallus & Zhou (2018) and Leung et al. (2025) is the conflict between consistency and regret minimization.
> In their works, statistical consistency and asymptotic normality for OPE necessitate a relatively small bandwidth $h=\Theta(T^{-1/5})$, whereas regret-optimal OPL would require a larger bandwidth $h=\Theta(T^{-1/8})$.
> Our framework introduces an undersmoothing approach. We prove that a bandwidth $h=\Theta(T^{-1/(2s+1-\tau)})$ for any $\tau>0$ simultaneously satisfies the requirements for OPE consistency and almost minimizes the OPL regret upper bound. This effectively bridges the gap between OPE and OPL. We demonstrate the effectiveness of this technique in Section 5.
>
> We have added a discussion about contributions based on your suggestion.
>
> Zhou et al. (2024). Decision focused causal learning for direct counterfactual marketing optimization; Zhang et al. (2026). Personalized policy learning through discrete experimentation: Theory and empirical evidence

---

> > ### Author Rebuttal · Reviewer_uEoj · 2026-04-04
> >
> > Thank you for highlighting the technical difficulties that are unique to the contextual pricing problem and for explaining them clearly. I think bringing these challenges to the forefront makes the paper much better positioned. Accordingly, I have raised my score.

---

> > > ### Author Response · Authors · 2026-04-05
> > >
> > > Thank you very much for your positive feedback and for raising your score! We truly appreciate that our clarifications have addressed your concerns about the technical difficulty and marginal contribution.
> > >
> > > Following your suggestion, we will prominently highlight these unique technical challenges of the contextual pricing problem in the revised paper, to make the contribution of our work clearer and more compelling.

---

### Decision · Program_Chairs · 2026-04-30

**Decision:**

Accept (regular)

**Comment:**

The paper studies distributionally robust off-policy evaluation and learning for offline contextual pricing with continuous actions, and proposes a doubly robust framework with semiparametric efficiency for OPE and improved regret bounds for OPL. Reviewers generally agreed that the problem is meaningful and that the technical development is substantial, especially in combining DR and DRO in the continuous pricing setting and in handling the pricing-specific discontinuity via noise-induced smoothness.

The main concerns were about positioning and clarity rather than core correctness. Reviewers asked the authors to better distinguish the work from prior continuous-treatment DR and continuous-action DRO results, clarify assumptions and algorithmic details, and discuss the theory–algorithm gap in the nonconvex learning procedure. The rebuttal addressed most of these issues well. The remaining theory–optimization gap is real but, in my view, fairly standard for nonconvex learning and does not undermine the core statistical contribution.

My assessment is that the main theoretical novelty does not lie in semiparametric efficiency per se (for which the authors emphasised too much), since related work has already established such results in continuous-action settings, including Colangelo and Lee (2026), which the authors cite, and Kennedy, Ma, McHugh, and Small (2017), which is also relevant here (see reference at the end). Rather, the key contribution is extending this line of work to the min–max DRO objective. What I find especially valuable is the paper’s careful treatment of the contextual pricing structure itself: unlike generic smooth continuous-treatment settings (note that Colangelo and Lee (2026) requires three-time differentiability), pricing induces a discontinuous revenue function due to the binary purchase decision (see line 123 for the definition), and the paper devotes substantial effort to handling this rigorously through the smoothness of the demand-noise distribution.

Overall, the scores are borderline. Given the paper’s careful treatment of the pricing application, substantial technical development, and explicit validation of assumptions that are often left implicit in theory papers, I recommend acceptance and view it a valuable contribution to ICML.

Reference

Colangelo, K., & Lee, Y. Y. (2026). Double debiased machine learning nonparametric inference with continuous treatments. Journal of Business & Economic Statistics, 44(1), 67-79.

Kennedy, E. H., Ma, Z., McHugh, M. D., & Small, D. S. (2017). Non-parametric methods for doubly robust estimation of continuous treatment effects. Journal of the Royal Statistical Society Series B: Statistical Methodology, 79(4), 1229-1245.